# Methionine adenosyltransferase 1a antisense oligonucleotides activate the liver-brown adipose tissue axis preventing obesity and associated hepatosteatosis

Diego Sáenz de Urturi[1], Xabier Buqué[1,2], Begoña Porteiro[3], Cintia Folgueira[4], Alfonso Mora[4], Teresa C. Delgado[5], Endika Prieto-Fernández[6], Paula Olaizola[7], Beatriz Gómez-Santos[1], Maider Apodaka-Biguri[1], Francisco González-Romero[1], Ane Nieva-Zuluaga[1], Mikel Ruiz de Gauna[1], Naroa Goikoetxea-Usandizaga[5], Juan Luis García-Rodríguez[1], Virginia Gutierrez de Juan[5], Igor Aurrekoetxea[1,2], Valle Montalvo-Romeral[4], Eva M. Novoa[3], Idoia Martín-Guerrero[6], Marta Varela-Rey[5,8,9], Sanjay Bhanot[10], Richard Lee[10], Jesus M. Banales[7,8,11,12], Wing-Kin Syn[1,13,14], Guadalupe Sabio[4], María L. Martínez-Chantar[5,8], Rubén Nogueiras[3,15,16] & Patricia Aspichueta[1,2,8✉]

Altered methionine metabolism is associated with weight gain in obesity. The methionine adenosyltransferase (MAT), catalyzing the first reaction of the methionine cycle, plays an important role regulating lipid metabolism. However, its role in obesity, when a plethora of metabolic diseases occurs, is still unknown. By using antisense oligonucleotides (ASO) and genetic depletion of *Mat1a*, here, we demonstrate that *Mat1a* deficiency in diet-induce obese or genetically obese mice prevented and reversed obesity and obesity-associated insulin resistance and hepatosteatosis by increasing energy expenditure in a hepatocyte FGF21 dependent fashion. The increased NRF2-mediated FGF21 secretion induced by targeting *Mat1a*, mobilized plasma lipids towards the BAT to be catabolized, induced thermogenesis and reduced body weight, inhibiting hepatic de novo lipogenesis. The beneficial effects of *Mat1a* ASO were abolished following FGF21 depletion in hepatocytes. Thus, targeting *Mat1a* activates the liver-BAT axis by increasing NRF2-mediated FGF21 secretion, which prevents obesity, insulin resistance and hepatosteatosis.

[1] Department of Physiology, Faculty of Medicine and Nursing, University of the Basque Country UPV/EHU, Leioa, Spain. [2] Biocruces Bizkaia Health Research Institute, Barakaldo, Spain. [3] Department of Physiology, CIMUS, University of Santiago de Compostela-Instituto de Investigación Sanitaria, Santiago de Compostela, Spain. [4] Myocardial Pathophysiology, Centro Nacional de Investigaciones Cardiovasculares (CNIC), Madrid, Spain. [5] Liver Disease Laboratory, CIC bioGUNE-BRTA (Basque Research & Technology Alliance), Derio, Spain. [6] Department of Genetics, Physical Anthropology and Animal Physiology, Faculty of Science and Technology, University of the Basque Country UPV/EHU, Leioa, Spain. [7] Department of Liver and Gastrointestinal Diseases, Biodonostia Health Research Institute—Donostia University Hospital, University of the Basque Country (UPV/EHU), San Sebastian, Spain. [8] National Institute for the Study of Liver and Gastrointestinal Diseases (CIBERehd, Instituto de Salud Carlos III), Madrid, Spain. [9] Gene Regulatory Control in Disease Laboratory, CIMUS, University of Santiago de Compostela-Instituto de Investigación Sanitaria, Santiago de Compostela, Spain. [10] IONIS Pharmaceuticals, Carlsbad, CA, USA. [11] Ikerbasque, Basque Foundation for Science, Bilbao, Spain. [12] Department of Biochemistry and Genetics, School of Sciences, University of Navarra, Pamplona, Spain. [13] Section of Gastroenterology, Ralph H Johnson, VAMC, Charleston, SC, USA. [14] Division of Gastroenterology and Hepatology, Medical University of South Carolina, Charleston, SC, USA. [15] CIBER Fisiopatología de la Obesidad y Nutrición (CIBERobn), Santiago de Compostela, Spain. [16] Galician Agency of Investigation, Xunta de Galicia, Spain. ✉email: patricia.aspichueta@ehu.eus

ncreased prevalence of obesity has led to a tsunami of metabolic diseases such as type 2 diabetes mellitus, dyslipidemia, and nonalcoholic fatty liver disease (NAFLD)[1].

NAFLD is the most common cause of chronic liver disease in Western countries[2]. It ranges from hepatosteatosis to nonalcoholic steatohepatitis (NASH), a more advanced form of the disease. There are still no specific pharmacological treatments for NAFLD but as obesity is a major risk factor, treatments that impact body weight and glucose control have been investigated[3].

In patients with obesity, where NAFLD prevalence is ~80%, serum levels of S-adenosylmethionine (SAMe) are elevated and correlate with abdominal adiposity, fat mass and higher calorie intake, suggesting the increased synthesis of liver SAMe[4,5]. SAMe is synthesized in the first reaction of the methionine cycle through the action of methionine adenosyltransferase (MAT), which uses methionine and ATP[6]. Obesity development associates with high methionine intake[7,8], which in turn, produces an increase of serum SAMe levels[9]. In concordance, dietary methionine restriction (MR) reduces adiposity and leptin levels, increases adiponectin release and recovers insulin sensitivity[10,11].

Deficiency in several enzymes involved in the methionine cycle confers resistance to obesity and the associated co-morbidities. In mice, lack of phosphatidylethanolamine N-methyltransferase (PEMT) prevents high-fat diet (HFD)-induced obesity and insulin resistance by promoting energy expenditure[12,13]. Nicotinamide N-methyltransferase (NNMT) regulates glucose and cholesterol metabolism[14]; however, knockdown of Nnmt also induces energy consumption and protects mice from obesity and hepatosteatosis[15].

Chronic changes in liver SAMe levels in lean mice have been associated with the onset and progression of NAFLD with age. Accumulation of liver SAMe due to its decreased catabolism in glycine N-methyltransferase (Gnmt)-knockout (KO) mice results in the spontaneous development of NAFLD and hepatocarcinogenesis with increasing age[16,17]. Curiously, chronic low liver SAMe levels in mice due to the absence of methionine adenosyltransferase 1a (Mat1a), which is mainly expressed in liver and encodes MATα1, that oligomerizes to generate MATI/III[6,18], also leads to the spontaneous onset of NASH with age[6,19]. Thus, maintaining physiological levels of liver SAMe in an appropriate range is essential to fine-tune liver function during aging.

Here, we investigated if the pharmacological (antisense oligonucleotide (ASO) treatment) knockdown of the Mat1a gene, which is involved in the first reaction of the methionine cycle, provides beneficial outcomes in diet- and genetically- induced obesity and obesity-related NAFLD. We designed two different ASO to knockdown Mat1a, and used diet-induced obese, ob/ob, and hepatocyte-specific fibroblast growth factor (Fgf)21-KO mice. Mat1a-KO mice were also used. Metabolic fluxes using radiolabeled substrates, very-low density lipoprotein (VLDL) secretion and dietary lipid metabolism were analyzed. The results showed that targeting Mat1a prevents and reverses obesity as well as obesity-associated dyslipidemia, insulin resistance and hepatosteatosis. Studies performed in vivo and in vitro showed that the hepatocyte secretion of FGF21 mediates increased thermogenesis of brown adipose tissue (BAT) and that FGF21 is regulated, in part, by nuclear factor erythroid 2-related factor 2 (NRF2).

## Results

**Targeting Mat1a reverses diet-induced obesity (DIO).** Several changes in methionine cycle are linked to protection against obesity in mice[11,12,15]. Therefore, we investigated if targeting Mat1a could improve and/or prevent obesity and the associated whole-body metabolic dysregulation. Mat1a knockdown (KD) was performed using two different Mat1a ASO (Mat1a ASO and Mat1a ASO2). The results showed that administration of both Mat1a ASO or Mat1a ASO2 did not induce changes in liver or renal damage markers in HFD-fed mice (Table 1 and Supplementary Table 1). Treatment with Mat1a ASO or Mat1a ASO2 led to a 90-100% downregulation of Mat1a in HFD-fed mice liver (Supplementary Fig. 1a, b). The results showed that feeding a HFD did not alter liver levels of MATI/III (Fig. 1a), or induce the transcription of Mat1a in BAT or white adipose tissue (WAT), in which protein levels were absent (Supplementary Fig. 1c). However, targeting Mat1a with ASO markedly reduced liver MATI/III levels (Fig. 1a) and induced a loss in body weight to that comparable with chow diet-fed (CD) mice (Fig. 1b, c and Supplementary Fig. 1d). Most body weight loss was due to the lower fat mass, although there was also a decrease in lean mass (Fig. 1d). Food intake (Fig. 1e) and rectal temperature (Supplementary Fig. 1e) were unchanged. The same profile was observed when HFD-fed mice were treated with Mat1a ASO2 (Supplementary Fig. 1f). Moreover, the HFD-induced impairment in glucose disposal, in the release of insulin in response to glucose (Fig. 1f), in fasting insulin levels (Supplementary Fig. 1g) and in insulin sensitivity (Fig. 1g) were prevented when HFD-fed mice were treated with the Mat1a ASO (Fig. 1f, g). Even more, the glucose tolerance test (GTT), the release of insulin and the insulin

---

**Table 1 Corporal and hepatic and renal function parameters in ASO-treated mice.**

| Parameter | Normal Range | CD Control ASO (n = 5) | HFD Control ASO (n = 5) | HFD Mat1a ASO (n = 5) |
|---|---|---|---|---|
| ALB (g/l) | 26–54 | 34.2 ± 0.9 | 28.3 ± 5.4# | 27.5 ± 1.8 |
| ALT (IU/l) | 22–133 | 17.2 ± 4.1 | 34.6 ± 25.2 | 31.4 ± 6.9 |
| AST (IU/l) | 46–221 | 36.0 ± 6.3 | 53.4 ± 32.3 | 61.0 ± 24.9 |
| TBIL (mg/dl) | 0.1–0.7 | 0.03 ± 0.01 | 0.04 ± 0.03 | 0.04 ± 0.02 |
| CRE (mg/dl) | 0.1–1.8 | 0.23 ± 0.03 | 0.19 ± 0.07 | 0.17 ± 0.01 |
| Urea (mg/dl) | 4.3–153.9 | 47.0 ± 2.9 | 48.6 ± 5.6 | 46.0 ± 9.6 |
| Body weight (g) | - | 29.1 ± 2.3 | 42.1 ± 3.3### | 32.8 ± 3.1** |
| Liver (g) | - | 1.43 ± 0.18 | 1.89 ± 0.45 | 1.15 ± 0.05** |
| Kidney (g) | - | 0.29 ± 0.03 | 0.34 ± 0.02# | 0.31 ± 0.05 |
| Spleen (g) | - | 0.09 ± 0.01 | 0.10 ± 0.01 | 0.08 ± 0.01* |

2-month-old C57BL/6J mice were fed a chow diet (CD) or a high-fat diet (HFD) for 10 weeks. During the last 4 weeks mice were treated with Mat1a antisense oligonucleotide (ASO) (25 mg/kg/week) (n = 5) or with control ASO (25 mg/kg/week) (n = 5) until sacrifice. Serum albumin (ALB), alanine aminotransferase (ALT), aspartate aminotransferase (AST), total bilirubin (TBIL), creatinine (CRE) and urea, and body, liver, kidney and spleen weight were measured. Values are presented as means ± SD. Statistically significant differences between groups are indicated by *p < 0.05, and **p < 0.01 when comparing Mat1a ASO vs. control ASO; and by #p < 0.05 and ###p < 0.001 when comparing CD vs. HFD (two-tailed Student's test). Source data are provided as a Source data file.

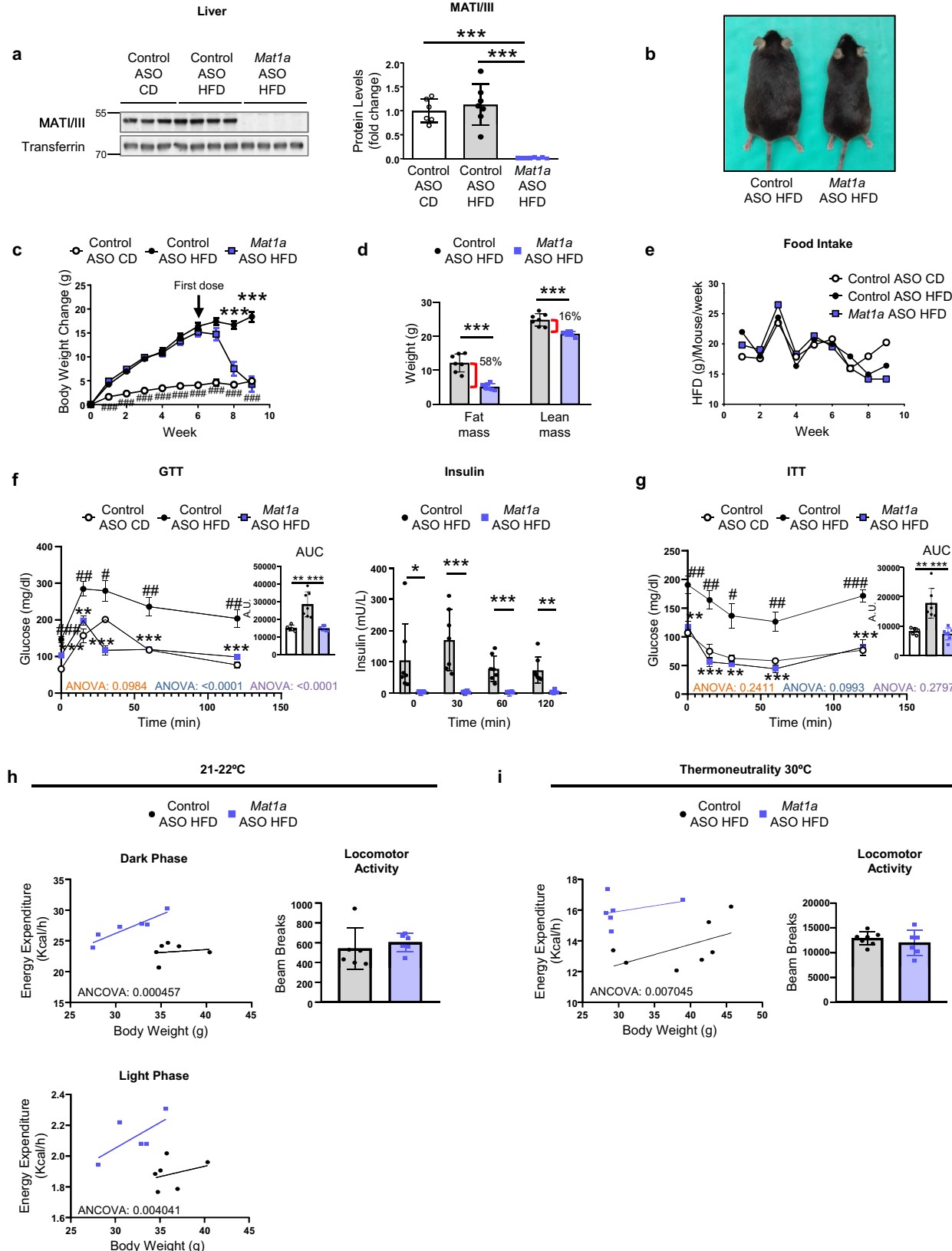

tolerance test (ITT) after the first dose of ASO, showed that targeting *Mat1a*, prevented the impairment in glucose disposal before the marked body weight loss (Supplementary Fig. 1h).

To investigate if the improved phenotype was a result of altered energy expenditure, metabolic studies were performed using an indirect calorimetric system. The analysis revealed that in HFD-fed mice, after the four injections of treatment, *Mat1a* ASO induced increased energy expenditure with lower body weight and the same locomotor activity (Fig. 1h) and respiratory quotient (Supplementary Fig. 1i) when compared with control ASO treatment. The same profile was observed during the light period, when animals were inactive (Fig. 1h) and in thermoneutrality (Fig. 1i), a specific

**Fig. 1 Targeting *Mat1a* reverses diet-induced obesity.** Two-month-old C57BL/6J mice were fed a chow-diet (CD) or a high-fat diet (HFD) for 10 weeks. During the last 4 weeks mice were treated with *Mat1a* antisense oligonucleotide (ASO) or control ASO (25 mg/kg/week) until sacrifice. **a** Knockdown of *Mat1a* ASO. MATI/III and transferrin blots and densitometries in liver of CD- ($n = 6$) and HFD-fed control ($n = 7$) and *Mat1a* ($n = 8$) ASO-treated mice. **b** Representative photograph of HFD-fed ASO-treated mice. **c** Body weight change for CD- ($n = 6$) and HFD-fed control ($n = 7$) and *Mat1a* ($n = 8$) ASO-treated mice. **d** Body composition in HFD-fed control ($n = 7$) and *Mat1a* ($n = 7$) ASO-treated mice. **e** Food intake for CD- ($n = 2$ cages) and HFD-fed control ($n = 2$ cages) and *Mat1a* ($n = 2$ cages) ASO-treated mice. **f** Glucose tolerance test (GTT), insulin serum levels and **g** insulin tolerance test (ITT) of CD- ($n = 5$) and HFD-fed control ($n = 7$) and *Mat1a* ($n = 7$) ASO-treated mice at the third week of treatment. Data are also indicated as area under the curve (AUC) expressed in arbitrary units (A.U.). **h** Dark-phase energy balance, locomotor activity, and light phase energy balance in HFD-fed control ($n = 6$ in light and dark phase) and *Mat1a* ($n = 5$ in light and $n = 6$ in dark phase) ASO-treated mice housed in metabolic cages over 2 days at the end of the treatment period. **i** Overall energy balance and locomotor activity in thermoneutrality in HFD-fed control ($n = 7$) and *Mat1a* ($n = 6$) ASO-treated mice housed in metabolic cages over 2 days at the end of the treatment period. Values are means ± SEM for time course representations, and means ± SD for histograms. Statistically significant differences are indicated by *$p < 0.05$, **$p < 0.01$, and ***$p < 0.001$ for Control ASO HFD vs. *Mat1a* ASO HFD; and #$p < 0.05$, ##$p < 0.01$, and ###$p < 0.001$ for Control ASO CD vs. Control ASO HFD (two-tailed Student's test). Statistical analysis performed by ANOVA test comparing Control ASO CD vs. Control ASO HFD; Control ASO HFD vs. *Mat1a* ASO HFD; Control ASO CD vs. *Mat1a* ASO HFD is presented in GTT and ITT curves. Statistical analysis for energy expenditure was performed by two-way ANCOVA test. Source data are provided as a Source data file.

measurement of non-shivering thermogenesis. The long-term treatment with the *Mat1a* ASOs showed that after 7 injections of *Mat1a* ASO, the body weight reached the CD-fed mice body weight, and maintained stable (during the following 2 injections), with no changes in food intake (Supplementary Fig. 1j). These results are in concordance with the fact that targeting *Mat1a* in CD-fed mice did not induce changes in body weight (Supplementary Table 2). The analysis of damage markers of liver and kidney function showed that the long-term treatment did not increase levels of these parameters above the normality range for mice. However, there was a slight increase in some parameters of liver function (Supplementary Table 3).

To assess if the chronic *Mat1a* deficiency could prevent the DIO and the associated insulin resistance, *Mat1a*-KO mice were fed a HFD for 10 weeks. *Mat1a*-KO mice were fully resistant to DIO, whereas the HFD-fed wild-type (WT) mice gained the expected weight (Supplementary Fig. 2a). Similarly, as occurred with mice treated with *Mat1a* ASOs, food intake maintained unaltered (Supplementary Fig. 2b) whereas the GTT and ITT showed that *Mat1a*-KO mice were protected from the HFD-induced glucose intolerance and insulin resistance (Supplementary Fig. 2c, d).

***Mat1a* ASOs induce thermogenesis in BAT.** The results showed that targeting liver *Mat1a* increased energy expenditure even in thermoneutrality, specific of non-shivering thermogenesis. Therefore, we performed a detailed study in adipose tissue, and observed that *Mat1a* ASO reduced the HFD-induced BAT (Fig. 2a) and WAT adipocyte size (Supplementary Fig. 3a). HFD-fed *Mat1a* ASO (Fig. 2b), *Mat1a* ASO2 (Supplementary Fig. 4a), and *Mat1a* KO mice (Supplementary Fig. 4a), all exhibited an increase in fatty acid oxidation (FAO) in BAT when compared to the corresponding controls. Targeting *Mat1a* also enhanced the oxygen consumption rate (OCR) in BAT mitochondria (Fig. 2c), which was associated with increased release of glycerol when lipolysis was measured ex vivo from freshly isolated BAT (Fig. 2d). However, release of fatty acids maintained unchanged probably because they are being used as substrate for the increased FAO in BAT (Fig. 2d). Inhibition of liver *Mat1a* led to increased interscapular temperature (Fig. 2e), increased uncoupling protein1 (UCP1) levels and S6 signaling, while no effects were observed in the mitogens activated protein kinase p38 or in protein kinase A (PKA) phosphorylation (Fig. 2f).

Similar to BAT, in WAT, the acid-soluble metabolites (ASM) release during FAO (Supplementary Fig. 3b) and lipolysis were elevated in *Mat1a* ASO-treated mice (Supplementary Fig. 3c). The increased lipolysis, coupled a decreased serum fatty acid levels (Supplementary Fig. 3d) and a tendency to increase UCP1

and pS6 signaling while no changes were observed in p38 or its phosphorylated form (Supplementary Fig. 3e).

Treatment with *Mat1a* ASO did not induce increases in serum epinephrine or norepinephrine levels (Fig. 2g) nor in BAT epinephrine levels (Supplementary Table 4). The increased expression of *Ucp1* induced by *Mat1a* ASO in BAT was associated with increased peroxisome proliferator-activated receptors *(Ppar) alpha (a)*, *Ppar gamma (g)* and adiponectin Q *(Adipoq)* expression (Fig. 2h). In WAT, *Mat1a* ASO induced the increase in *Adipoq* expression while the expression of *Pparg* remained unaltered (Supplementary Fig. 3f). These changes were associated with increased serum adiponectin levels (Supplementary Fig. 3g).

Supporting the increased thermogenesis in BAT, the acute administration of a β3 adrenergic agonist to *Mat1a* ASO-treated mice showed a higher consumption of $VO_2$ 30 min and, more markedly, 45 min after the injection (Supplementary Fig 4b), and an increased response in the induction of the expression of *Ucp1* and *Ppargc1a*, both involved in thermogenesis (Supplementary Fig. 4c). This indicates an increased sympathetic response, which is essential for BAT non-shivering thermogenesis. Finally, to investigate if the effect in BAT was cell-autonomous or was more linked to a peripheral effect, FAO and the expression of *Pparg* (in differentiation) and *Ucp1* (after the induction) were measured in BAT primary cells (Supplementary Fig. 4d, e). The results showed that BAT primary cells maintained in culture did not show the increased FAO or *Ucp1* and *Pparg* expression induced in BAT by the in vivo treatment with *Mat1a* ASO, suggesting that these effects are not cell-autonomous of BAT adipocytes.

Finally, the results showed that targeting liver *Mat1a* down-regulated *Acaca* expression (Supplementary Fig. 5a) and acetyl-CoA carboxylase (ACC) and fatty acid synthase (FAS) protein levels in BAT (Supplementary Fig. 5b), involved in the de novo lipogenesis, suggesting a contribution to the decreased lipid content in BAT.

Altogether, the results here demonstrate that targeting liver *Mat1a*, induces BAT thermogenesis, WAT lipolysis and the secretion of adiponectin. The results suggest that activation of S6 signaling might underline the induction of UCP1 in BAT, as has been demonstrated before[20].

***Mat1a* ASOs protect from high-fat diet-induced NAFLD.** The most effective treatments to reverse NAFLD are those that lead to weight loss[21]. Thus, we further investigated if the *Mat1a* ASO could also reduce the HFD-induced hepatosteatosis. The results showed that the HFD-induced liver storage of lipid droplets and increased triglyceride (TG) concentration (Fig. 3a) after 10 weeks of a HFD, were prevented when HFD-fed mice were treated with

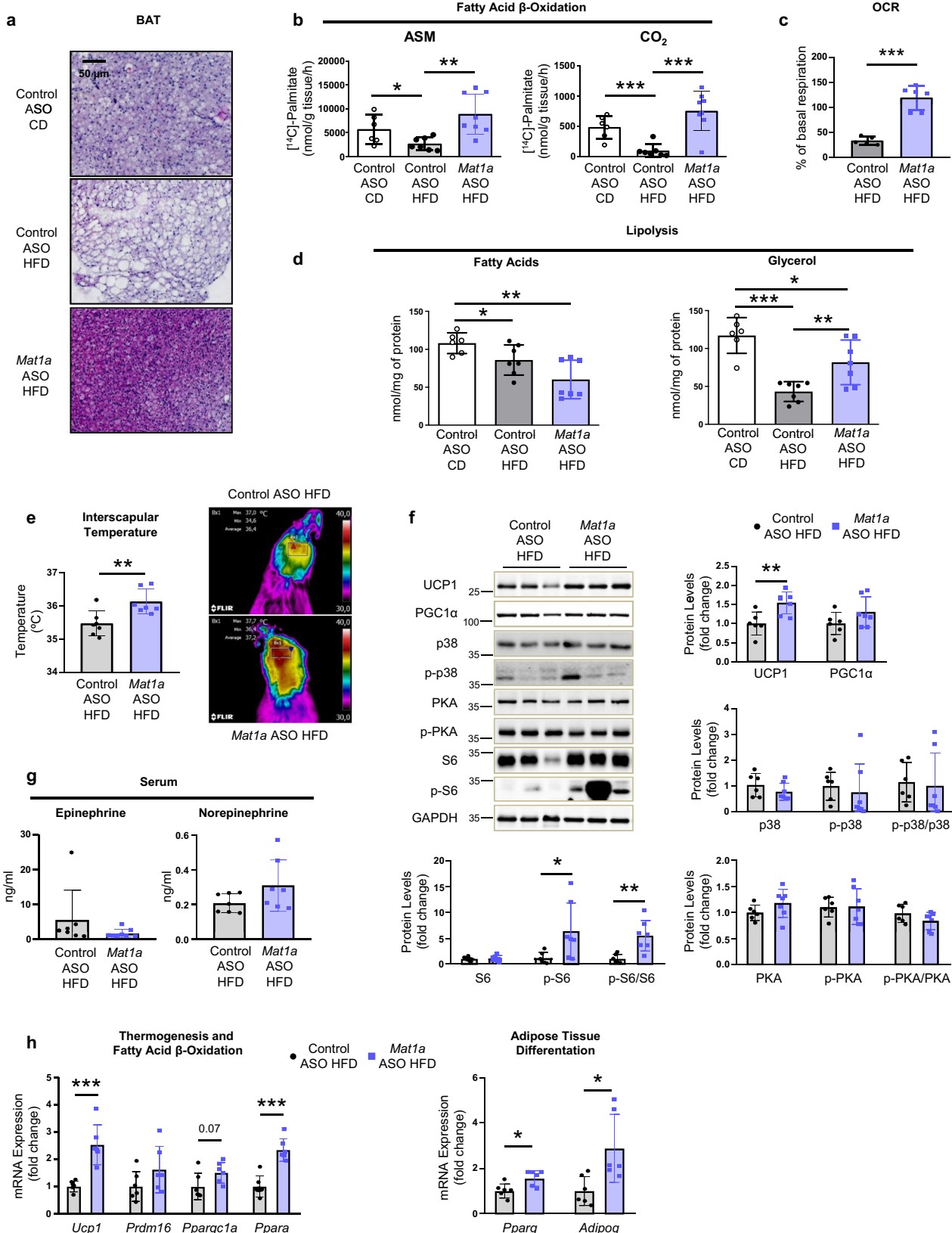

*Mat1a* ASO from the sixth week (one IP dose per week) (Fig. 3a). Furthermore, although dispersion of the data was high, the Sirius Red and F4/80 immunostaining (Fig. 3a) and the expression levels of genes involved in fibrosis and inflammation (Supplementary Fig. 6a), showed that treatment with *Mat1a* ASO did not induce the increase of liver fibrosis or inflammation. To validate the antisteatotic effect of *Mat1a* knockdown, another cohort of HFD-fed mice were treated with *Mat1a* ASO2, and *Mat1a*-KO mice were fed a HFD. Similar results were obtained: both *Mat1a* ASO2 treated (Supplementary Fig. 6b, c) and *Mat1a*-KO (Supplementary Fig. 6d) mice did not develop hepatosteatosis as evident from the absence of liver lipid droplets and reduced

**Fig. 2 *Mat1a* antisense oligonucleotides induce thermogenesis in brown adipose tissue (BAT).** Two-month-old C57BL/6J mice were fed a chow diet (CD) or a high-fat diet (HFD) for 10 weeks. During the last 4 weeks mice were treated with *Mat1a* antisense oligonucleotide (ASO) or control ASO (25 mg/ kg/week) until sacrifice. **a** Representative microphotographs of BAT sections stained with Hematoxylin/Eosin of CD- ($n = 6$) and HFD-fed ASO-treated mice ($n = 6$). **b** Fatty acid β-oxidation was determined measuring the amount of [$^{14}$C]-acid-soluble metabolites (ASM) (incomplete oxidation of palmitate) and [$^{14}$C]-$CO_2$ (complete oxidation of palmitate) in BAT of CD- ($n = 6$) and HFD-fed control ($n = 7$) and *Mat1a* ($n = 8$) ASO-treated mice. **c** Oxygen consumption rate (OCR) expressed as % of basal respiration of HFD-fed control ($n = 5$) and *Mat1a* ($n = 6$) ASO-treated mice. **d** BAT lipolysis was determined by measuring the amount of fatty acids and glycerol secreted ex vivo by BAT of CD- ($n = 6$) and HFD-fed control ($n = 7$) and *Mat1a* ($n = 7$) ASO-treated mice. **e** Quantification and infrared thermal images of BAT interscapular temperature of HFD-fed control ($n = 7$) and *Mat1a* ($n = 7$) ASO-treated mice. **f** Representative blots and densitometries of uncoupling protein1 (UCP1), peroxisome proliferator-γ activated receptor (PPAR)-γ co-activator 1α (PGC1α), phosphorylated and total forms of mitogen-activated protein kinase p38 (p38), protein kinase A (PKA) and S6 protein, and glyceraldehyde-3-phosphate dehydrogenase (GAPDH), as representative loading control, in BAT of HFD-fed control ($n = 6$) and *Mat1a* ($n = 7$) ASO-treated mice. The ratio between phosphorylated and total forms of the proteins was determined. **g** Serum epinephrine and norepinephrine levels in HFD-fed control ($n = 7$) and *Mat1a* ($n = 7$) ASO-treated mice. **h** mRNA expression levels in BAT of HFD-fed control ($n = 6$) and *Mat1a* ($n = 6$) ASO-treated mice of *Ucp1*, PR/SET Domain 16 (*Prdm16*), PGC1α (*Ppargc1a*) and PPAR alpha (*Ppara*), as indicators of thermogenesis and mitochondrial fatty acid β-oxidation), and PPAR gamma (*Pparg*) and adiponectin (*Adipoq*) genes, as indicators of adipocyte differentiation. Results were normalized with *Gapdh* and Actin (*Actb*). Values are presented as means ± SD. Statistically significant differences between groups are indicated by *$p < 0.05$, **$p < 0.01$, and ***$p < 0.001$ (two-tailed Student's test). Source data are provided as a Source data file.

TG levels (Supplementary Fig. 6b, d). In addition, targeting liver *Mat1a* also induced a decrease in muscle TG levels (Supplementary Fig. 7a).

One of the most important metabolic processes regulating liver lipid content is the mitochondrial FAO. Our results showed that targeting *Mat1a* in HFD-fed mice did not induce changes in FAO when compared with controls, as demonstrated by levels of [$^{14}$C]-Palmitate oxidation and serum ketone bodies concentration (Fig. 3b). In contrast, de novo lipogenesis fluxes, assessed in liver pieces, showed a marked decrease (Fig. 3c), mainly related to the reduction in levels of ACC and FAS proteins, key enzymes involved in de novo fatty acid synthesis (Fig. 3d). Thus, the results here demonstrated that targeting *Mat1a* improves the obesity-related hepatosteatosis (Fig. 3).

We next analyzed if this antisteatotic profile was also linked to reduced levels of serum TG. The results showed that during feeding and fasting, *Mat1a* ASO treatment reduced serum TG levels when compared to control ASO-treated mice (Fig. 4a). In vivo analysis of TG-rich lipoprotein metabolism showed that hepatic TG secretion rate maintained unaltered in HFD-fed *Mat1a* ASO-treated mice when compared to the control ASO-treated mice (Fig. 4b) whereas the chylomicron TG clearance was increased (Fig. 4c). To ascertain the fate of the dietary lipids when *Mat1a* was knocked-down in vivo, a bolus of olive oil with [$^{3}$H]-Triolein was administrated. The results showed that when HFD-fed mice were treated with *Mat1a* ASO, the uptake of dietary lipids was mainly increased in the BAT (Fig. 4d). This increase was not linked with higher BAT levels of lipoprotein lipase (LPL), involved in TG-rich lipoprotein catabolism, or of CD36, a fatty acid transporter (Fig. 4e). However, serum levels (Fig. 4f) and liver expression (Fig. 4g) of the LPL activator *ApoC2*, were increased, while those of the LPL inhibitor, *ApoC3* were decreased (Fig. 4f, g). These results suggest that when targeting liver *Mat1a*, serum TGs are transported towards the BAT, which will also contribute to a decreased accumulation in liver.

*Mat1a* ASO reduces DIO by increasing energy expenditure and increases BAT TG uptake, suggesting a higher utilization of lipids in BAT to cope with energy requirements for its thermogenic action, which could also contribute to the decreased hepatic lipid content in these mice.

**Mat1a ASOs reverse obesity and hepatosteatosis in ob/ob mice.** To demonstrate the improvement in obesity, insulin resistance and hepatosteatosis induced by *Mat1a* ASO in genetically obese mice, leptin deficient *ob/ob* mice, a well-established model of obesity[22] were used. Changes in body weight were assessed during the ASO

treatment; targeting hepatic *Mat1a* prevented the weight gain after the second dose and induced weight loss by the end of the treatment (Fig. 5a). As previously observed in DIO mice, the food intake maintained unchanged (Fig. 5b). Notably, *Mat1a* ASO improved GTT (Fig. 5c), ITT (Fig. 5d), fasting insulin levels (Fig. 5e) and induced a marked resolution of hepatosteatosis with a reduced accumulation of lipid droplets and liver TG concentration (Fig. 5f). The results further showed that targeting *Mat1a* ASO in *ob/ob* mice led to increased FAO in BAT (Fig. 5g), increased protein levels and expression of UCP1 in BAT (Fig. 5h, i) and increased expression of *Ppara*, *Pparg* and *Adipoq* in BAT (Fig. 5i); thus, confirming the role of *Mat1a*-induced BAT thermogenesis in the improved phenotype.

**Mat1a ASOs prevent obesity through induction of hepatocyte secretion of FGF21.** *Mat1a* ASO prevents and reverses obesity by inducing BAT thermogenesis. The products of *Mat1a* gene (MATI/III) catalyze the first reaction of the methionine cycle in liver[6], which is the conversion of methionine into SAMe. Thus, the loss of *Mat1a* results in the reduction of methionine utilization and a concomitant accumulation of liver methionine and the decrease in SAMe levels[6]. In concordance, the results here showed that liver methionine was increased 610% in *Mat1a* ASO-treated HFD-fed mice liver and 2350% in HFD-fed *Mat1a*-KO mice liver when compared to the corresponding WT mice (Supplementary Fig. 8a, b). This increase was linked with a decrease in liver SAMe of 20% in *Mat1a* ASO-treated mice and of 30% in *Mat1a*-KO mice when compared to the corresponding controls (Supplementary Fig. 8a, b). Considering that methionine deficiency induces thermogenesis through the secretion of liver FGF21[23], to asses if the decreased utilization led also to the secretion of FGF21, levels of FGF21 were also measured in serum of the generated models. Serum levels of FGF21 in HFD-fed *Mat1a* ASO and *Mat1a* ASO2 treated mice, in *Mat1a* ASO-treated *ob/ob* and *Mat1a*-KO mice fed a HFD were increased (Fig. 6a). In vitro experiments showed that hepatocytes isolated from *Mat1a* ASO-treated HFD-fed mice secreted more FGF21 into the media than hepatocytes isolated from control ASO-treated mice (Fig. 6b). To investigate if hepatocyte FGF21 was directly involved in the improvement of obesity, AlbCre-*Fgf21* and their control mice (AlbCre) were fed a HFD and were treated with *Mat1a* or control ASO from the sixth week of the HFD. Targeting *Mat1a* in HFD-fed AlbCre mice resulted in body weight loss (Fig. 6c and Supplementary Fig. 8c) without inducing changes in food intake (Supplementary Fig. 8d) whereas it did not affect body weight or food intake when *Fgf21* was absent in the liver (Fig. 6c and

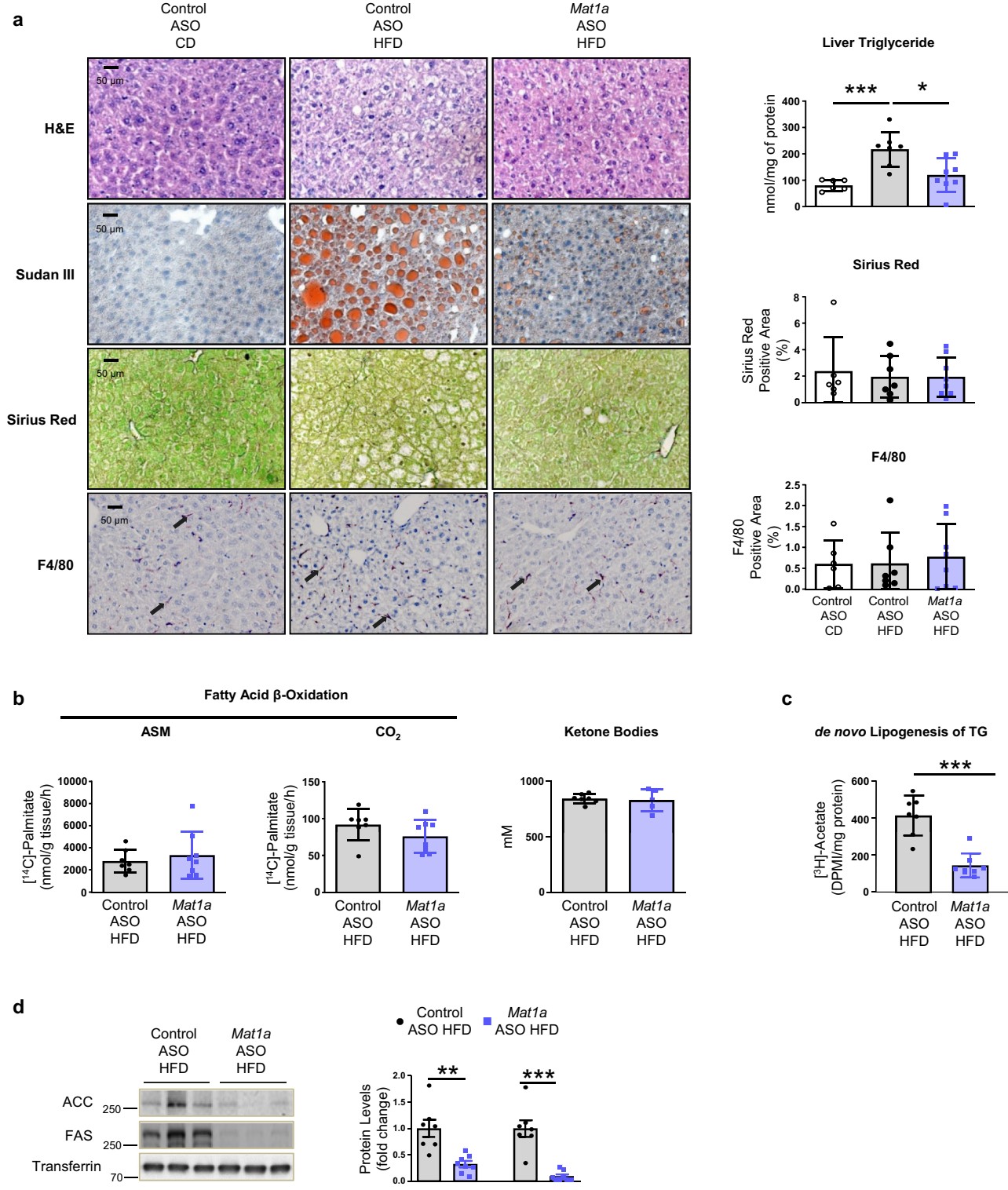

Supplementary Fig. 8d). As expected, *Mat1a* ASO treatment increased FGF21 in serum of HFD-fed AlbCre mice while it did not affect FGF21 levels in HFD-fed AlbCre-*Fgf21* mice, in which serum FGF21 levels were hardly detectable (Fig. 6d).

The increased energy expenditure (Fig. 6e), and BAT complete FAO (Fig. 6f) in *Mat1a* ASO-treated HFD-fed AlbCre mice was not observed when *Fgf21* was absent in hepatocytes (Fig. 6e, f). The results also showed that the increased UCP1 levels and S6 signaling (Fig. 6g) induced by *Mat1a* ASO was not found

when *Fgf21* was knocked out (Fig. 6g). Targeting *Mat1a* did not affect serum epinephrine nor norepinephrine levels in hepatocyte-specific *Fgf21*-KO mice (Supplementary Fig. 8e). However, the altered *ApoC*s expression observed in the liver when targeting *Mat1a*, was no longer observed when *Fgf21* was absent (Supplementary Fig. 8f).

To ascertain if the receptor of FGF21, the β-klotho, and UCP1 were involved in the increased thermogenesis, *Ucp1* and *β-Klotho* were silenced in BAT (Supplementary Fig. 9a).

**Fig. 3 *Mat1a* antisense oligonucleotides protect from high-fat diet (HFD)-induced hepatosteatosis.** Two-month-old C57BL/6 J mice were fed a chow diet (CD) or a high-fat diet (HFD) for 10 weeks. During the last 4 weeks mice were treated with *Mat1a* antisense oligonucleotide (ASO) or with control ASO (25 mg/kg/week) until sacrifice. **a** Representative microphotographs of liver sections stained with Hematoxylin/Eosin (H&E), Sudan III, Sirius Red and F4/80, and liver triglyceride (TG) content, and quantification of Sirius red and F4/80 in CD- ($n = 6$) and HFD-fed control ($n = 7$) and *Mat1a* ($n = 8$) ASO-treated mice. **b** Liver fatty acid β-oxidation was determined measuring the amount of [$^{14}$C]-acid-soluble metabolites (ASM) (incomplete oxidation of palmitate), [$^{14}$C]-$CO_2$ (complete oxidation of palmitate) and serum ketone bodies levels in HFD-fed control ($n = 7$ for ketone bodies and $n = 7$ for β-oxidation) and *Mat1a* ($n = 5$ for ketone bodies and $n = 8$ for β-oxidation) ASO-treated mice. **c** Liver TG de novo lipogenesis determined by incorporation of [$^{3}$H]-acetate into TG in HFD-fed control ($n = 7$) and *Mat1a* ($n = 8$) ASO-treated mice liver pieces. **d** Representative blots and densitometries of acetyl-coenzyme A carboxylase (ACC), fatty acid synthase (FAS) and transferrin, as representative loading control, in liver of HFD-fed control ($n = 7$) and *Mat1a* ($n = 8$) ASO-treated mice. Values are presented as means ± SD. Statistically significant differences between groups are indicated by *$p < 0.05$, **$p < 0.01$, and ***$p < 0.001$ (two-tailed Student's test). Source data are provided as a Source data file.

The results showed that the body weight loss (Supplementary Fig. 9b, c) and the increased energy expenditure (Supplementary Fig. 9d) induced by *Mat1a* ASO, were not evident when *Ucp1* or *β-Klotho* were silenced in BAT. To note, all these effects were independent of changes in locomotor activity (Supplementary Fig. 9e).

Finally, compared with control ASO treatment, HFD-fed AlbCre mice also exhibited improved GTT when given *Mat1a* ASO (Fig. 6h); HFD-fed AlbCre-*Fgf21* mice on the other hand did not show differences in GTT between ASO treatments (Fig. 6h).

These data collectively show that targeting *Mat1a* activates the liver-BAT axis through the induction of hepatocyte FGF21 secretion.

**NRF2 mediates hepatocyte FGF21 secretion.** The most studied regulator of FGF21 has been the PPARα[24]; however, as chronic deficiency in *Mat1a* leads to endoplasmic reticulum (ER) stress[19] and ER stress may also regulate generation of FGF21 through activation of activating transcription factor 4 (ATF4)[25], we hypothesized that this might be an important regulator of FGF21 in this context. In addition, the transcription factor NRF2, an ATF4-interacting factor, is a master regulator of the cellular defense system against oxidative stress and has been shown to modulate FGF21 expression in diabetic mice[26]. Thus, here PPARα, ATF4 and NRF2 were evaluated as potential regulators of FGF21 in our models. Experiments performed in hepatocytes isolated from HFD-fed mice treated with ASOs, in which *Nrf2*, *Atf4* or *Ppara* were silenced through the usage of siRNAs (Supplementary Fig. 10a) showed that the knockdown of *Nrf2* was able to reduce the increased FGF21 secretion induced by *Mat1a* ASO in HFD-fed mice hepatocytes (Supplementary Fig. 10b). However, *Atf4* or *Ppara* knockdown did not reduce FGF21 secretion (Supplementary Fig. 10b). Targeting *Mat1a* did not induce changes in the levels of nuclear PPARα or ATF4 (Supplementary Fig. 10c) and when hepatocytes were treated with inhibitors of PPARα (Supplementary Fig. 10d) or of ER stress (Supplementary Fig. 10e), neither was able to recover normal levels of FGF21 secretion (Supplementary Fig. 10d, e).

Here, we found that when isolated hepatocytes from ASO-treated HFD-fed mice were exposed to ML385, a specific inhibitor of NRF2 activity, FGF21 secretion decreased (Fig. 7a), in concordance with the decreased secretion induced by si*Nrf2* (Supplementary Fig. 10b). Moreover, when HFD-fed mice were treated with *Mat1a* ASO, liver NRF2 levels increased in nucleus (Fig. 7b), while levels in liver remained unaltered and those in liver cytoplasm decreased (Fig. 7b), and NRF2-target genes were upregulated (Fig. 7c), thus confirming that targeting *Mat1a* increases NRF2 activity. Even more, in hepatocytes from HFD-fed mice treated with *Mat1a* ASO, NRF2 was found directly binded to a promoter site in the *Fgf21* gene (Fig. 7d). These results indicate that NRF2 plays a key role in mediating *Mat1a* ASO-induced hepatocyte FGF21 secretion. Next, we investigated if the protection from hepatosteatosis induced by targeting *Mat1a*

was directly mediated by FGF21 or NRF2. The results showed that in the HFD-fed AlbCre-*Fgf21* mice (where *Fgf21* was absent in hepatocytes), NRF2 levels were preserved increased in nucleus (Fig. 7e), confirming that NRF2 is upstream FGF21. However, concentrations of liver TGs were enhanced when compared with HFD mice treated with the control ASO (Fig. 7f). These data imply that while *Mat1a* ASO increases NRF2 and this transcription factor is upstream FGF21, the deletion of FGF21 is sufficient to prevent the actions of *Mat1a* ASO treatment. In addition, the results showed that targeting liver *Mat1a* also induced the expression of *Nrf2* and, more markedly, *Fgf21* in BAT (Supplementary Fig. 11a), and that this increased expression was absent when liver *Fgf21* was absent (Supplementary Fig. 11b) or when BAT primary cells were maintained in culture (Supplementary Fig. 11c, d). Thus, the *Mat1a* ASO induced secretion of FGF21 by hepatocytes also regulates the expression of *Fgf21* in BAT.

*Mat1a*-KO hepatocytes, with low hepatic SAMe levels, have higher mitochondrial reactive oxygen species (ROS), which are normalized when *Mat1a* is overexpressed[27]. SAMe is required in the transsulfuration pathway (Fig. 7g). In this sense, we found that levels of glutathione (GSH) were decreased in *Mat1a* ASO-treated HFD-fed mice as compared to the ASO control treated mice while glutathione disulfide (GSSG) levels maintained unaltered (Fig. 7g) and levels of malondialdehyde (MDA), a measure of lipid peroxidation, were increased (Fig. 7g). Treatment of hepatocytes with GSH ether or the antioxidant N-acetylcysteine (NAC), decreased the *Mat1a* ASO-induced secretion of FGF21 (Fig. 7h), the same effect was obtained when hepatocytes were treated with SAMe (Fig. 7i), which effects inside the hepatocytes were verified by the increased expression of genes involved in the methionine cycle (Supplementary Fig. 12a) and the levels of small ubiquitin-related modifier 1 (SUMO1)-conjugated proteins and SUMO1 (Supplementary Fig. 12b). Thus, the results show that SAMe deficiency is involved in the increased FGF21 secretion by hepatocytes.

Given that methionine accumulates in hepatocytes due to *Mat1a* ASO treatments, we further incubated hepatocytes in a methionine-deficient media. The results showed that whilst the methionine-deficient media increased FGF21 secretion in control ASO-treated hepatocytes (Supplementary Fig. 12c); it did not affect secretion of FGF21 in *Mat1a* ASO-treated cells (Supplementary Fig. 12c).

Altogether, these data show that treatment with *Mat1a* ASO increases ROS despite regression of hepatosteatosis (and without fibrosis or inflammation).

**Discussion**

Obesity is a major public health issue with increasing prevalence worldwide. It is a well-established risk factor for a variety of diseases, including NAFLD[28], type 2 diabetes[29], dyslipemia[30], among others. Weight loss through pharmacological treatments,

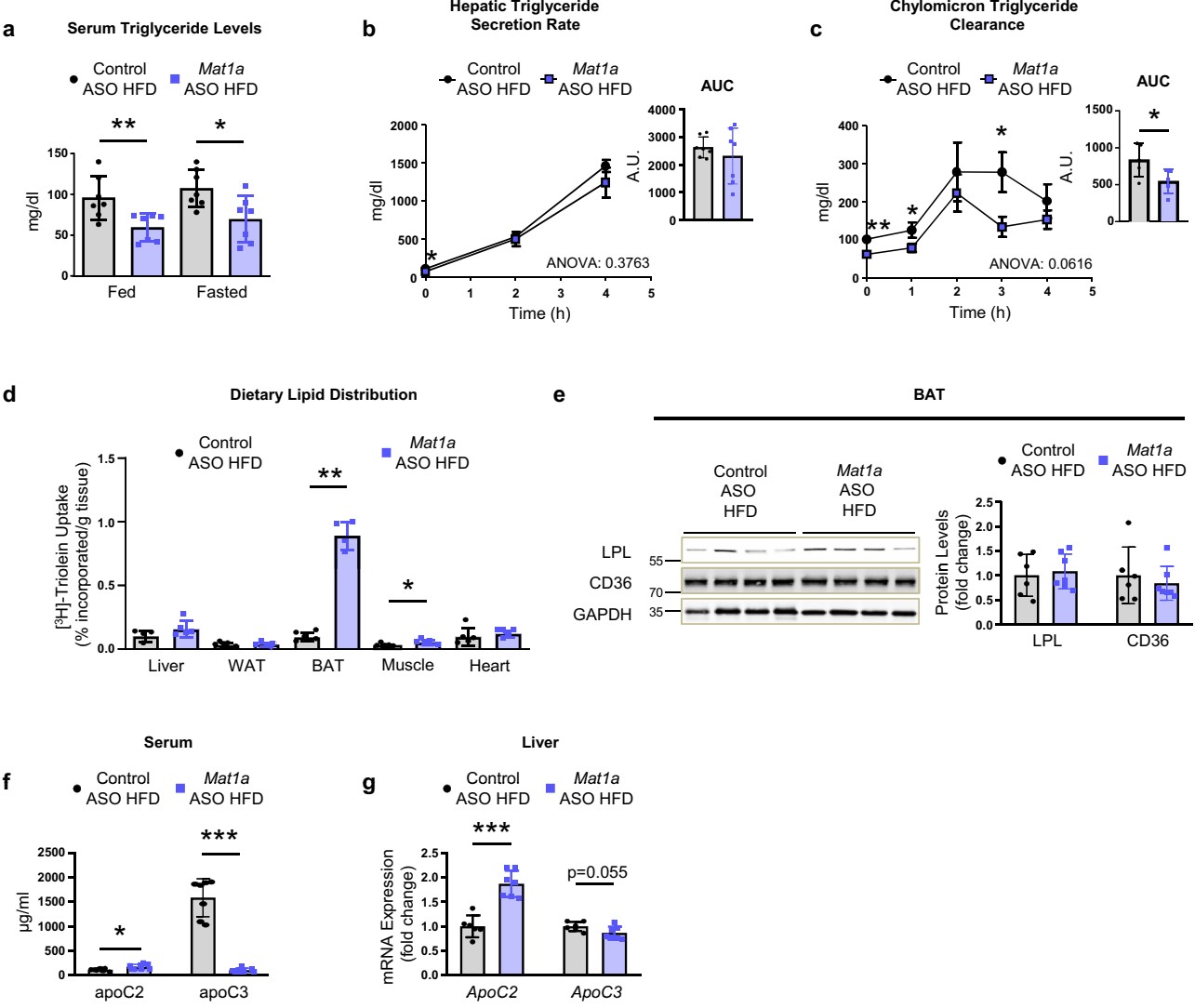

**Fig. 4 Mat1a antisense oligonucleotides channel plasma lipids towards the brown adipose tissue (BAT).** Two-month-old C57BL/6J mice were fed a high-fat diet (HFD) for 10 weeks. During the last 4 weeks mice were treated with *Mat1a* antisense oligonucleotide (ASO) or with control ASO (25 mg/kg/week) until sacrifice. **a** Serum triglycerides (TG) levels in fed and fasted conditions in HFD-fed control ($n = 7$) and *Mat1a* ($n = 7$) ASO-treated mice. **b** Circulating TG levels from mice fasted for 4 h prior (0 h), 2 h and 4 h after treatment with the LPL inhibitor, poloxamer P-407 in HFD-fed control ($n = 7$) and *Mat1a* ($n = 7$) ASO-treated mice. Data are also indicated as area under the curve (AUC) expressed in arbitrary units (A.U.). **c** TG serum levels during oral lipid tolerance test after overnight fasting in HFD-fed control ($n = 5$) and *Mat1a* ($n = 5$) ASO-treated mice. Data are also indicated as AUC expressed in A.U. **d** Tissue distribution of [³H]-triolein 4 h after oral gavage in HFD-fed control ($n = 5$) and *Mat1a* ($n = 5$) ASO-treated mice. **e** Representative blots and densitometries of lipoprotein lipase (LPL), fatty acid translocase (CD36) and glyceraldehyde-3-phosphate dehydrogenase (GAPDH) in BAT of HFD-fed control ($n = 6$) and *Mat1a* ($n = 7$) ASO-treated mice. **f** Serum apoC2 and apoC3 levels in HFD-fed control ($n = 7$) and *Mat1a* ($n = 8$) ASO-treated mice. **g** mRNA liver expression of *ApoC2* and *ApoC3* levels in HFD-fed control ($n = 6$) and *Mat1a* ($n = 7$) ASO-treated mice. Results were normalized with *Gapdh*. Values are presented as means ± SEM for time course representations, and as means ± SD for histograms. Statistically significant differences between groups are indicated by \*$p < 0,05$, \*\*$p < 0.01$, and \*\*\*$p < 0.001$ (two-tailed Student's test). Statistical analysis performed by two-way ANOVA test comparing Control ASO HFD vs. *Mat1a* ASO HFD is presented in hepatic TG secretion and chylomicron clearance curves. Source data are provided as a Source data file.

lifestyle interventions and/or bariatric surgery are also effective to treat the obesity-related co-morbidities[3]. However, there is still a need to better understand the connecting relationships between different metabolic disorders related to obesity so that new therapies might be proposed.

We demonstrate that ASO-mediated silencing of *Mat1a* reverses and prevents obesity, insulin resistance and the associated hepatosteatosis as well as reducing lipids in serum. *Mat1a* gene is mainly expressed in the liver (primarily in hepatocytes) and encodes the methionine adenosyltransferase subunit alpha 1 (MATα1). The oligomerization of MATα1 leads to the formation

of the enzyme MATI/III, which is the first enzyme of the methionine cycle[6,18]. MATI/III catalyzes the conversion of methionine to SAMe, a major methyl donor in the cell[18]. Altered methionine metabolism and SAMe synthesis have been associated with adiposity and weight gain in individuals with overweight and obesity[31].

These results showing that inhibition of *Mat1a* alleviates obesity and obesity-associated metabolic diseases are consistent with previous reports, which showed that disruption of other enzymes such as PEMT[12] or NNMT[15] involved in the methionine cycle leads to resistance to obesity. Some metabolic beneficial

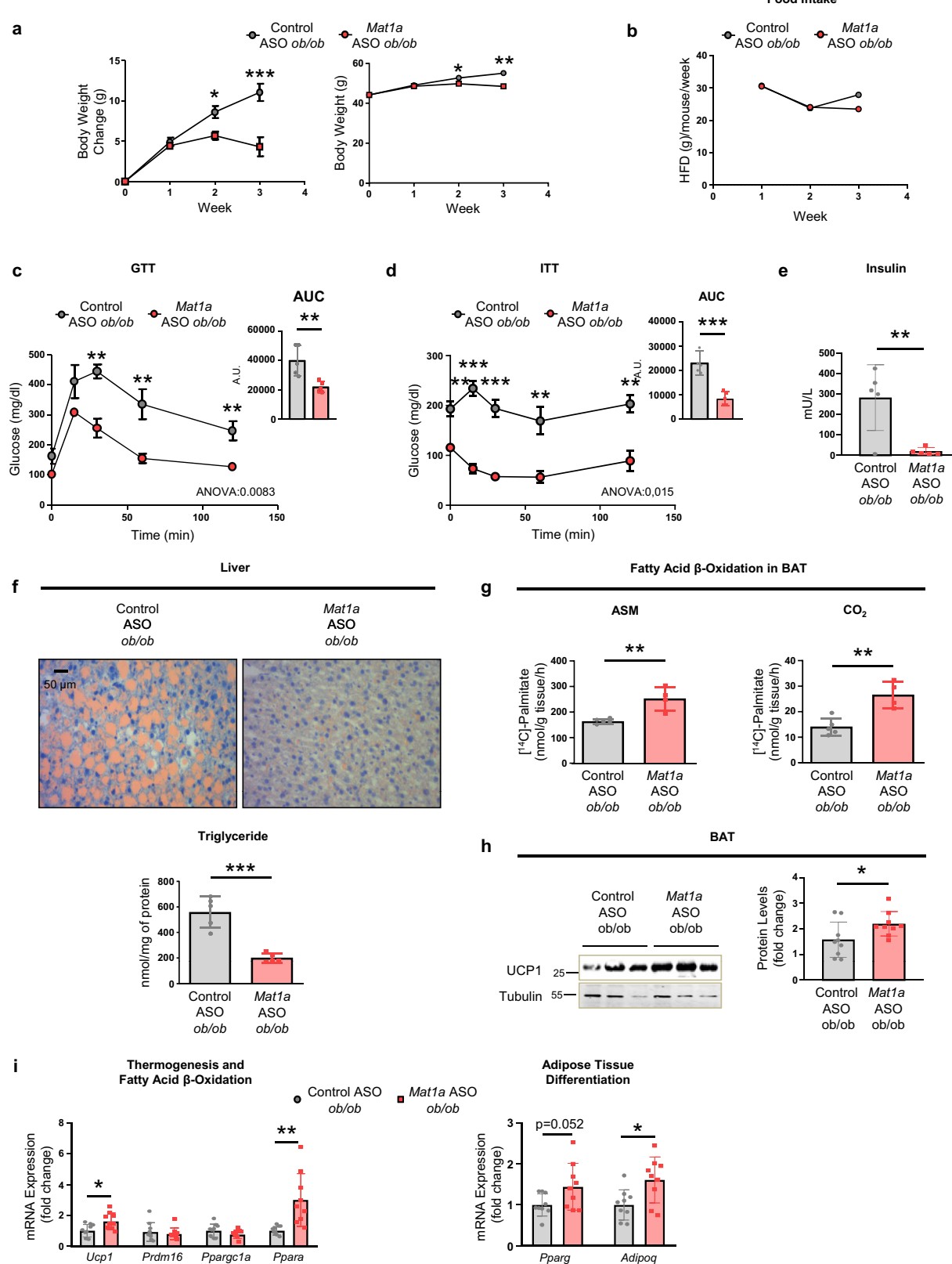

effects were also obtained in the absence of enzymes such as cystathionine-β-synthase or cystathionine gamma-lyase[31], involved in the transsulfuration pathway, which is strongly supported by the flux of methionine.

In the context of obesity, the beneficial effects driven by *Mat1a* deficiency closely resembles those produced by dietary MR. In

MR diets, SAMe synthesis is altered as when there is a down-regulation in *Mat1a*. MR leads to weight loss, improvement of insulin resistance and adiposity[32,33], reduction of hepatosteatosis and dyslipemia[33], and browning of adipose tissue activating its FAO and lipolysis[34]. The main mechanism by which MR induces its positive effects is the hepatic secretion of FGF21[23]. Our results

**Fig. 5 Mat1a antisense oligonucleotides reverse obesity and hepatoesteatosis in ob/ob mice.** Three-month-old B6.Cg-Lep[ob]/J (ob/ob) mice were fed a high-fat diet (HFD) for 4 weeks. During the diet mice were treated with Mat1a antisense oligonucleotide (ASO) or control ASO (25 mg/kg/week) until sacrifice. **a** Body weight change and body weight for HFD-fed control (n = 5) and Mat1a (n = 5) ASO-treated ob/ob mice. **b** Food intake for HFD-fed control (n = 2 cages) and Mat1a (n = 2 cages) ASO-treated ob/ob mice. **c** Glucose (GTT) and **d** insulin tolerance tests (ITT) in HFD-fed control (n = 5) and Mat1a (n = 5) ASO-treated ob/ob mice. Data are also indicated as area under the curve (AUC) expressed in arbitrary units (A.U.). **e** Serum insulin levels in HFD-fed control (n = 5) and Mat1a (n = 5) ASO-treated ob/ob mice fasted overnight. **f** Representative microphotographs of liver sections stained for Sudan III and liver triglyceride (TG) concentration of HFD-fed control (n = 5) and Mat1a (n = 5) ASO-treated ob/ob mice. **g** BAT fatty acid β-oxidation, determined measuring the amount of [$^{14}$C]-acid-soluble metabolites (ASM) (incomplete oxidation of palmitate) and [$^{14}$C]-$CO_2$ (complete oxidation of palmitate) in HFD-fed control (n = 5) and Mat1a (n = 4) ASO-treated ob/ob mice. **h** Representative blots and densitometries of uncoupling protein1 (UCP1) and tubulin in BAT of HFD-fed control (n = 9) and Mat1a (n = 9) ASO-treated ob/ob mice. **i** mRNA expression levels in BAT of HFD-fed control (n = 9) and Mat1a (n = 9) ASO-treated ob/ob of mice Ucp1, PR/SET Domain 16 (Prdm16), peroxisome proliferator-activated receptor (PPAR) gamma co-activator 1 (Ppargc1a) and PPAR alpha (Ppara) as indicators of thermogenesis and fatty acid β-oxidation; and PPAR gamma (Pparg) and adiponectin (Adipoq), as indicators of adipocyte differentiation. Results were normalized with Actin (Actb). Values are presented as means ± SEM for time course representations, and as means ± SD for histograms. Statistically significant differences between groups are indicated by *p < 0.05, **p < 0.01, and ***p < 0.001 (two-tailed Student's test). Statistical analysis performed by two-way ANOVA test comparing Control ASO HFD vs. Mat1a ASO HFD is presented in GTT and ITT curves. Source data are provided as a Source data file.

demonstrate that targeting Mat1a promotes the secretion of FGF21 in both, DIO and genetically induced obesity (ob/ob) models. Our data also show that these increased FGF21 levels originate from hepatocytes and are mediated by Mat1a.

The hepatokine FGF21 has a potential therapeutic effect in the treatment of obesity. Animal studies using a FGF21 analog[35] or overexpression of FGF21[36], have demonstrated the ability to reverse obesity and related co-morbidities such as insulin resistance, adiposity and NAFLD. In humans, serum FGF21 levels are increased in patients with NAFLD[37], and recent studies have demonstrated the potential beneficial effects of FGF21 in NASH. In animal models of NASH, treatment with FGF21 reduces lipotoxicity and ameliorates liver injury[38] while mice deficient in FGF21 are more prone to develop NAFLD on an obesogenic diet[39]. In a recent clinical trial, treatment with a FGF21 analog, Pegbelfermin, reduces the hepatic fat fraction, as measured by MRI, in patients with NASH[40]. Our results here confirm that targeting Mat1a protects the liver from the obesity-induced hepatosteatosis through the formation of FGF21. The antisteatotic role of Mat1a inhibition has been an unexpected finding as downregulation of Mat1a has always been associated with NAFLD and cancer development[6]. Indeed, patients with liver cirrhosis show low Mat1a levels and lower production of SAMe[41]. Moreover, Mat1a-KO mice spontaneously develop NAFLD with age; NASH at 8-month-old, which progresses to hepatocellular carcinoma (HCC) at 16-month-old[6,42]. Interestingly, in the context of obesity, we show here that Mat1a deficiency, rather than worsening the liver status, decreased the de novo lipogenesis in the liver, without affecting FAO, preventing lipid accumulation without producing liver fibrosis or infiltration. The same phenotype was observed when Mat1a-KO mice were treated with a HFD. We found that targeting Mat1a, induced BAT thermogenesis, in a mechanism in which FGF21-regulated signaling mTORC1/S6K is involved, as has been described before[20]. Targeting Mat1a in lean mice fed a CD, also increased serum FGF21 levels but not to levels found in HFD-fed mice. However, it was not linked to changes in liver FAO, as in HFD-fed mice, it did not induce changes in body weight or energy expenditure.

In this context of obesity, with high increased serum FGF21 levels, dietary lipids, fatty acids released by lipolysis of the WAT and those secreted by the liver into VLDL were mainly moved towards the BAT to be catabolized through FAO. This mechanism will promote the decrease in serum lipids, as was previously reported by Schlein et al.[43], as a role of FGF21. The transport of lipids towards the BAT along with the decreased de novo lipogenesis are mechanisms involved in protecting the liver from hepatostatosis. A role for FGF21 in modulation of de novo

lipogenesis in liver has been reported before; Xu et al.[44] found that the antisteatotic effect of FGF21 is linked to the inhibition of the nuclear sterol regulatory element binding protein-1 (SREBP-1) and the expression of a variety of genes involved in fatty acid and triglyceride synthesis. The fact that in the livers of Mat1a ASO-treated HFD-fed mice the levels of key enzymes involved in de novo lipogenesis, FAS and ACC, were highly decreased, together with the de novo lipogenesis flux, supports the hypothesis that FGF21 might be inhibiting SREBP1c also in this model. However, gene expression of Acaca, Acacb and Fasn remained unchanged suggesting that a mechanism regulating protein stability should be involved. Thus, our results confirm the beneficial effect of FGF21, which not only protects from obesity, insulin resistance, and dyslipidemia but also prevents NAFLD, as shown in DIO mice with FGF21 deficiency in hepatocytes, where the Mat1a ASO treatment increased liver TG to higher levels than in the ASO-treated DIO control mice.

A transcription factor that has been described as a regulator of FGF21 is NRF2[45], which is responsible for the antioxidant response program in the cell. It also confers resistance to various environmental stressors and plays a role in organism metabolic homeostasis[45]. The results here demonstrate that targeting Mat1a in DIO mice increased activation of NRF2 and that inhibiting NRF2 in Mat1a ASO-treated DIO mice hepatocytes reduced FGF21 secretion to normal values. Thus, in this context of obesity, the results suggest that deficiency of Mat1a induces the secretion of FGF21 through activation of NRF2.

Activation of NRF2 results in beneficial effects similar to those obtained with dietary MR; NRF2 activators prevent HFD-induced obesity and adiposity[46], increased energy expenditure, reduced blood glucose, insulin, and plasma lipid levels[47], and reduce liver lipid accumulation in DIO and diabetic mice[46,47]. However, as our results here showed, when HFD-fed mice were treated with the Mat1a ASO, the obtained phenotype was driven by FGF21 and not NRF2.

NRF2 is inactivated by Kelch-like ECH-associated protein (KEAP)1, a protein that binds it and acts as a substrate adaptor for the Cullin-3-containing E3 (CUL3-RBX1) ubiquitin ligase, which ubiquitinizes NRF2 and promotes its degradation in the proteasome[45]. Factors such as oxidative stress oxidizes KEAP1 cysteine residues, promoting a conformational change that leaves free NRF2, which is translocated directly into the nucleus[45,48]. Our results showed that the Mat1a ASO treatment in HFD-fed mice reduced the GSH levels in liver and increased lipid peroxidation, which suggest that mitochondria ROS might be elevated as it has been described in Mat1a-KO hepatocytes[27]. Moreover, the addition of GSH or the antioxidant NAC to hepatocytes from

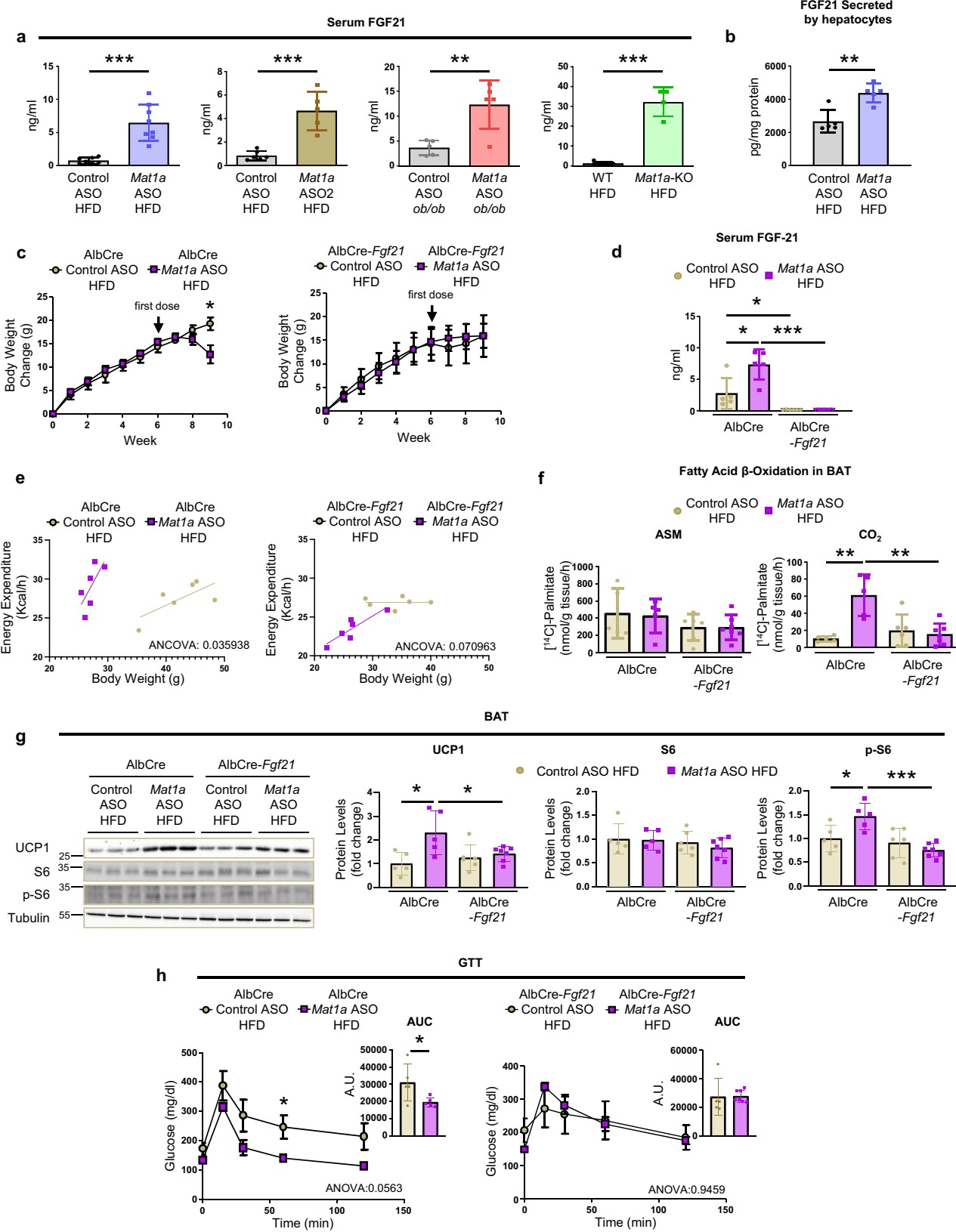

*Mat1a* ASO-treated HFD-fed mice, restored the secretion of FGF21. Thus, the increased ROS in *Mat1a* ASO-treated mice might inhibit the KEAP1-NRF2 link, releasing NRF2 and allowing its translocation into the nucleus, binding to the *Fgf21* promoter and protecting the liver from some deleterious effects driven by *Mat1a* deficiency. The results suggest that the

deficiency of SAMe, probably together with other factors related to *Mat1a* deficiency, contributes to the increased translocation of NRF2 to the nucleus. It has been recently reported that MATα1 negatively regulates cytochrome P450 2E1 (CYP2E1) expression and that *Mat1a* knockdown-mediated increase in mitochondrial ROS is CYP2E1 dependent[27]. Thus, our results suggest that

**Fig. 6 *Mat1a* deficiency induces hepatocyte secretion of fibroblast growth factor 21 (FGF21).** Two-month-old C57BL/6J, WT, *Mat1a*-KO, AlbCre and AlbCre-*Fgf21*, and 3-month-old *ob/ob* mice were fed a high-fat diet (HFD) for 4 (*ob/ob*) or 10 weeks (rest). Last 4 weeks mice were treated with *Mat1a* antisense oligonucleotides (ASO) (25 mg/kg/week), *Mat1a* ASO2 (50 mg/kg/week) or control ASO (25 and 50 mg/kg/week). Hepatocytes pooled from three HFD-fed ASO-treated C57BL/6J mice were seeded (75,000 cells/well) and incubated for 24 h. **a** Serum FGF21 levels from HFD-fed control ($n = 7$) and *Mat1a* ($n = 8$) ASO-treated C57BL/6J and HFD-fed control ($n = 5$) and *Mat1a* ($n = 5$) ASO-treated *ob/ob* mice; control ($n = 6$) and *Mat1a* ASO2 ($n = 5$)-treated C57BL/6J mice; and WT ($n = 5$) and *Mat1a*-KO ($n = 4$) mice. **b** FGF21 secreted by hepatocytes ($n = 5$/group) from HFD-fed ASO-treated mice. **c** Body weight change for HFD-fed control ($n = 5$) and *Mat1a* ($n = 5$) ASO-treated AlbCre and control ($n = 6$) and *Mat1a* ($n = 7$) ASO-treated AlbCre-*Fgf21* mice. **d** Serum FGF21 in HFD-fed control ($n = 5$) and *Mat1a* ($n = 5$) ASO-treated AlbCre and control ($n = 6$) and *Mat1a* ($n = 7$) ASO-treated AlbCre-*Fgf21* mice. **e** Energy balance of HFD-fed control ($n = 6$/group) and *Mat1a* ($n = 6$/group) ASO-treated AlbCre and AlbCre-*Fgf21* mice. **f** BAT fatty acid β-oxidation, as [$^{14}$C]-acid-soluble metabolites (ASM) and [$^{14}$C]-CO$_2$ in HFD-fed control ($n = 5$) and *Mat1a* ($n = 5$) ASO-treated AlbCre and control ($n = 6$) and *Mat1a* ($n = 7$) ASO-treated AlbCre-*Fgf21* mice. **g** Representative blots and densitometries of uncoupling protein1 (UCP1), total and phosphorylated protein S6 and Tubulin, as loading control, in BAT of HFD-fed control ($n = 5$) and *Mat1a* ($n = 5$) ASO-treated AlbCre and control ($n = 6$) and *Mat1a* ($n = 7$) ASO-treated AlbCre-*Fgf21* mice. **h** Glucose tolerance test (GTT) for HFD-fed control ($n = 5$) and *Mat1a* ($n = 5$) ASO-treated AlbCre and control ($n = 5$) and *Mat1a* ($n = 7$) ASO-treated AlbCre-*Fgf21* mice. Data are also indicated as area under the curve (AUC). Values are means ± SEM for time-courses, and means ± SD for histograms. Statistically significant differences are indicated by *$p < 0.05$, **$p < 0.01$, and ***$p < 0.001$ (two-tailed Student's test). Statistical analysis in GTT is performed by two-way ANOVA test comparing Control ASO HFD vs. *Mat1a* ASO HFD. Statistical analysis for energy expenditure is performed by two-way ANCOVA test. Source data are provided as a Source data file.

---

activation of NRF2 might also be directly mediated by *Mat1a* deficiency, through the cooperation with CYP2E1.

Finally, we propose that silencing *Mat1a* in obesity activates NRF2 in hepatocytes and induces the secretion of FGF21 to the general circulation, which increases WAT lipolysis and BAT thermogenesis, decreasing de novo lipogenesis in the liver. The movement of circulating lipids towards the BAT, to be catabolized, will reduce circulating lipids in the bloodstream (Fig. 8). In conclusion, targeting *Mat1a* prevents and reverses obesity and the obesity-related insulin resistance and hepatosteatosis.

## Methods

**Animals and housing conditions**. 10-week-old male C57BL/6J mice and C57BL/6J liver-specific fibroblast growth factor 21 (*Fgf21*) knockout (AlbCre-*Fgf21*) and 12-week-old male B6.Cg-*Lep^{ob}* (*ob/ob*) mice were used for *Mat1a* gene knockdown in the liver. 10-week-old *Mat1a* knockout (*Mat1a*-KO) male mice were also included.

Mice were fed a rodent chow diet (Teklad Global 18% Protein Rodent Diet 2018S; Envigo INC., USA) or a high-fat diet (HFD) (60% fat calories, Bioserv. F3282) during 10 weeks. For *ob/ob* mice, HFD treatment was maintained during the weeks of the ASO treatments. Mice body weight and food intake were measured weekly. All mice were housed in a temperature of 21–22 °C and 40% humidity-controlled room, with a 12 h-light/dark cycle and *ad libitum* access to food and water. Animal procedures were approved by the Ethics Committee for Animal Welfare of the University of the Basque Country UPV/EHU (CEEA 401/2015), CIMUS, University of Santiago de Compostela-Instituto de Investigación Sanitaria (15010/17/007) and Centro Nacional de Investigaciones Cardiovasculares (CNIC) (PROEX 215/18) and were conducted in conformity with the EU Directives for animal experimentation.

### In vivo assays

*Mat1a ASO treatment.* For *Mat1a* knockdown, two different antisense oligonucleotides (ASO), provided by IONIS pharmaceuticals (USA), were used. Mice fed a HFD for 10 weeks were injected intraperitoneal (IP) 25 mg/kg/week of *Mat1a* Gen 2.0 ASO (5'-CCACTTGTCATCACTCTGGT-3') or control ASO (5'-CCTTCCCT GAAGGTTCCTCC-3'), or 50 mg/kg/week of *Mat1a* Gen 2.0 ASO2 (5'-GCTCAG GAGACATTGACCAT-3') or control ASO (mentioned above), in a single dose from the sixth week of the diet. Mice were sacrificed 48 h after the last dose.

*Long-term treatments and Mat1a downregulation.* 10-week-old male C57BL/6J mice were fed a HFD for 16 weeks. During the last 9 weeks, mice were injected intraperitoneally (IP) 25 mg/kg/week of *Mat1a* ASO or control ASO, in a single dose, until the mice weight loss was stabilized. Mice were sacrificed 48 h after the last dose.

*Lentivirus vector production and administration.* Lentiviruses were produced as previously described[49] with some modifications. Human embryonic kidney (HEK)-293T cells were plated at 25–35% confluence in Dulbecco's Modified Eagle Medium (DMEM) (Gibco) supplemented with 10% fetal bovine serum (FBS) (Sigma), 200 mM L-glutamine (Lonza), and 10,000 U/ml penicillin/streptomycin (P/S) (1:1, Lonza). Transient calcium phosphate co-transfection of HEK-293T cells was done with the pGIPZ empty vector or short hairpin RNAs (shRNAs) against UCP1

(pGIPZ.UCP1 vector, V2LMM_51198, Dharmacon) and β-klotho (pGIPZ.Klb vector, V2LMM_10501, Dharmacon), together with pΔ8.9 and pVSV-G packaging plasmids. The supernatants containing the lentiviral particles were collected 48 h after removal of the calcium phosphate precipitate and were filtered through 0.45-μm filters and concentrated by ultracentrifugation for 2 h at 115,500 × *g* at 4 °C (Ultra-Clear Tubes, SW28 rotor and Optima L-100 XP Ultracentrifuge; Beckman Coulter). Viruses were resuspended in cold sterile phosphate-buffered saline (PBS) and mice were bilaterally injected in the BAT under sevoflurane anesthetics in a volume of 100 μl lentiviral particles as previously described[50].

*Corporal and interscapular temperature measurement.* Body temperature was detected by a rectal probe connected to a digital thermometer (AZ 8851K/J/T Handheld Digital Thermometer-Single, AZ Instruments Corp., Taiwan)[51]. BAT interscapular temperature was visualized and quantified by thermographic images using a FLIR T430sc Infrared Camera (FLIR Systems, Inc., Wilsonville, OR) and analyzed through Flir R-Tools specific software package[51].

*Indirect calorimetric system.* For 48 h, at the end of the ASO treatment, animals were analyzed for Energy Expenditure (EE, kcal/h), Respiratory Quotient (RQ, VCO$_2$/VO$_2$) and Locomotor Activity (LA) using an indirect calorimetric system (LabMaster; TSE Systems; Bad Homburg, Germany) as described previously[52]. The system is an open-circuit instrument that determines the energy consumed by the amount of caloric intake (kilocalories) along time (h); the ratio between the CO$_2$ production and O$_2$ consumption (VCO$_2$/VO$_2$); and the total horizontal locomotion, measured as beam breaks. Previously, all mice were acclimated to the experimental room and habituated to the system for 48 h before starting the measurements.

To test the β-adrenergic stimulation sensitivity in mice, VO$_2$ consumption was measured after a β3 adrenergic agonist treatment using the indirect calorimetric system. For that purpose, animals were acclimated to the room for 4 h, and then were treated with a single intraperitoneal dose of the β3 adrenergic agonist CL316243 in a dose of 1 mg/kg[53]. The relationship between VO$_2$ consumption and mice weight was analyzed 30 and 45 min after β-adrenergic agonist administration using an ANCOVA test.

*Insulin and glucose tolerance test.* Insulin tolerance tests (ITT) were performed following a 4 h fasting by IP injection of 0.75 U/kg insulin. Blood glucose measurements were taken using a Contour Ultra blood glucometer before and 15, 30, 60 and 120 min after injection of insulin. Glucose tolerance tests (GTT) were performed by oral gavage of glucose at a dose of 1.5 g/kg after overnight fasting. Blood glucose was measured before and 15, 30, 60 and 120 min after glucose administration. In addition, during the GTT, serum insulin levels were measured before and 30, 60, and 120 min after oral glucose administration using a commercially available ELISA kit.

*Hepatic triglyceride (TG) secretion rate.* Mice were subjected to an IP injection with Pluronic F-127 (Poloxamer P-407, Invitrogen Life Technologies, USA) at 1 g/kg after an overnight fasting[54]. Prior to injection, 2 and 4 h after, blood samples were collected, serum prepared, and TG concentrations determined. Commercially available kits were used to measure TG levels (A. Menarini Diagnostics, Spain).

*Fat tolerance test.* Fat tolerance test was performed with 10 μl/g body weight of olive oil by oral gavage after overnight fasting[16]. Circulating TG levels were measured in plasma prior to oral gavage and 1, 2, 3, and 4 h after oil administration using a commercially available kit (A. Menarini Diagnostics, Spain).

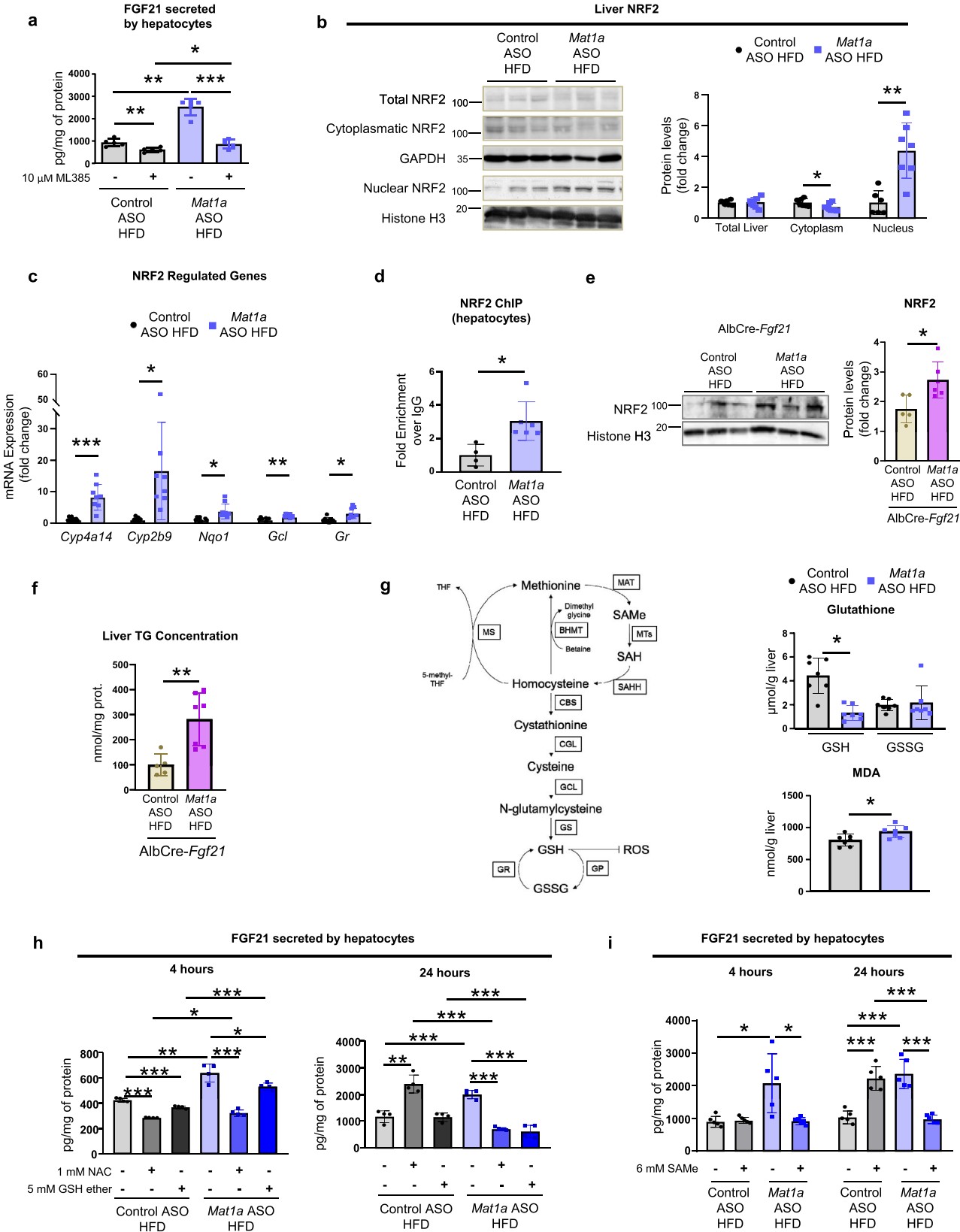

*Tissue lipid uptake*. Mice were fasted for 4 h and subjected to oral gavage with 10 μl/g body weight of olive oil emulsion containing 2 μCi [³H]-Triolein (Perkin Elmer INC, USA). Liver, epididymal white adipose tissue (WAT), brown adipose tissue (BAT), muscle and heart were harvested after 4 h. In all, 30–50 mg pieces of each tissue were weight and homogenized in PBS. Homogenate [³H]-radioactivity was measured with a scintillation counter (Tri-Carb 2810 TR, PerkinElmer, USA)[16,55].

*Tissue and serum extraction*. For serum extraction, blood was drawn from cava vein and was allowed to clot at room temperature (RT) for 30 min, then it was centrifuged at 2000 × g for 30 min at 4 °C and the supernatant was centrifuged at 10,000 × g for 10 min at 4 °C.

Liver, epididymal WAT, BAT, muscle and heart, were collected and washed in fresh cold PBS. Then, tissues were immediately split for metabolic assays,

**Fig. 7 Nuclear factor E2-related factor 2 (NRF2) mediates *Mat1a* antisense oligonucleotide-induced FGF21 hepatocyte secretion.** Two-month-old C57BL/6J mice were fed a high-fat diet (HFD) for 10 weeks. The last 4 weeks mice were treated with *Mat1a* antisense oligonucleotide (ASO) or control ASO (25 mg/kg/week) until sacrifice. Hepatocytes pooled from three HFD-fed ASO-treated mice were isolated, seeded ($10^6$ or 75,000 cells/well) and cultured for 4 or 24 h. **a** FGF21 secreted by hepatocytes from HFD-fed ASO-treated mice exposed during 24 h to 10 μM of the NRF2 inhibitor ML385 ($n = 5$) or vehicle ($n = 5$) for control ASO and NRF2 inhibitor ML385 ($n = 4$) or vehicle ($n = 5$) for *Mat1a* ASO. **b** Representative blots, and densitometries of total liver, cytoplasmic and nuclear NRF2 and glyceraldehyde-3-phosphate dehydrogenase (GAPDH) and Histone H3 from HFD-fed control ($n = 7$, 7, and 6) and *Mat1a* ($n = 7$, 7, and 7) ASO-treated mice. **c** NRF2 target genes cytochrome P450 4A14 (*Cyp4a14*), cytochrome P450 2B9 (*Cyp2b9*), NAD(P)H dehydrogenase [quinone] 1 (*Nqo1*) (Control ($n = 7$) and *Mat1a* ($n = 8$) ASO HFD), Glutamate-cysteine ligase (*Gcl*) and Glutathione reductase (*Gr*) (Control ($n = 6$) and *Mat1a* ($n = 7$) ASO HFD) mRNA expression, normalized with *Gapdh*. **d** ChIP-q-PCR analysis of NRF2 target gene *Fgf21* in hepatocytes from HFD-fed control ($n = 4$) and *Mat1a* ($n = 6$) ASO-treated mice. **e** Representative blots, and densitometry of nuclear NRF2 and Histone H3 in liver of HFD-fed control ($n = 5$) and *Mat1a* ($n = 6$) ASO-treated AlbCre-*Fgf21* mice. **f** Liver triglyceride (TG) content of HFD-fed control ($n = 5$) and *Mat1a* ($n = 7$) ASO-treated AlbCre-*Fgf21* mice. **g** Methionine cycle and transsulfuration pathway scheme. Liver reduced (GSH) and oxidized (GSSG) glutathione and malondialdehyde (MDA) levels in HFD-fed control ($n = 7$) and *Mat1a* ($n = 7$) ASO-treated mice. **h** and **i** FGF21 secreted by hepatocytes ($n = 4$ and 5/group, respectively) from HFD-fed ASO-treated mice exposed during 4 and 24 h to **h** 5 mM of GSH ether, 1 mM of N-acetylcysteine (NAC) or vehicle; and **i** to 6 mM of S-adenosylmethionine (SAMe) or vehicle. Values are represented as means ± SD. Statistically significant differences between groups are indicated by *$p < 0.05$, **$p < 0.01$, and ***$p < 0.001$ (two-tailed Student's test). Source data are provided as a Source data file.

---

histological analysis, or to be frozen in liquid nitrogen and stored at −80 °C until biochemical analyses were performed.

## Metabolic assays

*Fatty acid oxidation.* The fatty acid B-oxidation (FAO) was assessed as described before[56–59]. In all, 30 mg of freshly isolated liver and BAT and 60 mg of WAT pieces were homogenated in cold homogenization buffer and sonicated for 10 s. Then, the homogenates were centrifuged at $420 \times g$ for 10 min at 4 °C and the supernatant was collected. Approximately 500 μg of protein from the liver and BAT homogenates and 250 μg of protein from WAT homogenates were used for the assay in a volume of 60 μl. The reaction was started adding 340 μl of assay buffer containing 0.5 μCi/ml [1-14 C]-Palmitic acid to the samples, and was incubated for 30 min at 37 °C in eppendorf tubes with a Whatman paper circle in the cap. The reaction was stopped by adding 200 μl of 1 M perchloric acid, after adding 1 M NaOH in the Whatman cap to collect all the evaporated [$^{14}$C]-$CO_2$.

After 1 h, the Whatman caps were retired, and the radioactivity associated was measured in a scillation counter. The eppendorf tubes were centrifuged at $21,000 \times g$ for 10 min at 4 °C and 400 μl from the supernatant were collected to measure the radioactivity incorporated in acid-soluble metabolites (ASM).

*Measurements of oxygen consumption rate.* The respiration of BAT mitochondria was measured at 37 °C by high-resolution respirometry with the Seahorse Bioscience XF24-3 Extracellular Flux Analyzer, as described before[60]. For the measurement of the oxygen consumption rate (OCR), as the rate change of dissolved $O_2$, BAT mitochondria were isolated by a method similar to Schnaitman and Greenawalt[61] and basal respiration measurements were made in the presence of succinate and rotenone. The normalized data were expressed as pmol of $O_2$ per minute or milli-pH units (mpH) per minute, per μg mitochondrial protein.

*De novo lipogenesis.* For de novo lipogenesis fluxes, 40 mg liver pieces were incubated with the lipogenesis assay medium during 4 h. Then, liver pieces and media were collected. To assess the [$^3$H]-acetate radioactivity incorporated into TGs, lipids were extracted and separated by thin layer chromatography (TLC) as described before[62]. Lipid species were developed with iodine vapor, scratched, and the silica containing the TGs was introduced in vials with the scintillation liquid. The radioactivity was determined in a scintillation counter.

*Adipose tissue lipolysis.* Fresh WAT and BAT pieces were incubated and after 4 h, media were collected and the secreted glycerol and fatty acids were measured with commercial available kits (Wako Chemicals, USA, for fatty acids; Sigma-Aldrich, USA, for glycerol)[16].

## Histochemistries

*Hematoxylin and eosin staining.* Liver, WAT, and BAT pieces were fixed in 10% (v/v) non-buffered formalin for 24 h at 4 °C and were kept in 50% (v/v) ethanol until they were paraffinized. The paraffin blocks were prepared and cut in 5 μm-thick sections, and sections were subjected to conventional hematoxylin and eosin staining. Briefly, sections were submerged for 2.5 min in Shandon TM Harris hematoxylin (Thermo Scientific; USA), washed in water for 5 min, decolorized by immersion in 0.5% (v/v) HCl and washed with d$H_2O$. Then samples were counterstained with Eosin-Y Alcoholic (Thermo Scientific, USA) for 25 s, washed and dehydrated with increasing ethanol solutions. stained in eosin for 15 min. Finally, samples were mounted using DPX mounting medium. Representative micrographs were taken under 20x objective from upleft optical microscope.

*Sirius red.* For the evaluation of liver fibrosis, liver pieces were fixed in 10% (v/v) non-buffered formalin (Sigma-Aldrich, USA) for 24 h at 4 °C and were kept in 50% (v/v) ethanol until they were paraffinized. The paraffin blocks were prepared and cut in 5 μm-thick sections with the microtome and stained with Sirius red solution (0.01% (w/v) Fast Green FCF/0.1% (w/v) Sirius red in picric acid (Sigma-Aldrich; USA) for 30 min. Sections were then dehydrated directly in 100% alcohol and mounted in DPX mounting media. Representative micrographs were taken under 20x objective from upright optical microscope. Stained area percentage of each sample were calculated using FRIDA 1.0 software (FRamework for Image Dataset Analysis, Johns Hopkins University, USA).

*Sudan Red.* For the histological evaluation of lipid storage in liver, Sudan Red staining was performed. OCT frozen livers were cut in 8 μm-thick sections and fixed with 10% (v/v) non-buffered formalin for 2 min. Then samples were incubated with freshly prepared Sudan III stain (Sigma-Aldrich; USA) and counterstained with Mayers Hematoxylin (Sigma-Aldrich; USA). Finally, they were mounted with aqueous mounting media. Representative micrographs were taken under 20x objective. Percentage of stained area of each sample was calculated using FRIDA 1.0 software (Framework for Image Dataset Analysis, http://bui3.win.ad.jhu.edu/frida/, Johns Hopkins University; USA).

## Immunoassays

*Immunohistochemical analysis of F4/80.* For the evaluation of liver inflammation, F4/80 immunostaining was performed. OCT frozen livers were cut in 8 μm-thick sections, unmasked according to the primary antibody to be used and subjected to peroxide blocking, 3% (v/v) $H_2O_2$ in PBS, during 10 min at RT. For stainings, samples were blocked with goat anti-mouse FAB fragment (Jackson Immunor-esearch; USA), and blocked with 5% (v/v) goat serum. Then, sections were incubated with the primary antibody in DAKO antibody diluent in a 1:50 dilution during 1 h at 37 °C, followed by Envision anti-rabbit or anti-mouse (DAKO; Denmark) HRP-conjugated secondary antibody incubation. Colorimetric detections were confirmed with vector VIP chromogen (Vector; USA) and sections were counterstained with hematoxylin. Samples were mounted using DPX mounting medium. For the analysis, images were taken with an upright light microscope. Representative micrographs were taken under 20x objective. Stained area percentage of each sample were calculated using FRIDA 1.0 software.

*Western blotting.* The liver, BAT, and WAT tissues were homogenized with the homogenization buffer. In some cases, when required, liver nucleus were extracted using a subcellular proteome extraction kit (Calbiochem, Germany) following the manufacturer's instructions. The protein lysates were subjected to sodium dodecyl sulphate–polyacrylamide gel electrophoresis, electrotransferred and immobilized onto nitrocellulose membranes, and incubated with commercial primary antibody. For protein detection, membranes were incubated with a horseradish peroxidase (HRP) or a fluorescent dye containing secondary antibody. Fluorescent dye-linked secondary antibody bound proteins were detected by fluorescent excitation and emission in the fluorescent detection system Chemidoc (Bio-Rad, USA). For quantification, Quantity One 29.0 and ImageLab 6.0.1 softwares (Bio-Rad, USA) were used, and the antibodies that were used are detailed in Supplementary Table 7.

*Chromatin immunoprecipitation.* Chromatin immunoprecipitation (ChIP) analysis in isolated hepatocytes were performed as describe before[58]. Briefly, after hepatocytes isolation and attachment, crosslinking were directly performed on the culture plates by addition of formaldehyde to a 1% final concentration. Crosslinking was stopped with glycine to 0.125 M final concentration. Next, cell pellets were collected, lysated and chromatin was sonicated on a Diagenode Bioruptor to

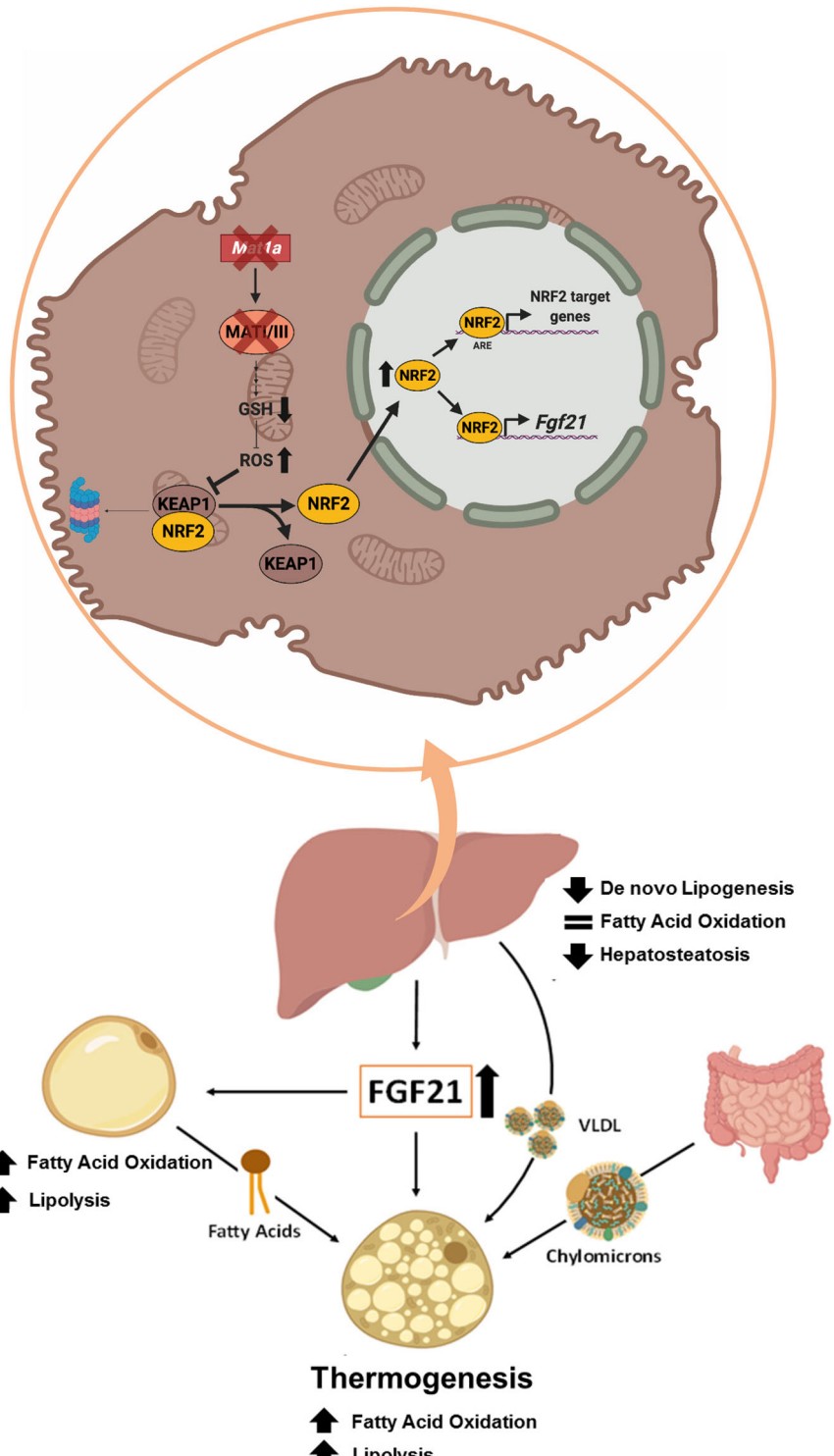

**Fig. 8 Proposed model of reversion of obesity and obesity-related hepatosteatosis by *Mat1a* antisense oligonucleotides.** *Mat1a* deficiency leads to a reduction in glutathione levels, leading to ROS accumulation in hepatocytes. The increased ROS levels modifies KEAP1, triggering NRF2 release to the nucleus. NRF2 induces transcription of target genes including FGF21. The increased FGF21 expression and secretion to the circulation induces BAT thermogenesis, lipolysis and release of fatty acids from white adipose tissue (WAT) to the blood. Lipids released to plasma as very-low-density lipoproteins (VLDL), free fatty acids and dietary lipids (chylomicrons) are channeled to the BAT to be used as energy source in the fatty acid oxidation. The increased FGF21 also reduces liver triglyceride storage and de novo lipogenesis. Thus, the NRF2-induced FGF21 secretion reverses obesity protecting from insulin resistance, hepatosteatosis and reducing plasma lipid levels. This picture was designed using BioRender graphic tool (BioRender.com).

an average length of 200–500 bp. Inmunoprecipitation of crosslinked-Protein/DNA complexes was achieved by using protein G magnetic beads (ThermoFischer), which were incubated at 4 °C for 3 h in a rotation mixer with anti-NRF2 as antibody of interest and normal mouse IgG as negative control (see Supplementary Table 7 for antibodies). Complexes were eluted and reverse crosslinking was performed. The DNA obtained was extracted and purified with Qiagen QIAquick PCR purification kit and quantification of immunoprecipitate-enriched DNA sequences was performed by real-time PCR. PCR primers were designed for the already described NRF2 binding site in the *Fgf21* regulatory region[63], which was confirmed considering the INSECT 2.0 predicted binding sites for NRF2

in the regulatory region of *Fgf21*. Primers sequences are detailed in Supplementary Table 8.

*ELISA tests.* Serum FGF21 (FGF21 Quantikine ELISA kit, Cat. No. MF2100, R&D Systems), Insulin (Ultra Sensitive Mouse Insulin ELISA Kit, Crystal Chem, Cat. No. 90080) epinephrine/adrenaline (Epinephrine/Adrenaline ELISA kit, Cat. No. CSB-E08679m, Cusabio), noradrenaline (ELISA Kit for Noradrenaline, Cat. No ABK1-E1601, Abyntek), adiponectin (Adiponectin ELISA Kit, Cat. No. ab108785, Abcam), apolipoprotein C2 (Apolipoprotein CII ELISA kit, Cat. No. CSB-EL001932MO, Cusabio) and apolipoprotein C3 (Apolipoprotein CIII ELISA Kit, Cat. No. ab217777, Abcam) levels were quantified using commercially available ELISA kits according to the manufacturers' protocols.

**Quantification of biochemical parameters**

*Quantification of TGs in liver.* Pieces of livers (30 mg) were homogenized in 10 volumes of ice-cold saline buffer. Lipids were extracted following the Folch method[64], and dissolved in isopropanol. TG quantification was performed with a commercial kit (A. Menarini Diagnostics, Spain) following manufacturer´s instructions.

*Serum measurements.* Serum ALT levels were measured using a commercially available kit (Randox, UK) following manufacturer's instructions. Serum TG and fatty acid (FA) levels in fasted and fed conditions were measured using commercially available kits (A. Menarini Diagnosis, Spain and Wako Chemicals, USA, respectively) following manufacturer's instructions.

*Lipid peroxidation assay kit.* Liver malondialdehyde (MDA) content, used as a marker for lipid peroxidation, was quantified using a commercially available kit (Sigma-Aldrich, USA) following manufacturer's instructions.

*Quantification of glutathione.* Liver reduced glutathione (GSH) and oxidized glutathione (GSSG) levels were quantified using a commercially available kit (Abcam, USA) following manufacturer's instructions.

*RNA isolation and quantitative PCR.* RNA was extracted using Trizol Reagent (Invitrogen, Spain) and cDNAs were obtained by retrotranscription (SuperScript III RT, Invitrogen, USA) following the manufacturers' instructions. Real-Time qPCRs were performed using SYBR® Green Supermix. All reactions were performed in duplicate, and expression levels were normalized to the average level of *Gapdh* and *Actb* genes in each sample using the geNorm 3.1 software. The oligonucleotides and their sequences used for qPCR analysis are collected in the Supplementary Table 8.

**In vitro experiments**

*Hepatocyte Isolation.* Hepatocytes from *Mat1a* ASO and control ASO-treated HFD-fed mice were isolated. For this, perfusion with collagenase type I was used as described previously[65]. In brief, animals were anesthetized with IP injection of sodium pentobarbital (Euthasol) (150 mg/kg of body weight), the abdomen was opened, and a catheter was inserted into the inferior vena cava while the portal vein was cut. Next, liver was washed by perfusion with Krebs-Henseleit (KH) perfusion medium equilibrated with fizzed carbogen at 37 °C. After the washing, EGTA 0.05% (w/v) was added to the KH medium and the perfusion was maintained for 5 min. Finally, an enzymatic digestion was performed during 10–12 min with KH perfusion medium supplemented with $Ca^{2+}$, 300 µg/ml collagenase and 60 µg/ml trypsin inhibitor (Roche, Switzerland). After perfusion, the liver was gently disaggregated. The viable cells were purified by density centrifugation at $40 \times g$ for 2 min at RT. Isolated pure hepatocytes were seeded over collagen and fibronectin-coated culture dishes (24-well dishes) at a density of $75 \times 10^5$ cells/well in the medium for cell adhesion. Cells were placed at 37 °C in a humidified atmosphere of 5% $CO_2$ -95% air.

*Cultures of hepatocytes.* Isolated hepatocytes were suspended in serum-free Dulbecco's modified Eagle's medium (DMEM). After 2 h of attachment, $75 \times 10^5$ cells/well in 24-well dishes were incubated with or without SAMe (6 mM), methionine-deficient medium, the protein kinase RNA-like endoplasmic reticulum kinase (PERK) inhibitor GSK2606414 (2 µM), the PPARα inhibitor GW6471 (100 µM), N-acetylcysteine (1 mM) and reduced glutathione (GSH) ester (5 mM) and the NRF2 inhibitor ML385 (10 µM). ML385 is a small molecule that binds to NRF2 and inhibits its downstream target gene expression. Specifically, ML385 binds and inhibits to the Neh1, the Cap "N" Collar Basic Leucine Zipper (CNC-bZIP) domain of NRF2, interfering with the binding of the V-Maf Avian Musculoaponeurotic Fibrosarcoma Oncogene Homolog G (MAFG)-NRF2 protein complex to regulatory DNA binding sequences[66]. Cultured hepatocytes were also treated with gene silencing small interfering RNAs (siRNAs) for *Nrf2*, *Atf4*, and *Ppara*. After the incubation period, media were collected and FGF21 levels were measured using a commercially available ELISA kit. The amount of FGF21 secreted by hepatocytes was normalized by the mg of the hepatocyte total protein.

*Brown adipose tissue (BAT) primary cell isolation and culture.* For primary adipocyte cultures, stromal vascular cells were obtained from interscapular BAT excised from HFD-fed ASO-treated C57BL/6J mice as described by Oeckl et al.[67], with minor modifications. Briefly, for brown adipocyte differentiation, confluent precursor cells were cultured with DMEM, supplemented with 10% fetal bovine serum (v/v), 40 µg/ml gentamicin and penicillin/streptomycin mixture, 850 nM insulin, 1 nM triio-dothyronine (T3), and 1 µM rosiglitazone. Cells were used 8 days later, when more than 90% of the cells were differentiated, showing fat depots. For brown adipocyte lipolysis, fatty acid β-oxidation and FGF21 secretion, confluent precursor cells were induced for 2 days in differentiation media supplemented with 1 µM dexamethasone, 500 µM 3-isobutyl-1-methylxanthine (IBMX) and 125 µM indomethacin and differentiated for the next 6 days, when more than 90% of the cells were differentiated.

*Statistical analysis.* Multiple independent experiments were conducted to verify the reproducibility of the data. For this study 310 mice have been used, from those 293 mice were treated with the ASOs (152 with control ASO, 126 with *Mat1a* ASO, and 15 with *Mat1a* ASO2), the rest were *Mat1a*-KO mice and their controls. Experiments in vivo were repeated at least ten times and the in vitro experiments were repeated at least three times. All attempts to replicate experiments were successful.

Data are represented mean ± SEM or mean ± SD as detailed in each figure. Differences between groups were tested using the two-tailed Student's *t*-test and two-way ANOVA. The relationship between Energy expenditure and body weight, and VO$_2$ consumption and body weight was tested with two-way ANCOVA[68]. Significance was defined as $p \leq 0.05$. These analyses were performed using GraphPad Prism 8.0 and Excel software 2016.

**Reporting summary**. Further information on research design is available in the Nature Research Reporting Summary linked to this article.

## Data availability

No data sets with mandated depositions are presented in the study. The authors declare that the data supporting the findings of this study are available within the paper and its supplementary information files. The data generated in this study are provided in the supplementary information/Source data file. Source data are provided with this paper.

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

## Acknowledgements

This work was supported by Ayudas para apoyar grupos de investigación del sistema Universitario Vasco (IT971-16) and MCIU/AEI/FEDER, UE (RTI2018-095134-B-100) (to P.A.), (RTI2018-099413-B-I00 and RED2018-102379-T) (to R.N.), PID2020-119486RB-100 (to M.V.R.) and (RTI2018-096759-A-100) (to T.C.D). EFSD/Lilly

European Diabetes Research Program, MICIU (PID2019-104399RB-I00), Fundación AECC PROYE19047SABI, and Comunidad de Madrid IMMUNOTHERCAN-CM B2017/BMD-3733 (to G.S.). La CAIXA Foundation LCF/PR/HP17/52190004, MINECO-FEDER SAF2017-87301-R, AYUDAS FUNDACIÓN BBVA A EQUIPOS DE INVESTIGACIÓN CIENTÍFICA UMBRELLA 2018 and AECC Scientific Foundation, grant name: Rare Cancers 2017 (to M.L.M.-C.). AECC Scientific Foundation (to T.C.D.). Xunta de Galicia 2020-PG015 (to R.N.) Gilead Sciences International Research Scholars Program in Liver Disease (to M.V.R.). Personal fellows: E.P.F. was awarded with Juan de la Cierva-Formación, FJC2018-035449-I. C.F. was awarded with Sara Borrell (CD19/00078). CIC bioGUNE thanks MCIU for the Severo Ochoa Excellence Accreditation (SEV-2016-0644). The authors thank Dr. Manuel Lafita´s laboratory (Getxo, Bizkaia, Spain) for his valuable help in the analysis of biochemical parameters.

## Author contributions

D.S.U., X.B., and P.A. designed the project; D.S.U., X.B., G.S., M.L.M.-C., R.N., and P.A. designed experimental protocols; D.S.U., X.B., P.O., M.A.-B., A.N.-Z., M.R.G., N.G., C.F., E.M.N., A.M., B.P., V.M.-R., T.C.D, J.L.G.-R., B.G.-S., V.G.J., F.G.-R., I.A., E.P.-F., I.M.-G., M.V.-R., S.B., R.L., J.M.B., W.-K.S., G.S., M.L.M-C, R.N., and P.A. contributed to investigations and data analysis; D.S.U., X.B., J.L.G.-R., W.-K.S., M.L.M.-C., G.S., R.N., and P.A. contributed to discussions; D.S.U., X.B., and P.A. prepared the figures with input from all the other authors, D.S.U. and P.A. wrote the manuscript with input from all the other authors.

## Competing interests

M.L.M.-C. is a consultant of Mitotherapeutix. P.A., X.B., and D.S.U. have a patent application related to this work (EP19762826.6). The remaining authors declare no competing interests.
