## [Peer Review File · Nature Communications]

Methionine adenosyltransferase 1a antisense oligonucleotides activate the liver-brown adipose tissue axis preventing obesity and associated hepatosteatosisREVIEWER COMMENTS

Reviewer #1 (Remarks to the Author):

In this manuscript, Sáenz de Urturi and colleagues assessed the role of methionine adenosyltransferase 1 (Mat1a) on systemic energy metabolism. The authors report that antisense oligonucleotides (ASO) targeting Mat1a protect from obesity and hepatic steatosis through mechanisms that include NRF2-mediated stimulation of FGF21 secretion, leading to increased lipid supply to BAT and activation of BAT thermogenesis and energy expenditure.

The manuscript is well written, easy to understand and deals with an interesting and innovative research question. The hypothesis is clearly defined and the conducted studies are overall informative. The manuscript is considered as of importance for its field. However, there remains some criticism.

-The authors statement that "administration of both Mat1a ASO or Mat1a ASO2 to mice was safe" is a quite hard statement that is solely based on the measures of some markers of liver and renal damage that are shown in Table 1. But measurement of these markers is not sufficient to claim that the ASOs are overall safe. I suggest to re-phrase the sentence to just say that the treatment had no effect on these liver and renal markers without statement on their overall safety.

-Another point is that the authors key message of this paper is that the administration of the Mat1a oligonucleotides prevent obesity and hepatic steatosis (as e.g. shown in SF1d and Fig. 1C). But in Table 1 the authors show that the administration of these oligos has no effect on body weight. This raises the question as to how long after treatment those safety measures in Table 1 were taken and whether the used doses and treatment durations are similar to figures in which administration of the ASOs affected body weight? If the safety studies were done at a different time point or using a lower dose, then the safety studies need to be repeated to measure the safety markers after drug-induced weight loss (so at a dose that affects body weight).

- The Table 1 just states the average values without standard deviation/SEM or statistics between the groups. Also, the figure legend of Table 1 lacks next to standard deviation also data on doses, treatment duration and statistics. How often and at which dose were the oligos injected and what is the resulting gene knock-out efficiency at this dose used in Table 1?

- The often here reported sample size of N=4-5 mice each group is a weakness of this manuscript. Many higher journals demand higher sample sizes to ensure reliability of the results. This is a frequently criticized factor when it comes to irreproducibility of data and journals should take this very serious.

- In Suppl. Fig 1a and b, the authors show solid decrease of Mat1a after treatment with the Mat1a ASO. Was the used dose of the oligos identical to the dose used in Table 1? If yes, then I wonder why the oligos shown in Table 1 didn't induce weight loss, as shown in SF1D and Fig 1C? If a different dose was used for SF1 and Fig 1C as compared to Table 1, then all the measures on the liver and renal safety shown in Table 1 say nothing about safety at a dose that is required to achieve weight loss. So the question would be whether there are changes in marker for liver and renal damage after drug induced weight loss? The question on drug safety after weight loss is actually of key importance since the data in Fig. 1C show that the oligo treated mice lose (based on a crude judgement of Fig1C) roughly 65% of their body weight in just 2 weeks and there is no plateau in weight loss reached at the end of the study, which means we don't even know the maximal weight loss. Since this is quite untypical for a safe pharmacological weight-loss study, a long-term study that shows the maximal weight loss efficiency with plateaued weight loss for at least 1-2 weeks and subsequent measurement of marker indicative of liver and renal damage seems advisable.

- I also would appreciate seeing the body weight data per gram in addition to the shown change in g.

- please give the values in SF1i as mg/dl as is done for the GTT in SF1h

- In Fig. 1g, I suggest to only show the ANCOVA graph and avoid the bar graph since it's meanwhile commonly accepted that energy expenditure can't be corrected by body weight. It would also be nice to see the resting metabolic rate.

- It would be nice to measure body core body temperature of these mice during the oligo-induced weight loss. Can it be that these mice get fever and have therefore an increased energy expenditure and dramatic weight loss? What about the increase in energy expenditure after cold exposure? Also seahorse measurement of oxygen consumption (at baseline and after treatment with a beta agonist) in isolated BAT primary cells would add value to the manuscript.

I noticed that the sample sizes in Fig 1 range from N=5-8 in the Mat1a oligo treated group and from 4-7 in the control ASO groups. Since we have only one group of mice treated with the Mat1a oligos, I conclude that up to 3 out of 8 mice from this group were missing in some panels of the figure. It would be good to get an explanation here. Also the AUC data do not show individual data points as is required by the journal policies. Also, N=4 mice per group for the ASO treated controls seems very low and is likely underpowered.

How does the weight loss induced by the Mat1a oligos reflect to body composition (fat and lean tissue mass)?

The data showing resistance to DIO in the Mat1a KO mice are interesting. One aspect of this paper is that all data are based on a full global Mat1a ko (either full germline KO or the oligo-induced KO). Hence, it can't be ruled out that extrahepatic effects of Mat1a affect systems metabolism and energy expenditure. Since the authors particularly claim here that the liver-BAT axis is causal for the metabolic effects of Mat1a, a liver-specific KO of Mat1a would be appreciated to consolidate this hypothesis. If a floxed mouse is unavailable, then a faster approach would be to use an AAV-mediated approach to knock-down Mat1a in the liver.

- Please show individual data points in all AUC panels.

Fig. 4. The role of Mat1a1 on BAT function is interesting, but some additional data would help to strengthen the manuscript. It would be valuable to see energy expenditure increases after stimulation with a beta agonist, both in the Mat1a ASO mice but also using seahorse studies in isolated BAT primary cells. - Fig. 4f shows the energy expenditure induced by a beta agonist corrected by body weight, which is not appropriate unless the mice do not differ in body weight? So these data need to be re-analyzed if the mice differ in body weight. Also, the increase in energy expenditure by the beta agonist is very low and not higher as the first baseline data points. This is very untypical and indicates that this study didn't work well. I suggest to repeat this study. Do the BAT primary cells differ from the controls in adipocyte differentiation? Measurement of differentiation marker (Adiponectin, Pparg, FAS etc) during different time points of adipocyte differentiation would be valuable. Do Mat1a1 Ko mice show a difference in energy expenditure at cold exposure? Do Mat1a1 Ko mice show a difference in energy expenditure at thermoneutrality? Also, very interesting would be to metabolically phenotype UCP1 ko mice that have been treated with the Mat1a1 ASOs. Would the resistance to obesity and the thermogenic effect disappear in these mice? This would be very informative to consolidate the here proposed mechanism. This study also seems valuable since the UCP1 protein changes in BAT are only 0.5-fold increased (Fig. 4c), so also UCP1 independent effects can account for the energy expenditure phenotype of the Mat1a ASO mice. Are levels of catecholamines changed in plasma and BAT of the Mat1a ASO mice?

The molecular mechanism leading to the increased Ucp1 protein level can be strengthened. It would be nice to clarify whether this is a cell autonomous effect, hence demonstration of Ucp1 protein levels in isolated BAT primary cells (+/- treatment with a beta agonist) would be informative. Are key activators of Ucp1 (catecholamines) increased in the Ko mice? Are transcriptional regulator of UCP1 increased in the KO mice (p-p38 alpha and beta, other MAP Kinases, PKA, PGC1a, etc.)?

Figure 5a. Please show body weight data in gram, not only as change in gram. The body weight change in the ob/ob mice is likewise very dramatic despite unchanged food intake. Since the authors attribute this to changes in energy expenditure, it has to be ruled out that differences in food efficiency and metabolizable energy exist. So bomb calorimetry with assessment of assimilated/metabolized energy are required to truly attribute these changes to energy expenditure. Also, please show individual data points for the AUC and show the ITT data as mg/dl as has been done for the GTT in Fig 5d.

In Fig 6, the authors show increased FGF21 levels in response to treatment with Mat1a ASO. Since FGF21 is secreted under conditions of fasting/weight loss/increased fatty acid oxidation, it seems fair to ask whether this increase in FGF21 is causal to Mat1a deficiency or rather a consequence of the increased lipid oxidation and weight loss? So a calorie restriction study comparing Mat1a ASO mice with mice that were weight-matched to the Mat1a ASO mice would give valuable information as to whether FGF21 is regulated by Mat1a or rather the weight loss.

In Figure 6, the authors show that Mat1a ASO fails to affect body weight in liver-specific FGF21 ko mice. Since the key focus of this manuscript is that Mat1a KO increases energy expenditure via the liver FGF21-BAT axis, the reviewer is surprised that the authors did not measure energy expenditure in the Mat1a ASO treated liver-specific FGF21 ko mice. This seems crucial to consolidate their key working hypothesis. How is energy expenditure in the Mat1a ASO treated liver-specific FGF21 ko mice? What about protein levels of UCP1, PGC1a, p38, PKA etc? Are levels of catecholamines different between wt and the Mat1a ASO treated liver-specific FGF21 ko mice? None of this has been done but still the authors claim the FGF21-BAT axis as the key signaling mechanism leading to increased BAT thermogenesis. This has to be significantly strengthened. The ultimate experiment would be to knock-out beta Klotho in the BAT to see whether treatment with Mat1a ASO fails to affect body weight and energy expenditure in these mice. Unfortunately, in the current version, the Mat1a - FGF21-BAT link is not sufficiently addressed experimentally.

Fig. 7a and e lack data on NRF2 outside the nucleus. IP-studies to analyze whether FGF21 can bind NRF2 would help to better understand the NRF2-FGF21 link. These studies should be done in liver and BAT.

Reviewer #2 (Remarks to the Author):

Comments for Authors

The Introduction describes an association between high methionine intake and obesity, and dietary methionine restriction and reduced adiposity. Previous studies have shown that knocking out MAT1A also produces hypermethionemia and results in spontaneous development of fatty liver disease. What was the rationale for proposing that knocking down MAT1A with oligonucleotides would result in a phenotype that was opposite from the MAT1A knockout animals with respect to hepatic lipid metabolism?

Near the end of the Introduction, the authors state that chronic changes in liver SAME levels in lean mice have been associated with the onset and progression of NAFLD with age, but isn't that due to a decrease in MAT1A/MAT2A,2B expression ratios that decrease the production of SAME? Why wouldn't knocking down MAT1A produce the same changes in SAME (and serum methionine) and if those metabolites are causative to the onset and progression of NAFLD, why would those changes produce opposite metabolic effects in the current model?

The authors have called ex vivo synthesis of fatty acids in pieces of liver de novo lipogenesis while the term is normally reserved for in vivo measurement of fatty acid synthesis. This needs to be clarified for the reader.

In the Results, the authors have interpreted their triolein uptake data to mean that the substrate fueling an increased thermogenic rate is coming from outside brown adipose tissue. The results are convincing that triolein uptake is increased in BAT but what is the evidence that it is being oxidized? Significant previous work in BAT has shown that DNL and fat oxidation are coupled during cold-induced increases in BAT thermogenesis. This work suggests that the fatty acids being oxidized are being synthesized in BAT. These conclusions are supported by work showing that genetic or pharmacological impairment of DNL or oxidation compromises cold-induced

thermogenesis.

The authors evidence of increased thermogenesis comes from Fig. 4F where they show that a beta3-AR agonist increases O₂ consumption more in Mat1a ASO mice than Control ASO mice. However, these results are compromised by the problem of not describing how this experimental protocol was conducted and how the data were analyzed. How was the beta3-AR agonist given to the mice while they were in the calorimeters? The results in Fig. 4F show no break in the collection of data associated with this event. More troubling is how the VO₂ consumption data are being scaled. The ordinate is labeled ml/kg/h, indicating that the O₂ consumption data is being scaled by BWs, which are ~15g lower in the Mat1a ASO group. As described in multiple manuscripts describing the correct approach to analyzing IDC data in recent years, big differences in BW confound group comparisons by over-amplifying the calculated EE in smaller animals within the groups.

The data presented in Fig. 7 provide support for the idea that knocking down Mat1a in liver activates FGF21 release from the liver through a NRF2-dependent mechanism. The authors propose that this occurs after nuclear localization of NRF2 and transcriptional activation of FGF21. If the authors are able to detect nuclear localization of NRF2 with available reagents, they should confirm the direct role of NRF2 using ChIP assays to show that NRF2 is in fact being recruited to the FGF21 promoter. If NRF2 is necessary and sufficient for transcriptional activation of FGF21 after Mat1a ASO treatment, they should also show that hepatic deletion of NRF2 blocks the induction of FGF21. In addition, more information is needed about the NRF2 inhibitor used in Fig. 7D. Its source needs to be identified, along with a reference to a publication demonstrating its efficacy and specificity.

Specific Points

1. The description of the methods used to conclude that EE was affected by the treatments is inadequate. It refers to a previous publication but when that publication is examined it provides no details on how the calorimetry data were analyzed. The problem must be rectified in the current paper. Fig. 1G is used to make a claim that EE is increased by the MAT1A ASO but the data in the figure shows that EE was scaled by kg but does not define kg of what. Moreover, the accompanying figure panel indicates that there is no relationship between EE and BW over a weight range of 24-29 grams. This seems improbable since EE should increase roughly in proportion to BW raised to some power < 1 . An additional problem is that Fig. 1C shows that the two groups of mice being compared in Fig. 1G have approximately the same spread in BWs of mice within each group while Fig. 1C shows that the mice in the two groups were ~15 g different in BW. Together, this creates a lot of confusion for the reader, even for one experienced in the analysis of indirect calorimetry data. Please see <https://www.nature.com/articles/s41592-019-0513-9> for guidance in analyzing and presenting EE data.
2. OGTTs measure a composite of glucose absorption rates, alterations in B-cell sensitivity to glucose, and insulin-dependent glucose excursions. Given the relatively small improvement in insulin tolerance and only at the last time point, the substantial improvement in glucose tolerance shown in Fig 1E points to changes in the first two factors rather than solely insulin sensitivity. <https://pubmed.ncbi.nlm.nih.gov/18670420/> The 60 min time point shouldn't even be considered as a marker of insulin action. If you don't see an improvement in the first 20 min, it's probably not increased insulin action. What's more, the substantial improvement in glucose tolerance after Mat1a ASO is likely secondary to the loss in BW.
3. What were the fasting insulin concentrations in the two groups of mice?
4. From looking at supplementary Fig. 1F, there appears to be a problem in the way the MAT1A KO mice responded to the HF diet. After it was introduced, the mice lost ~5 g in the first week and then stabilized their weight over the following 8 wks while the WT mice gained over 20 g. It would help the reader to understand the data better if the authors would present the actual BWs of the mice in the two groups and the actual mean food intakes for the two groups over time. Presenting the data as cumulative food intake makes it more difficult to see what is happening over time to food intake. This suggestion also applies to the MAT1A ASO mice, where it would be better to see the actual BWs of the mice in the two groups and their week by week food consumption rates. Looking at supplemental Figs 1f and 1g, it is hard to believe that mice differing in BW by 25 g are eating exactly the same amount of food. Lastly the actual BWs of the animals in each group is important to know relative to the analysis of the indirect calorimetry data. The authors should add another figure panel and show the actual BWs of the mice in each group.
5. The variability of the Sirius Red and F4/80 measures is concerning, indicating that this approach to detecting fibrosis and inflammation is insensitive to the point of being useless.

6. In Fig. 3D, triolein uptake is expressed per g of tissue. What do the results look like when corrected for differences in tissue weight between treatment groups?
7. There is a problem with the interpretation of the fatty acid oxidation data in the sense that the assays do not measure in situ rates of fatty acid oxidation. With a fixed amount of C14-palmitic acid present in the in vitro incubations, the assays are providing an estimate of Vmax or capacity of the tissues from the two groups to oxidize fatty acids. This is not a measure of the rate yet it is how the figures summarizing this data is labeled.
8. It's unclear what the isoproterenol-dependent increase in ex vivo lipolysis adds to the story and why the responses from the Control ASO CD mice were not included in Fig. 4d. Lipolytic responsiveness to beta-adrenergic agonists is impacted by a number of factors and what the authors are showing is probably mostly dependent on the decrease in adiposity and fat cell size in the Mat1a ASO group.
9. Which WAT depot was used to produce the data presented in Fig 4? It doesn't seem to be specified in either the Methods or the fig legend. This is important because the capacity of WAT to undergo remodeling is depot specific.
10. As with Fig 1, please present the actual BW means of the mice in each group, as well as their actual food intakes each week.
11. Ob/ob mice typically have high fasting insulin levels by 12-15 wks of age. What impact did the 5-6 g difference in BW produced by the Mat1a ASA have on fasting insulin levels?
12. The problem noted above for FA oxidation measurements is also applicable in Fig. 5F. The authors are measuring the amount of palmitate oxidation with a fixed amount of palmitate and concluding that the measurement is indicative of in vivo fatty acid oxidation in BAT. The data do not fully support this conclusion.
13. The authors measurement of PPAR α expression in response to Mat1a treatment is not an effective approach to determining whether the increase in serum FGF21 is due to PPAR α activation. The same criticism should be considered for measurement of nuclear ATF4 by Western, particularly since it is widely known in the field that reliable measurement of endogenous ATF4 by Western is difficult to establish. The data in supplementary Fig 5 certainly do not meet the standard for ruling out involvement of either transcription factor in Mat1a ASO-dependent induction of hepatic FGF21.
14. In the last paragraph of the Results, the authors examine FGF21 in the media of hepatocytes from mice after knocking down Mat1a and adding SAME to the media. How much SAME was being added to the media and what proportion of the amount being added is ending up inside the cells? There seems to be an assumption that SAME is crossing the cell membrane. What is the evidence for that occurring?

Additional Points

1. It would have been helpful if page numbers were provided in the MS.
2. Please define which oligos were used in supplementary table 1. Is ASO the control oligo and ASO2 the antisense oligo? In addition, measures of reproducibility must be provided.
3. Please specify the temperature at which the mice were housed.

Reviewer #3 (Remarks to the Author):

Saenz de Urturi and colleagues present an interesting set of findings in which deletion of MAT1, the enzyme that catalyzes the first step in the methionine cycle, alters energy balance and prevents obesity of dietary and genetic models of obesity. This was associated with reductions in hepatic steatosis and improvements in glucose tolerance. The energetic phenotype was attributed to increased levels of FGF21 which in turn, the investigators posit, are due to activation of NRF1. The experiments were well designed and the use of two separate ASO's against MAT1 demonstrate experimental rigor. The manuscript was also well written, though there are sections that could have benefited from additional editing for clarity in English. The overall conclusions are interesting and important though there are additional considerations that should be addressed:

1. Perhaps the most intriguing finding is the enhancement of BAT function in MAT1 ASO treated mice. The authors point to increased fatty acid uptake after an oral challenge of lipid, increased UCP1 expression, oxygen consumption and lipolysis that seem to depend on FGF21. But, several questions remain. First, what about BAT mass in MAT1 ASO treated animals? Is this different? Though FA uptake was not altered in WAT, was there any evidence of browning in WAT? Was there

an alteration in RQ?

2. The authors hypothesize that there is a “channeling” of lipids to BAT of MAT1 ASO treated animals? How is this channeling occurring? Is there an increased expression of LPL in BAT?

3. How do the authors reconcile the finding of decreased DNL (using an ex vivo assay) and decreased expression of key lipogenic enzymes with the absence of any difference in absolute VLDL secretion rates? Also, since ACC2 expression may regulate rates of lipid oxidation, mRNA expression of ACC1 and ACC2 should be assessed to determine if the main difference is in ACC1.

4. Though there are impressive differences in liver TG and improvements in glucose tolerance, there may also be changes in muscle TG concentration. A reduction in muscle TG would better account for the improvements in glucose tolerance. It would be helpful to report plasma insulin concentration under fasting and during the glucose tolerance tests to better assess the changes in glucose tolerance.

5. The increase in NRF2 expression is of significant interest. The investigators present data that suggest that the increase in oxidative stress may account for the increased expression of NRF2. The experiments in Fig 4c/d (which would benefit from a clearer explanation) suggest that simply trying to manipulate methionine levels does not result in the expected changes in FGF21. Is it possible to experimentally modulate GSH concentration to see if this is the mechanism by which MAT1 ko regulates NRF2 expression? Perhaps using an ester of GSH (e.g. GSH-methyl or ethyl esters?) Alternatively, perhaps other mechanisms are at work? What about NRF2 mRNA expression? If transcription is altered, perhaps MAT1 ko is altering DNA methylation.. a mechanism that perhaps could also explain alteration in ACC and FAS expression? If transcription is not altered.

6. Is MAT1a really a viable target? Some have suggested that MAT1 deficiency might impair liver regeneration (PMID: 15033934). Are there negative consequences to MAT1 inhibition that we should be aware of?

Point-by-point response to reviewers:

Reviewer #1 (Reviewer Comments to the Author):

In this manuscript, Sáenz de Urturi and colleagues assessed the role of methionine adenosyltransferase 1 (Mat1a) on systemic energy metabolism. The authors report that antisense oligonucleotides (ASO) targeting Mat1a protect from obesity and hepatic steatosis through mechanisms that include NRF2-mediated stimulation of FGF21 secretion, leading to increased lipid supply to BAT and activation of BAT thermogenesis and energy expenditure.

The manuscript is well written, easy to understand and deals with an interesting and innovative research question. The hypothesis is clearly defined and the conducted studies are overall informative. The manuscript is considered as of importance for its field. However, there remains some criticism.

Response: Thank you for your kind words and all of your suggestions along the revision. We have performed mostly all the proposed experiments and we have concluded this revision with more information about the mechanism involved. Before going point by point we just want to **highlight** the following:

- 1) The **high reproducibility** of this study. The treatment with the ASOs has been performed several times in different experiments and different labs, obtaining the same phenotype.
- 2) This is a very **comprehensive study** showing mechanistically, how coordinated changes in gene expression, protein levels and metabolic fluxes occur. Experiments have been performed *in vivo* and *in vitro*, in different tissues and with different cell types (preadipocytes and hepatocytes).
- 3) **Treatment with Mat1a ASO** does not induce liver or renal damage.

-The authors statement that “administration of both Mat1a ASO or Mat1a ASO2 to mice was safe” is a quite hard statement that is solely based on the measures of some markers of liver and renal damage that are shown in Table 1. But measurement of these markers is not sufficient to claim that the ASOs are overall safe. I suggest to re-phrase the sentence to just say that the treatment had no effect on these liver and renal markers without statement on their overall safety.

Response: As suggested, we have re-phrased the sentence.

-Another point is that the authors key message of this paper is that the administration of the Mat1a oligonucleotides prevent obesity and hepatic steatosis (as e.g. shown in SF1d and Fig. 1C). But in Table 1 the authors show that the administration of these oligos has no effect on body weight. This raises the question as to how long after treatment those safety measures in Table 1 were taken and whether the used doses and treatment durations are similar to figures in which administration of the ASOs affected body weight? If the safety studies were done at a different time point or using a lower dose, then the safety studies need to be repeated to measure the safety markers after drug-induced weight loss (so at a dose that affects body weight). The Table 1 just states the average values without standard deviation/SEM or statistics between the groups. Also, the figure legend of Table 1 lacks next to standard deviation also data on doses, treatment duration and statistics. How often and at which dose were the oligos injected and what is the resulting gene knock-out efficiency at this dose used in Table 1?

Response: The information presented in the **Supplementary table 1** of the first version of the manuscript (below) was provided by IONIS Pharmaceuticals. Some of the original information was not available; we now present a complete set of information in the Table below. As per IONIS Pharmaceuticals protocols, all ASO that are produced are evaluated for potential toxicities: chow-diet (CD)-fed mice are treated with 50 mg/kg/week ASO for 4 weeks; results shown in this table below were obtained under these conditions. As per results in this Table, ASOs did not induce any alteration in liver or kidney function parameters or in body weight in CD-fed mice (as the reviewer noticed). In response to the reviewer, we have now included NEW information on the effects of *Mat1a* ASOs. These new studies showed that under prolonged HFD feeding, ASO treatment resulted in initial weight loss, but the weight stabilizes when this reaches a level comparable with CD-fed mice (New Supplementary Fig 1j) Thus, these results show that *Mat1a* ASO induces body weight loss in obese mice until they reach a weight of a lean mouse (new **supplementary table 2** and new **supplementary Fig. 1j**).

	Saline	Mat1a ASO	Mat1a ASO2
ALB (g/dl) (2.5-4.8)	3.1±0.2	2.9±0.2	3.0±0.1
ALT (IU/l) (28-184)	66±33	26±3*	31±11
AST (IU/l) (55-251)	97±39	61±26	60±13
TBIL (mg/dl) (0.1-0.7)	0.24±0.06	0.15±0.03	0.19±0.03
CRE (mg/dl) (0.097-0.184)	0.13±0.03	0.13±0.01	0.17±0.03*
BUN (mg/dl) (13.6-34.8)	26±5	23±1	28±1***
Body Weight (g)	26±1	25±3	26±2
Liver (g)	1.3±0.1	1.6±0.2	1.6±0.04*
Kidney (g)	0.4±0.06	0.3±0.02	0.3±0.04
Spleen (g)	0.1±0.01	0.1±0.01	0.1±0.02

Supplementary Table 1 of the first version of the manuscript. Effects of *Mat1a* ASO and *Mat1a* ASO2 on liver and kidney parameters. Chow diet (CD) fed mice were treated with 50 mg/kg/week ASO for 4 weeks (n=4). Values are represented as means ± SD. Statistically significant differences between groups are indicated by * when p<0.05 and *** when p<0.001 (Student's test) for comparison between the *Mat1a* ASO and the saline.

To avoid any confusions in this Revised manuscript, and as suggested by the reviewer, Supplementary table 1 from the first version of the manuscript has been removed and this is replaced with a NEW table 1, a NEW supplementary table 1 and a NEW supplementary table 2. In the New table 1 and NEW supplementary table 1, we present liver and kidney function parameters obtained after targeting *Mat1a* with ASO or ASO2, respectively, in mice fed a HFD for 10 weeks, and where ASOs were administrated from the 6th week. In the New supplementary table 2, the parameters of liver and kidney function were obtained from a separate long-term study where mice fed a HFD were treated with the ASOs from the 6th week until maximal weight had occurred and after weight was stable for 2 weeks (**Supplementary Fig. 1j**). Of note, for the analysis of all these parameters, serum samples were sent to Dr. Manuel Lafita's laboratory (Getxo, Bizkaia), where creatinine and urea were quantified as parameters of kidney function (instead of creatine and BUN; Ionis Pharmaceuticals measurements)

- The often here reported sample size of N=4-5 mice each group is a weakness of this manuscript. Many higher journals demand higher sample sizes to ensure reliability of the results. This is a frequently criticized factor when it comes to irreproducibility of data and journals should take this very serious.

Response: Thank you for your comments. We should have presented our experimental design and data more clearly. Essentially, the data obtained were highly reproducible. For this study 310 mice have been used, from those 293 mice were treated with the ASOs (152 with Control ASO, 126 with *Mat1a* ASO and 15 with *Mat1* ASO2), the rest were *Mat1a*-KO mice and their controls. Independent experiments using *Mat1a* ASO and control ASO have been performed several times, even in different laboratories (UPV/EHU, CNIC and CIMUS). Weight loss in HFD fed mice was consistently observed. These multiple experiments were conducted to address specific end-points/ purpose: i.e., BAT experiments (metabolic fluxes analysis, where the amount of tissue required is high, western blottings, qPCRs, histochemistry, isolation of preadipocytes), liver experiments (metabolic fluxes, western blottings, qPCRs, histochemistry, isolation of hepatocytes), WAT experiments (metabolic fluxes, western blottings, qPCRs, histochemistry), *in vivo* lipoprotein metabolic assays (hepatic triglyceride (TG) secretion rate, dietary lipid tolerance test and uptake by tissues), glucose (GTT) and insulin tolerance tests (ITT), serum analysis of parameters and metabolic cages.

Besides, we have also corroborated some results and mechanisms with:

- a) ASO2 injections (Fig 6a, Supplementary Fig. 1f, Supplementary Fig.4a, Supplementary Fig.6b, c).
- b) *Mat1a*-KO mice (Fig. 6a, Supplementary Fig. 2, Supplementary Fig.4a, Supplementary Fig.6d, Supplementary Fig. 8b).
- c) *ob/ob* mice to check the results induced by the ASO (Fig. 5, Fig. 6a).
- d) Liver specific FGF21-KO mice (Fig. 6; Supplementary Fig. 8c, d, e, f; Supplementary Fig. 11b).
- e) Knock-down of UCP1 and β -klotho in BAT (Supplementary Fig. 9).

In summary, besides many data obtained after the administration of *Mat1a* ASO, we have used 6 different animal models, and all of them point towards the same direction. Thus, we strongly believe that the reproducibility of the data is a strength of this project. That being said, we agree with the reviewer that it has been a mistake on our side not being able to show it clearly in the first version.

For this Revision, we have performed additional animal experiments with ASOs: i.e., GTT and ITT experiments. GTT and ITT analysis were performed after the first ASO injection (when weight loss was not that evident) (Supplementary Fig. 1h), and at the end of the treatment (after significant weight loss) (Fig. 1f, g).

- In Suppl. Fig 1a and b, the authors show solid decrease of Mat1a after treatment with the Mat1a ASO. Was the used dose of the oligos identical to the dose used in Table 1? If yes, then I wonder why the oligos shown in Table 1 didn't induce weight loss, as shown in SF1D and Fig 1C? If a different dose was used for SF1 and Fig 1C as compared to Table 1, then all the measures on the liver and renal safety shown in Table 1 say nothing about safety at a dose that is required to achieve weight loss. So the question would be whether there are changes in marker for liver and renal damage after drug induced weight loss? The question on drug safety after weight loss is actually of key importance since the data in Fig. 1C show that the oligo treated mice lose (based on a crude judgement of Fig1C) roughly 65% of their body weight in just 2 weeks and there is no plateau in weight loss reached at the end of the study, which means we don't even know the maximal weight loss. Since this is quite untypical for a safe pharmacological weight-loss study, a long-term study that shows the maximal weight loss efficiency with plateaued weight loss for at least 1-2 weeks and subsequent measurement of marker indicative of liver and renal damage seems advisable.

Response: Thank you for these interesting suggestions. As described above, we have now performed additional experiments to address these questions. . The results obtained show that targeting *Mat1a* did not lead to any significant perturbations in liver or kidney function parameters even after demonstrable weight loss (Table 1, supplementary table 1 and supplementary table 2). However, in the long-term study where weight loss has stabilized for 2 weeks, parameters of liver function were slightly increased albeit within the normal range (Supplementary Table 2).

- I also would appreciate seeing the body weight data per gram in addition to the shown change in g.

Response: As proposed, we have added in each figure the body weight in g. Sometimes this data has been added as supplementary material.

- please give the values in SF1i as mg/dl as is done for the GTT in SF1h

Response: As proposed, we have given the values of the ITT in mg/dl. In order to corroborate insulin sensitivity, levels of insulin were also measured in each GTT time-point (**Fig. 1f and supplementary Fig. 1h**) and overnight fasting, in both the HFD-fed mice and in the *ob/ob* mice (**Supplementary Fig.1g and Fig.5e**). These experiments demonstrate that *Mat1* ASO treatment in HFD fed mice leads to improvement in insulin sensitivity and that results are highly reproducible.

- In Fig. 1g, I suggest to only show the ANCOVA graph and avoid the bar graph since it's meanwhile commonly accepted that energy expenditure can't be corrected by body weight. It would also be nice to see the resting metabolic rate.

Response: Thank you for your comments. We now present only the ANCOVA test in each energy expenditure study performed, including the ANCOVA obtained from the light phase metabolic rate (mice are less active) (**Fig. 1h, 1i**) and new ones obtained from new experiments (**Fig. 6e, Suppl. Fig. 4b, Suppl. Fig. 9d, e**).

It would be nice to measure body core body temperature of these mice during the oligo-induced weight loss. Can it be that these mice get fever and have therefore an increased energy expenditure and dramatic weight loss?

Response: As suggested by the reviewer we have now measured rectal and interscapular temperatures. The results show that while the rectal temperature maintains unaltered after targeting *Mat1a* in HFD-fed mice (**Supplementary Fig. 1e**), the interscapular temperature is clearly increased (**Fig. 2e**). Thus, taking all the results together (detailed and shown in the manuscript), we interpret that the body weight loss is due to increased BAT thermogenic activity that ultimately leads to higher energy expenditure. There are no signs of fever in these mice.

What about the increase in energy expenditure after cold exposure?

Response: We have measured energy expenditure at 4 °C for 300 minutes. The results (figure below) show that energy expenditure (Kcal/h) remained slightly higher in *Mat1a* ASO treated mice, whose body weight were markedly lower than the control ASO treated mice (30.6±4.1 vs 40.3±5.1). The significant differences in body weight make interpretation of these result challenging. In addition, it is important to highlight that during cold, shivering, where muscle function is also involved, is the expected major response to compensate from the increased heat loss following acute transfer from 22 °C to 4 °C and differences in this process might be found between the *Mat1a* ASO and control ASO treated mice. In fact, the amount of TG in muscle is also decreased after targeting *Mat1a* in HFD-fed mice (**Supplementary Fig. 7a**). The capacity of an animal for nonshivering thermogenesis, more specific of BAT thermogenesis, is more accurately measured under thermoneutrality, as this reviewer has further suggestes a comment below. We therefore, also analyzed energy expenditure under thermoneutrality, and found that energy expenditure remained higher in *Mat1a* ASO-HFD fed mice (**Fig. 1i**).

Thus, our overall interpetration of these results is that targeting liver *Mat1a* does not seem to be involved (at least in a relevant way) in body temperature defense in response to cold. This has not been included in the manuscript. However, the results obtained in thermoneutrality, reinforce our data indicating that *Mat1a* ASO triggers BAT thermogenesis and energy expenditure.

Rebuttal Figure. Energy expenditure during 5 hours of cold exposure in *Mat1a* ASO and Control ASO treated HFD-fed mice.

Also Seahorse measurement of oxygen consumption (at baseline and after treatment with a beta agonist) in isolated BAT primary cells would add value to the manuscript.

Response: We have performed these additional experiments: 1) the measurement of oxygen consumption in mitochondria extracted from freshly isolated BAT after HFD-fed mice were treated with the ASOs (**Fig. 2c**); 2) experiments of differentiation and induction of preadipocytes from the ASO treated HFD-fed mice (**Supplementary Fig. 4d, e; Supplementary Fig. 11c, d**). The main purpose of these BAT primary cell experiments was to determine if targeting liver *Mat1a* *in vivo* could lead to changes in preadipocytes, so that self-autonomous changes might be induced and maintained during differentiation and after induction in cell cultures. The BAT primary cell experiments are further described below.

The experiments of oxygen consumption rate have been performed in BAT-mitochondria since BAT thermogenesis increases as a consequence of inhibition of liver *Mat1a* (*Mat1a* expresses in the adult liver, it is not expressed in BAT; as **Supplementary Fig. 1c** shows). As anticipated, experiments showed that oxygen consumption was increased in BAT when HFD-fed mice were treated with the *Mat1a* ASO (**Fig. 2c**).

As suggested by the reviewer, we also attempted to measure oxygen consumption after beta-adrenergic stimulation *in vivo*. We were unable to detect any induction in OCR 30 minutes after the *in vivo* treatment (see below). We found that OCR was always higher when HFD-fed mice were treated with the *Mat1a* ASO but, the addition of beta-adrenergic stimulation *in vivo* did not increase OCR in either *Mat1a* or Control ASO treated mice. These results have not been included in the Revised manuscript.

Rebuttal Figure. Oxygen consumption rate (OCR) in mitochondria isolated from extracted BAT. ASO treated HFD-fed mice were sacrificed 30 min after the administration of the β 3 adrenergic agonist.

We next proceeded to analyze RNA expression from freshly isolated BAT after 30 min of *in vivo* beta-adrenergic stimulation and found increased expression of UCP1 and PGC1a mRNA in the BAT of *Mat1a* ASO treated HFD-fed mice (**Supplementary Fig. 4c**). Thus, we conclude that this *in vivo* stimulation has been enough to induce changes in the transcriptomic program but not enough to observe changes in the OCR.

I noticed that the sample sizes in Fig 1 range from N=5-8 in the Mat1a oligo treated group and from 4-7 in the control ASO groups. Since we have only one group of mice treated with the Mat1a oligos, I conclude that up to 3 out of 8 mice from this group were missing in some panels of the figure. It would be good to get an explanation here. Also the AUC data do not show individual data points as is required by the journal policies. Also, N=4 mice per group for the ASO treated controls seems very low and is likely underpowered.

Response: We apologize for not describing our experiments clearly enough. As described in our earlier responses, 310 mice were used in this project; from those 293 were treated with ASOs. However, only a representative study was presented to show the body weight loss. Multiple experiments were performed to complete the large number of assays required for this study. Some of these assays include: metabolic fluxes, metabolic cages, ITT and GTT, western blottings, qPCRs, ELISAs, histochemistries, *in vivo* lipoprotein metabolic assays with and without radioactive substrates, isolation of preadipocytes,

hepatocytes.... In addition, most of the studies of metabolic fluxes, qPCRs, Western blottings were performed using different tissues (liver, BAT and WAT); as such, in order to obtain sufficient tissues multiple set of experiments had to be performed, under various conditions and times (i.e., multiple sets of ASO-treated HFD-fed mice, under different experimental conditions (over several years), and each experiment performed using slightly different number of mice) . Experiments were also performed in other labs (CNIC or CIMUS). Nowe in each figure legend the n for each group of mice have been added. All these show that experiments are high reproducibility and results are reliable.

How does the weight loss induced by the Mat1a oligos reflect to body composition (fat and lean tissue mass)?

Response: The results show that there is a marked decrease in fat mass (58%) and a slight decrease in lean mass (16%) (**Fig. 1d**). We interpret the loss of lean mass as a consequence of the metabolic remodeling in the different tissues, there is a decrease in TG concentration in liver (**Fig. 3a**) or muscle (**Supplementary Fig. 7a**), which is link with the rewiring of methionine metabolism, specially relevant for muscle physiology. However, despite the decrease in lean mass, it is clear that the reduction of adiposity is the main feature explaining the weight loss experienced after the administration of Mat1a ASO.

The data showing resistance to DIO in the Mat1a KO mice are interesting. One aspect of this paper is that all data are based on a full global Mat1a ko (either full germline KO or the oligo-induced KO). Hence, it can't be ruled out that extrahepatic effects of Mat1a affect systems metabolism and energy expenditure. Since the authors particularly claim here that the liver-BAT axis is causal for the metabolic effects of Mat1a, a liver-specific KO of Mat1a would be appreciated to consolidate this hypothesis. If a floxed mouse is unavailable, then a faster approach would be to use an AAV-mediated approach to knock-down Mat1a in the liver.

Response: As mentioned above *Mat1a* gene is expressed mainly in the adult liver, and mostly in hepatocytes, and there is no *Mat1a* expression in either the BAT or WAT, as we have shown in the western blotting of the *Mat1a* product MATI/III (**Supplementary Fig. 1c and figure below**). So, in *Mat1a* KO mice or *Mat1a* ASO treated mice, *Mat1a* is silenced especially in liver. This is an important point in

our manuscript, since we found that when *Mat1a* was silenced, hepatocyte secretion of FGF21 increased (**Fig. 6b and Fig. 7a**), and was responsible for the resistance to developing obesity, insulin resistance and hepatosteatosis, and this was corroborated by *in vivo* experiments using the **liver specific FGF21-KO mice**. We propose that activation of BAT thermogenesis is a consequence of the liver (hepatocyte) secretion of FGF21. Given the specificity of *Mat1a* expression, we do not believe additional studies using AAVs would provide additional information.

Figure. MAT1/III levels in Brown adipose tissue (BAT) and White adipose tissue (WAT) of chow-diet (CD) and high fat diet (HFD)-fed mice. The positive control is a liver sample.

Note: We provide some links from the GTEx portal (<https://gtexportal.org/home/gene/MAT1A>) and the protein atlas (<https://www.proteinatlas.org/ENSG00000151224-MAT1A/tissue>), showing that *Mat1a* is expressed mainly in liver.

- Please show individual data points in all AUC panels.

Response: As proposed, we have added the data points in all AUC panels.

Fig. 4. The role of Mat1a1 on BAT function is interesting, but some additional data would help to strengthen the manuscript. It would be valuable to see energy expenditure increases after stimulation with a beta agonist, both in the Mat1a ASO mice but also using seahorse studies in isolated BAT primary cells. Fig. 4f shows the energy expenditure induced by a beta agonist corrected by body weight, which is not appropriate unless the mice do not differ in body weight? So these data need to be re-analyzed if the mice differ in body weight. Also, the increase in energy expenditure by the beta agonist is very low and

not higher as the first baseline data points. This is very untypical and indicates that this study didn't work well. I suggest to repeat this study.

Response: The Fig. 4f has been removed. For this Revised manuscript we have now determined the ANCOVA of energy expenditure vs body weight 30 min and 45 min after the *in vivo* beta adrenergic-stimulation of the ASO treated HFD-fed mice. We also measured expression levels of genes involved in thermogenesis *Ucp1* and *Pgc1a* in BAT after the *in vivo* beta adrenergic-stimulation, as described earlier. The results showed that energy expenditure, *Ucp1* and *Pgc1a* expression after the beta adrenergic-stimulation are always higher in the *Mat1a*-ASO HFD-fed mice than in the Control ASO treated mice (**Supplementary Fig. 4b and 4c**).

Do the BAT primary cells differ from the controls in adipocyte differentiation? Measurement of differentiation marker (Adiponectin, Pparg, FAS etc) during different time points of adipocyte differentiation would be valuable.

Response: For this Revised manuscript, we performed additional experiments using primary preadipocytes from ASO treated HFD-fed mice, as suggested by the reviewer. On the 8th day of differentiation (where >90% of the cells exhibit presence of lipid droplets), *PPARg* expression remained unaltered (**supplementary Fig. 4e**). This is an interesting observation since *Pparg* expression is increased in freshly isolated BAT. The results confirm our hypothesis that the observed changes in BAT are a consequence of the alterations in the liver, so when preadipocytes are isolated and maintained in culture the observed *in vivo* changes disappear.

Do Mat1a1 Ko mice show a difference in energy expenditure at cold exposure? Do Mat1a1 Ko mice show a difference in energy expenditure at thermoneutrality?

Response: All the new experiments have been performed in *Mat1a* ASO treated mice; one of the purposes of this study was to validate ASOs as a therapeutic option. We have used the *Mat1a*-KO in some (limited) experiments simply to corroborate our findings that targeting *Mat1a* in a different model, induces weight loss, improves insulin sensitivity, restores liver TG content and increases FAO in BAT in the context of obesity. With all due respect, we do not believe that additional experiments in

Mat1a-KO mice are required given the large number of new experiments already performed, and these include multiple sets of mice; some treated with ASOs (including long-term experiments), others with liver specific *Fgf21*-KO mice, and some where BAT UCP1 or β -Klotho have been knocked-down *in vivo*. We have also performed studies of ASO treated mice, where we evaluated energy expenditure after cold exposure and at thermoneutrality (as described above). In this measurement, we did not detect significant changes in energy expenditure (taking into account that cold is a condition where shivering thermogenesis is crucial). However, measurements at thermoneutrality have shown that *Mat1a* ASO increases energy expenditure, confirming its relevance in non-shivering thermogenesis.

Also, very interesting would be to metabolically phenotype UCP1 ko mice that have been treated with the Mat1a1 ASOs. Would the resistance to obesity and the thermogenic effect disappear in these mice? This would be very informative to consolidate the here proposed mechanism. This study also seems valuable since the UCP1 protein changes in BAT are only 0.5-fold increased (Fig. 4c), so also UCP1 independent effects can account for the energy expenditure phenotype of the Mat1a ASO mice.

Response: Thank you for this suggestion. Taking this proposal together with the one below considering the knockdown of β -Klotho, we have performed the following experiment in which 30 mice have been used: 10 mice were injected in BAT with lentivirus containing Sh-Scramble, another 10 mice were injected with those with the the Sh-*Ucp1* and another 10 with the sh- *β -klotho*. All the mice were fed a HFD and from the 6th week, following the same treatment as in the other experiments, 5 mice from each group received the *Mat1a* ASO and the other 5 the Control ASO (25 mg/kg/week). Body weight loss was measured along the experiments and energy expenditure was also measured by the end of the treatments (**Supplementary Fig. 9b, c and d**). In the manuscript, the energy expenditures are represented as ANCOVA. The data showed that the increased energy expenditure and the marked body weight loss observed in the *Mat1a* ASO treated HFD-fed mice, disappeared after knocking-down *Ucp1* or *β -Klotho*. Thus, UCP1 is mediating the increased thermogenesis and β -klotho is involved in the process. We really appreciate the suggestion of the reviewer since these new data consolidate the proposed mechanism.

Are levels of catecholamines changed in plasma and BAT of the *Mat1a* ASO mice?

Response: We have now measured epinephrine and norepinephrine in serum and in BAT of *Mat1a* ASO and control ASO treated HFD-fed mice. Targeting liver *Mat1a* did not lead to changes in serum catecholamine levels (**Fig. 2g and below**). However, the concentration of catecholamines per g of BAT was decreased when targeting liver *Mat1a*. Of **note:** the BAT levels of Norepinephrine is below the detection range of the ELISA kits, so this Norepinephrine result may not be reliable and may need to be validated in the future. Thus, the BAT levels of catecholamines have not been included in the manuscript, and we have simply added a sentence about the epinephrine levels.

Figure. Epinephrine and Norepinephrine in serum and in BAT of *Mat1a* ASO and Control ASO treated HFD-fed mice.

The molecular mechanism leading to the increased Ucp1 protein level can be strengthened. It would be nice to clarify whether this is a cell autonomous effect, hence demonstration of Ucp1 protein levels in isolated BAT primary cells (+/- treatment with a beta agonist) would be informative. Are key activators of Ucp1 (catecholamines) increased in the Ko mice? Are transcriptional regulator of UCP1 increased in the KO mice (p-p38 alpha and beta, other MAP Kinases, PKA, PGC1a, etc.)?

Response: As mentioned above, after the isolation, differentiation of preadipocytes was performed and *Pparg* expression was measured. In addition, another group of preadipocytes were differentiated and induced and expression levels of *Ucp1*, *Fgf21* and *Nrf2* were measured together with FGF21 secretion and the fatty acid oxidation rate. The results demonstrated that the changes observed in BAT were not detected after differentiation and induction of preadipocytes *in vitro*, which supports the idea that these effects are not cell autonomous (**Supplementary Fig. 4d, e and Supplementary Fig. 11a, c, d**).

As mentioned above, serum catecholamine levels were not altered in *Mat1a* ASO treated HFD-fed mice or in the liver specific *Fgf21*-KO or in their corresponding controls (**Supplementary Fig. 8e**). On the other hand, signaling pathways involved in UCP1 regulation (p38, PKA and S6 signaling) as well as expression levels of genes involved in thermogenesis, differentiation and lipid metabolism were measured. The results showed that targeting liver *Mat1a* led to activation of the mTOR pathway in BAT, as demonstrated by pS6 levels in *Mat1a* ASO treated HFD-fed mice (**Fig. 2f**). No changes were observed in PKA or p38 signaling. The results also showed that the mTOR activation did not occur when *Fgf21* was knocked-down in liver, a condition linked to low levels of serum FGF21 (**Fig. 6d**). Thus, our results suggest that a major FGF21- regulated signaling node in BAT is mTORC1/S6K, as was previously reported (Minard *et al.* Cell Reports 2016 - they had showed that UCP1 and FGF21 induction, increased adiponectin secretion, and enhanced glucose uptake were mediated through mTORC1). Our results suggest that the increased thermogenesis and the beneficial effect of targeting liver *Mat1a* could be mediated through activation of the mTORC1/S6K in BAT. Some sentences about this have been included in the discussion section of the manuscript.

Figure 5a. Please show body weight data in gram, not only as change in gram. The body weight change in the ob/ob mice is likewise very dramatic despite unchanged food intake. Since the authors attribute

this to changes in energy expenditure, it has to be ruled out that differences in food efficiency and metabolizable energy exist. So bomb calorimetry with assessment of assimilated/metabolized energy are required to truly attribute these changes to energy expenditure. Also, please show individual data points for the AUC and show the ITT data as mg/dl as has been done for the GTT in Fig 5d.

Response: As suggested, we have now added the body weight data in grams, not just the body weight changes. Additional *ob/ob* mice were purchased for the calorimetric studies. However, when when we treated this second set of *ob/ob* mice with ASO, we observed that body weight change occurred later than in the first set of *ob/ob* experiments (shown in the manuscript, **Fig. 5a**). It is possible that the housing conditions may be different and could lead to these temporal differences in weight loss. The calorimetric study showed that energy expenditure (**Figure below, panel B**) were similar to what we had observed in other sets of experiments; however, the heterogeneity and the diversity in the locomotor activity precluded definitive conclusions. It is very likely that increasing the number of mice per group would lead to significant statistical changes. However, given the extremely high price of these mice we were reluctant to order a new set of *ob/ob* mice. We hope that the reviewer understands that doing experiments with *ob/ob* are difficult to afford. These studies have not been included in the manuscript. Levels of UCP1 in BAT and expression of genes involved in thermogenesis, differentiation and lipid metabolism however, showed that targeting liver *Mat1a* in *ob/ob* mice results in changes in BAT transcriptomic signature that relatesto increased thermogenesis (**Fig. 5**).

A

B

Rebuttal Figure. HFD-fed *ob/ob* mice received four injections of ASOs (25 mg/kg/week) (n=5 for each group). A. Body weight and body weight change in *ob/ob* mice. B. Energy expenditure vs body weight (ANCOVA) and locomotor activity.

In Fig 6, the authors show increased FGF21 levels in response to treatment with Mat1a ASO. Since FGF21 is secreted under conditions of fasting/weight loss/increased fatty acid oxidation, it seems fair to ask whether this increase in FGF21 is causal to Mat1a deficiency or rather a consequence of the increased lipid oxidation and weight loss? So a calorie restriction study comparing Mat1a ASO mice with mice that were weight-matched to the Mat1a ASO mice would give valuable information as to whether FGF21 is regulated by Mat1a or rather the weight loss.

Response: Thank you for this suggestion. We do not believe that a calorie restriction study is necessary since *Mat1a* ASOs in chow diet (CD) fed mice did not lead to any weight loss or changes in the fatty acid oxidation (FAO) rate (Figure below). We have performed calorimetric studies in these mice and have measured FGF21 in serum. The results showed that targeting liver *Mat1a* in CD-fed mice resulted in an

increase in circulating FGF21 levels (**Figure below**), which is lower to levels observed in HFD-fed mice (**Fig. 6a**). This increase is not linked to changes in liver FAO (which is also unaltered when targeting *Mat1a* in HFD-fed mice; **Fig. 3b**) or with body weight changes. These results suggest that the increase in serum FGF21 was likely a consequence of targeting liver *Mat1a*. We have not included these results in the manuscript, we have included this information in the discussion section of the manuscript.

Rebuttal Figure. Chow-diet (CD) fed mice mice received four injections of ASOs (25 mg/kg/week). A. Body weight and food intake. B. ANCOVA of Energy expenditure vs Body weight. C. Liver fatty acid oxidation rate. D. Serum FGF21 levels.

In Figure 6, the authors show that Mat1a ASO fails to affect body weight in liver-specific FGF21 ko mice. Since the key focus of this manuscript is that Mat1a KO increases energy expenditure via the liver FGF21-BAT axis, the reviewer is surprised that the authors did not measure energy expenditure in the Mat1a ASO treated liver-specific FGF21 ko mice. This seems crucial to consolidate their key working hypothesis.

How is energy expenditure in the Mat1a ASO treated liver-specific FGF21 ko mice? What about protein levels of UCP1, PGC1a, p38, PKA etc? Are levels of catecholamines different between wt and the Mat1a ASO treated liver-specific FGF21 ko mice? None of this has been done but still the authors claim the FGF21-BAT axis as the key signaling mechanism leading to increased BAT thermogenesis. This has to be significantly strengthened. The ultimate experiment would be to knock-out beta Klotho in the BAT to see whether treatment with Mat1a ASO fails to affect body weight and energy expenditure in these mice. Unfortunately, in the current version, the Mat1a - FGF21-BAT link is not sufficiently addressed experimentally.

Response: Thank you for the comments and suggestions. We have performed additional experiments and have analyzed their energy expenditure and examined the activation status of the mTORC1/S6K signaling pathway. Levels of UCP1 were also measured (**Fig. 6g**). The results showed that loss of liver derived FGF21 (in liver specific *Fgf21*-KO) abrogated the increased UCP1, pS6 and thermogenesis observed when targeting *Mat1a* in HFD-fed mice. Furthermore, there was no demonstrable changes in catecholamine levels when mice were treated with the *Mat1a* ASO, and consistent with our prior findings (**Supplementary Fig. 8e**). These results confirm the link between liver *Mat1a*-FGF21 and BAT thermogenesis.

Fig. 7a and e lack data on NRF2 outside the nucleus. IP-studies to analyze whether FGF21 can bind NRF2 would help to better understand the NRF2-FGF21 link. These studies should be done in liver and BAT.

Response: In this Revised manuscript, we have measured NRF2 in liver and in liver cytoplasm (**Fig. 7b**). We have performed ChIP analysis in hepatocytes isolated from ASO treated HFD-fed mice. The results showed that targeting *Mat1a* led to NRF2 binding to a consensus site on the *Fgf21* promoter (**Fig. 7d**). We also measured expression of *Nrf2* and *Fgf21* in BAT in ASO treated HFD-fed mice, and found that targeting liver *Mat1a* also induced the expression of *Nrf2* and *Fgf21* in BAT (**Supplementary Fig. 11a**). Finally, expression levels of *Nrf2* and *Fgf21* were measured in ASO treated HFD-fed liver specific *Fgf21*-KO mice (**Supplementary Fig. 11b**), and the results showed that the increased expression in BAT were lost when liver FGF21 was knocked-out (**Supplementary Fig. 11b**). Furthermore, expression levels

of *Nrf2* and *Fgf21* were also measured in BAT primary adipocytes isolated from ASO treated HFD-fed mice, and the results showed that there was no increased expression (**Supplementary Fig. 11c**).

Reviewer #2 (Reviewer Comments to the Author):

The Introduction describes an association between high methionine intake and obesity, and dietary methionine restriction and reduced adiposity. Previous studies have shown that knocking out MAT1A also produces hypermethionemia and results in spontaneous development of fatty liver disease. What was the rationale for proposing that knocking down MAT1A with oligonucleotides would result in a phenotype that was opposite from the MAT1A knockout animals with respect to hepatic lipid metabolism?

Response: As this reviewer indicated *Mat1a* knockout induces hypermethionemia, and this is due to the lack of usage of methionine in liver. The first reaction of the methionine cycle is the conversion of methionine into SAMe and it is catalyzed by MAT1/III, the product of *Mat1a* gene. So a knock out in the *Mat1a* gene leads to the decreased usage of methionine, which accumulates in liver and consequently, in serum. Given that the dietary methionine restriction leads to the reversion of obesity, insulin resistance and fatty liver, we wanted to know if the decreased usage of methionine in liver could lead to the same phenotype. In addition, it has been demonstrated in other mouse models that the knockdown of enzymes of the methionine cycle (e.g. NNMT) induce resistance to diet induced obesity, the associated insulin resistance and hepatosteatosis. On the other hand, it has been reported that the dietary methionine restriction leads to the increase in serum FGF21. Thus, given the known beneficial effect of FGF21 in nonalcoholic fatty liver disease, we wanted to know if in the context of obesity, targeting *Mat1a* could also provide those beneficial effects. The results here demonstrated, as anticipated, that when *Mat1a*-KO mice were fed a HFD, there was a resistance to body weight gain, to develop insulin resistance, a high increase in serum FGF21 levels, an increase in FAO in BAT and resistance to develop fatty liver. Thus, the results obtained here using the *Mat1a*-KO mice are similar to those obtained when *Mat1a*-ASOs were used. The conclusion is that resistance to weight gain and to insulin resistance, is driven by hepatocyte secretion of FGF21, which protects the liver from nonalcoholic fatty liver disease. It is widely recognized that weight loss is the cornerstone of effective NAFLD treatment and it has been recently published that bariatric surgery reduces cancer risk in adults with

NAFLD and severe obesity (Vinod K. Rustgi et al, Gastroenterology 2021); hence the rationale for such a study.

Near the end of the Introduction, the authors state that chronic changes in liver SAMe levels in lean mice have been associated with the onset and progression of NAFLD with age, but isn't that due to a decrease in MAT1A/MAT2A,2B expression ratios that decrease the production of SAMe? Why wouldn't knocking down MAT1A produce the same changes in SAMe (and serum methionine) and if those metabolites are causative to the onset and progression of NAFLD, why would those changes produce opposite metabolic effects in the current model?

Response: As mentioned above, the hypothesis was that the decreased usage of methionine due to the lack of *Mat1a* could be linked, as demonstrated in this study, to increased secretion of FGF21 by hepatocytes, and that this increased secretion will lead to increased plasma levels and all the beneficial effects that are associated, including protection from hepatosteatosis. As mentioned above, weight loss is associated with regression of hepatic steatosis, and we have shown in this report that the activation of BAT by FGF21 is the putative mechanism. These results overall, suggest that activation of BAT even under certain non-favourable metabolic conditions, may prevent development and progression of chronic liver disease.

The authors have called ex vivo synthesis of fatty acids in pieces of liver de novo lipogenesis while the term is normally reserved for in vivo measurement of fatty acid synthesis. This needs to be clarified for the reader.

Response: The *in vivo* fatty acid synthesis is performed when the radioactive substrates are added intravenously to the mice. Slices from freshly isolated liver were also incubated with the ³H-acetate; thus, this is an *ex vivo* assay. These have all been fully described in the Methods section and the *ex vivo* term has been removed.

In the Results, the authors have interpreted their triolein uptake data to mean that the substrate fueling an increased thermogenic rate is coming from outside brown adipose tissue. The results are convincing

that triolein uptake is increased in BAT but what is the evidence that it is being oxidized? Significant previous work in BAT has shown that DNL and fat oxidation are coupled during cold-induced increases in BAT thermogenesis. This work suggests that the fatty acids being oxidized are being synthesized in BAT. These conclusions are supported by work showing that genetic or pharmacological impairment of DNL or oxidation compromises cold-induced thermogenesis.

Response: Thank you for the interesting comments. The results demonstrate that lipids in BAT are being oxidized and catabolized, no matter the source. We measured ACC, p-ACC and FAS protein levels, which are the main enzymes involved in *de novo* lipogenesis (**Supplementary Fig. 5b**). The expression of *Acaca*, *Acacb* and *Fasn* mRNA were also measured (**Supplementary Fig. 5a**). The results showed a decrease in the protein levels of ACC, p-ACC and FAS and decreased expression of *Acaca* suggesting that the *de novo* lipogenesis was decreased.

The authors evidence of increased thermogenesis comes from Fig. 4F where they show that a beta3-AR agonist increases O2 consumption more in Mat1a ASO mice than Control ASO mice. However, these results are compromised by the problem of not describing how this experimental protocol was conducted and how the data were analyzed. How was the beta3-AR agonist given to the mice while they were in the calorimeters? The results in Fig. 4F show no break in the collection of data associated with this event. More troubling is how the VO2 consumption data are being scaled. The ordinate is labeled ml/kg/h, indicating that the O2 consumption data is being scaled by BWs, which are ~15g lower in the Mat1a ASO group. As described in multiple manuscripts describing the correct approach to analyzing IDC data in recent years, big differences in BW confound group comparisons by over-amplifying the calculated EE in smaller animals within the groups.

Response: Thank you for your comments. We agree, this reviewer is correct, and we have now removed Fig. 4F in the Revised manuscript. This reviewer is also correct in pointing out about how we should be representing energy expenditure. There is certainly debate how we should be interpreting energy expenditures in mice with varying body weights. The ANCOVA analysis of the energy expenditure vs the body weight is widely accepted (for guidance we followed: Tschop M et al. A guide to analysis of mouse energy metabolism. *Nat Methods*. 2011 Dec 28;9(1):57-63). Therefore, we have performed this analysis

(also following the recommendation of reviewer 1) for all the measurements of energy expenditure in the Revised manuscript. To study the effects of β -adrenergic agonist, we performed the ANCOVA analysis at 30 and 45 mins after the administration of the drug. We also measured *Ucp1* and *Pgc1a* expression in BAT after the after the β -adrenergic stimulation (**Supplementary Fig. 4c**).

The data presented in Fig. 7 provide support for the idea that knocking down Mat1a in liver activates FGF21 release from the liver through a NRF2-dependent mechanism. The authors propose that this occurs after nuclear localization of NRF2 and transcriptional activation of FGF21. If the authors are able to detect nuclear localization of NRF2 with available reagents, they should confirm the direct role of NRF2 using ChIP assays to show that NRF2 is in fact being recruited to the FGF21 promoter. If NRF2 is necessary and sufficient for transcriptional activation of FGF21 after Mat1a ASO treatment, they should also show that hepatic deletion of NRF2 blocks the induction of FGF21. In addition, more information is needed about the NRF2 inhibitor used in Fig. 7D. Its source needs to be identified, along with a reference to a publication demonstrating its efficacy and specificity.

Response: Thank you for this interesting suggestion. We have performed the ChIP assay in hepatocytes isolated from ASO treated HFD-fed mice. The results confirmed the binding of NRF2 in the *Fgf21* promoter (**Fig. 7d**). We also performed an *in vitro* experiment in which *Nrf2* was silenced by siRNAs, and found that silencing *Nrf2* abolished the increased FGF21 secretion that occurred when targeting liver *Mat1a* (**Supplementary Fig. 10b**). A similar result was observed when ML385 (a specific NRF2 inhibitor) was used (**Fig. 7a**), and we have now included information on this inhibitor in the Methods section. In brief, the NRF2 inhibitor ML385 is a probe molecule that binds to NRF2 and inhibits its downstream target gene expression. Specifically, ML385 binds to Neh1, the Cap 'N' Collar Basic Leucine Zipper (CNC-bZIP) domain of NRF2, and interferes with the binding of the V-Maf Avian Musculoaponeurotic Fibrosarcoma Oncogene Homolog G (MAFG)-NRF2 protein complex to regulatory DNA binding sequences. ML385 is a novel and specific NRF2 inhibitor (Singh A et al, ACS Chem Biol. 2016)

Specific Points

1. *The description of the methods used to conclude that EE was affected by the treatments is inadequate. It refers to a previous publication but when that publication is examined it provides no details on how the calorimetry data were analyzed. The problem must be rectified in the current paper. Fig. 1G is used to make a claim that EE is increased by the MAT1A ASO but the data in the figure shows that EE was scaled by kg but does not define kg of what. Moreover, the accompanying figure panel indicates that there is no relationship between EE and BW over a weight range of 24-29 grams. This seems improbable since EE should increase roughly in proportion to BW raised to some power < 1 . An additional problem is that Fig. 1C shows that the two groups of mice being compared in Fig. 1G have approximately the same spread in BWs of mice within each group while Fig. 1C shows that the mice in the two groups were ~ 15 g different in BW. Together, this creates a lot of confusion for the reader, even for one experienced in the analysis of indirect calorimetry data. Please see <https://www.nature.com/articles/s41592-019-0513-9> for guidance in analyzing and presenting EE data.*

Response: We have removed this publication and we have added another one. On the other hand, as mentioned above, taking into account the reviewer 1, this reviewer suggestions and the published information (Tschop M et al. A guide to analysis of mouse energy metabolism. *Nat Methods*. 2011 Dec 28;9(1):57-63), we agree that the ANCOVA analysis should be used for comparison of energy expenditure when body mass or composition differs between groups. Thus, ANCOVA has been used for all these analyses, including former Fig. 1g, **now Fig. 1h**.

2. *OGTTs measure a composite of glucose absorption rates, alterations in B-cell sensitivity to glucose, and insulin-dependent glucose excursions. Given the relatively small improvement in insulin tolerance and only at the last time point, the substantial improvement in glucose tolerance shown in Fig 1E points to changes in the first two factors rather than solely insulin sensitivity. <https://pubmed.ncbi.nlm.nih.gov/18670420/> The 60 min time point shouldn't even be considered as a marker of insulin action. If you don't see an improvement in the first 20 min, it's probably not increased insulin action. What's more, the substantial improvement in glucose tolerance after Mat1a ASO is likely secondary to the loss in BW.*

Response: Thank you for your comments. For this Revised manuscript, we have added New GTTs, ITTs and measured serum insulin levels in various time points (**Fig. 1f and supplementary Fig. 1h**). The levels of glucose in the ITTs have been represented as mg/dl, as recommended by the first reviewer; fasting insulin levels have also been quantified in the DIO induced mice (time 0 of the insulin levels of the GTT time points) (**Fig. 1f and supplementary Fig. 1g**) and in the *ob/ob* mice (**Fig. 5e**). In addition, the GTT, ITTs and measurements of insulin released during the GTTs have been performed at two different situations in the HFD-fed mice; 1) after the first injection of ASOs, when body weight loss is not that evident (**supplementary Fig. 1h**), and 2) by the end of the treatments (**Fig. 1f**). The results showed that after the first dose of ASOs (7th week of HFD), insulin levels were already lower before the bolus of glucose in the GTT (time 0), and after 30 min of the glucose administration. The results showed that the insulin released in each time point and the levels during fasting were markedly lower when targeting *Mat1a*. Fasting levels of insulin were also decreased when *Mat1a* was targeted in the *ob/ob* mice. These collective results show that targeting liver *Mat1a* protects against insulin resistance.

3. *What were the fasting insulin concentrations in the two groups of mice?*

Response: Fasting insulin concentrations were measured as described above, under various conditions.

4. *From looking at supplementary Fig. 1F, there appears to be a problem in the way the MAT1A KO mice responded to the HF diet. After it was introduced, the mice lost ~5 g in the first week and then stabilized their weight over the following 8 wks while the WT mice gained over 20 g. It would help the reader to understand the data better if the authors would present the actual BWs of the mice in the two groups and the actual mean food intakes for the two groups over time. Presenting the data as cumulative food intake makes it more difficult to see what is happening over time to food intake. This suggestion also applies to the MAT1A ASO mice, where it would be better to see the actual BWs of the mice in the two groups and their week by week food consumption rates. Looking at supplemental Figs 1f and 1g, it is hard to believe that mice differing in BW by 25 g are eating exactly the same amount of food. Lastly the actual BWs of the animals in each group is important to know relative to the analysis of the indirect calorimetry data. The authors should add another figure panel and show the actual BWs of the mice in each group.*

Response: As the reviewer suggests we have included the BWs of all animal models (sometimes as supplementary figures due to the lack of space) and the food intake for all the different groups. In all these models, the results show always unchanged food intake. The energy expenditures are now all represented vs the body weight with the ANCOVA test performed so that the two parameters can be easily related. In all animal models tested, the inhibition of *Mat1a* was associated to higher energy expenditure. The fact that *Mat1a* ASO mice show a lower weight without changes in food intake is not that surprising, since many previous reports have found other animal models with a marked decrease in body weight in a feeding-independent manner. Just to mention some previously published examples from coauthors of the manuscript (Folgueira C et al, Nat Metab. 2019) or from other groups (Pirzgalska RM et al, Nat Med. 2017).

On the other hand, we had the same thought when we showed this for the first time, so before sending the first version of the manuscript we performed again the analysis of the response of *Mat1a*-KO mice to the HFD; for this, we fed *Mat1a*-KO mice the HFD for a week and we obtained the same result. These results are shown in the figure below. Although we don't have a clear explanation for this, it seems that *Mat1a* KO mice are somehow protected against HFD-induced early metabolic dysfunctions. Indeed, this interesting question deserves to be investigated in a further study.

Figure: Body weight change and body weight of *Mat1a*-KO (n=3) and WT mice (n=5) fed a HFD for 7 days.

5. The variability of the Sirius Red and F4/80 measures is concerning, indicating that this approach to detecting fibrosis and inflammation is insensitive to the point of being useless.

Response: It is true that there is variability in the results from the histochemistry procedures; however, do not believe that histochemistries are 'useless'. We have now also analyzed the expression of genes involved in fibrosis and inflammation (**Supplementary Fig. 6a**). The heterogeneity of the results is also high but there were no statistical differences. We have performed the analysis also when using the *Mat1a* ASO2 (**Supplementary Fig. 6c**). The studies show that targeting *Mat1a* does not lead to any increase in the expression of genes involved in fibrosis or inflammation.

6. In Fig. 3D, triolein uptake is expressed per g of tissue. What do the results look like when corrected for differences in tissue weight between treatment groups?

Response: In the Figure below, we show the uptake of labelled lipids (DPM per total tissue or per mg of protein). The results confirm that targeting liver *Mat1a* leads to an increase in the uptake of dietary lipids in BAT.

Figure. Uptake of dietary lipids. A. uptake per tissue; B. uptake per mg of tissue protein. Abrev: C ASO: Control ASO, M ASO: *Mat1a* ASO.

7. *There is a problem with the interpretation of the fatty acid oxidation data in the sense that the assays do not measure in situ rates of fatty acid oxidation. With a fixed amount of C14-palmitic acid present in the in vitro incubations, the assays are providing an estimate of Vmax or capacity of the tissues from the two groups to oxidize fatty acids. This is not a measure of the rate yet it is how the figures summarizing this data is labeled.*

Response: We have performed the fatty acid oxidation analysis *ex vivo*, not *in vitro*, as previously reported and described in other published manuscripts (e.g. Hirschey MD et al, Nature 2010). The [1-¹⁴C]-palmitic acid that is added to the samples is always 0.5 µCi/ml, but this corresponds to a very low quantity of palmitic acid. Thus, we believe the reviewer is probably referring to the carrier that is usually added in the reaction, which is the cold palmitate (500 µM in the assay that has been used in this study), usually added to improve the signal and to promote the uptake of the radioactive substrate, given that the quantity of [1-¹⁴C]-palmitic acid is so low. In this assay, the flux or the rate of fatty acid oxidation has been measured as reported in multiple publications (e.g., Hirschey MD et al, Nature 2010; Frank K. Huynh et al. Methods Enzymol. 2014; Gao X et al. Biochim Biophys Acta. 2015; Lucía Barbier-Torres et al, Nat Comm. 2020; Gonzalez-Romero F, Mestre D et al, Cancer Res 2021). In addition, in our mice models, as frequently happens in other projects with different treatments or diseases, the amount of fatty acids that reach the mitochondria varies, so in order to determine the FAO rate, one should know the mitochondria capacity to catabolize those lipids. For this, an external amount of fatty acids should always be added in the same concentration in all the groups being analyzed. Of relevance in this study, is the FAO rate analysis in these conditions taking into account that the lipid that reaches the BAT mitochondria is higher when targeting liver *Mat1a*; however, the amount of lipid in BAT is lower than when mice are treated with the Control ASO. To get an accurate result of the flux or the rate of FAO, the researchers need to add the same amount of external fatty acid in the two groups that are being analyzed. This is something widely accepted to measure the FAO rate as reported in the manuscripts above. If the reviewer does not feel comfortable with the terminology FAO rate (which is because the units are always per hour), we will remove rate when describing the results.

8. *It's unclear what the isoproterenol-dependent increase in ex vivo lipolysis adds to the story and why the responses from the Control ASO CD mice were not included in Fig. 4d. Lipolytic responsiveness to*

beta-adrenergic agonists is impacted by a number of factors and what the authors are showing is probably mostly dependent on the decrease in adiposity and fat cell size in the Mat1a ASO group.

Response: The assay of lipolysis is frequently performed in the presence and/or absence of the β -adrenergic receptors agonist isoproterenol as has been described in other studies (Bradlee L. Heckmann et al, J Biol Chem., 2014; Gao X et al. Biochim Biophys Acta. 2015), so that the capacity of that tissue under basal and stimulated conditions are measured.

In this Revised manuscript, we have added the control ASO CD mice and have calculated and represented the lipolysis results in nmol of fatty acid or glycerol released per mg of tissue protein; these will provide clarity on the influence of fat cell size or adiposity (Supplementary Fig. 3c).

9. Which WAT depot was used to produce the data presented in Fig 4? It doesn't seem to be specified in either the Methods or the fig legend. This is important because the capacity of WAT to undergo remodeling is depot specific.

Response: As has been frequently used for this kind of analysis, we have used epididymal WAT. This has been added to the methodology section.

10. As with Fig 1, please present the actual BW means of the mice in each group, as well as their actual food intakes each week.

Response: As mentioned above this have been performed.

11. Ob/ob mice typically have high fasting insulin levels by 12-15 wks of age. What impact did the 5-6 g difference in BW produced by the Mat1a ASA have on fasting insulin levels?

Response: We thank the reviewer for this suggestion. Now, we have been performed this analysis, levels of fasting insulin are markedly lower in *Mat1a* ASO than in the Control ASO *ob/ob* treated mice. In the new version of the manuscript has been added in **Fig. 5e**.

12. *The problem noted above for FA oxidation measurements is also applicable in Fig. 5F. The authors are measuring the amount of palmitate oxidation with a fixed amount of palmitate and concluding that the measurement is indicative of in vivo fatty acid oxidation in BAT. The data do not fully support this conclusion.*

Response: As mentioned above, the FA oxidation rate has been measured *ex vivo*. We did not perform *in vivo* fatty acid oxidation; this must be a misunderstanding. Our results show that FAO rate is increased in BAT when targeting liver *Mat1a*. FAO rate (nmol of ASM or CO₂ generated from palmitate per g of tissue and per hour) has been measured as described above. This assay measures the whole FAO rate and not just a measure of an enzyme activity. This method is well established in our lab (Lucía Barbier-Torres et al, Nat Comm. 2020; Gonzalez-Romero F, Mestre D et al, Cancer Res 2021) and is a well accepted assay (Hirschey MD et al, Nature 2010; Frank K. Huynh et al. Methods Enzymol. 2014; Gao X et al. Biochim Biophys Acta. 2015).

13. *The authors measurement of PPAR α expression in response to Mat1a treatment is not an effective approach to determining whether the increase in serum FGF21 is due to PPAR α activation. The same criticism should be considered for measurement of nuclear ATF4 by Western, particularly since it is widely known in the field that reliable measurement of endogenous ATF4 by Western is difficult to establish. The data in supplementary Fig 5 certainly do not meet the standard for ruling out involvement of either transcription factor in Mat1a ASO-dependent induction of hepatic FGF21.*

Response: We also performed experiments with an inhibitor of PPAR α or ER stress. In this Revised Manuscript, additional experiments where isolated hepatocytes from ASO treated HFD-fed mice were exposed to siRNAs to silence *Ppara* and *Atf4* were performed (**Supplementary Fig. 10a**). Similar to prior experiments using PPAR α and of ER stress inhibitor, these new experiments showed that *Ppara* and *Atf4* are not involved in the increased FGF21 secretion by hepatocytes (**Supplementary Fig. 10b**). We also measured levels of PPAR α and ATF4 in the nucleus, and found that targeting *Mat1a* did not lead to changes in their levels of expression (**Supplementary Fig. 10c**). In concert, both these studies confirm that neither PPAR α nor ATF4 are involved in the increased secretion of FGF21.

14. In the last paragraph of the Results, the authors examine FGF21 in the media of hepatocytes from mice after knocking down *Mat1a* and adding SAME to the media. How much SAME was being added to the media and what proportion of the amount being added is ending up inside the cells? There seems to be an assumption that SAME is crossing the cell membrane. What is the evidence for that occurring?

Response: This is a very interesting point. We have now added 6mM of SAME to the media and have measured expression levels of *Bhmt*, *Sahh* and *Cbs* mRNA, genes involved in the methionine cycle (4 and 24 hours after adding SAME) and have also quantified SUMO1 and SUMO1-conjugated proteins (24 hours after adding SAME) (**Supplementary Fig. 11b**) (since SAME is a SUMOylation inhibitor; Pui Y. Lee-Law et al, J Hepatol 2020).

Additional Points

1. It would have been helpful if page numbers were provided in the MS.

Response: Page numbers are now provided.

2. Please define which oligos were used in supplementary table 1. Is ASO the control oligo and ASO2 the antisense oligo? In addition, measures of reproducibility must be provided.

Response: The ASOs sequence have now been provided in the methodology section and here:

Control ASO: 5'-CCTCCCTGAAGGTTCTCC-3'

Mat1a ASO: 5'-CCACTTGTCATCACTCTGGT-3'

Mat1a ASO2: 5'-GCTCAGGAGACATTGACCAT-3'

IONIS Pharmaceuticals uses the same Control ASO. For this study, to knock-down *Mat1a* they created two different ASOs, which we have labelled as *Mat1a* ASO and *Mat1a* ASO2. In our preliminary studies, we found that the ASO dose required to silence 90% *Mat1a* was 25 mg/kg/week for *Mat1a* ASO and 50 mg/kg/week of *Mat1a* ASO2. The majority of studies were performed with the *Mat1a* ASO. *Mat1a* ASO2 were used in some experiments to confirm changes.

3. Please specify the temperature at which the mice were housed.

Response: Mice were housed at 21-22 °C, this has been added now in the methodology section.

Reviewer #3 (Remarks to the Author):

Saenz de Urturi and colleagues present an interesting set of findings in which deletion of MAT1, the enzyme that catalyzes the first step in the methionine cycle, alters energy balance and prevents obesity of dietary and genetic models of obesity. This was associated with reductions in hepatic steatosis and improvements in glucose tolerance. The energetic phenotype was attributed to increased levels of FGF21 which in turn, the investigators posit, are due to activation of NRF1. The experiments were well designed and the use of two separate ASO's against MAT1 demonstrate experimental rigor. The manuscript was also well written, though there are sections that could have benefited from additional editing for clarity in English. The overall conclusions are interesting and important though there are additional considerations that should be addressed:

Response: Thank for your kind words.

1. Perhaps the most intriguing finding is the enhancement of BAT function in MAT1 ASO treated mice. The authors point to increased fatty acid uptake after an oral challenge of lipid, increased UCP1 expression, oxygen consumption and lipolysis that seem to depend on FGF21. But, several questions remain. First, what about BAT mass in MAT1 ASO treated animals? Is this different? Though FA uptake was not altered in WAT, was there any evidence of browning in WAT? Was there an alteration in RQ?

Response: The BAT mass was lower when targeting liver *Mat1a* (0.09 ± 0.019 g in *Mat1a* ASO treated HFD-fed mice; 0.31 ± 0.09 g in Control ASO treated HFD fed mice and 0.11 ± 0.021 in Control ASO chow diet-fed mice). Thus, there is a decrease in the BAT size resulting in the CD-fed mice mass. However, as we have shown to Reviewer 2 and as shown below, fatty acid uptake maintains increased in BAT when calculating the results per whole tissue or per mg of protein.

Figure. Uptake of dietary lipids. A. uptake per tissue; B. uptake per mg of tissue protein. Abrev: C ASO: Control ASO, M ASO: *Mat1a* ASO.

Western blotting of UCP1, pS6 and S6, were performed in WAT, and similar to what was found in BAT, WAT protein levels of UCP1 and pS6 increased in some mice with the expression of genes involved in thermogenesis (**Supplementary Fig. 3**). The results show that there is a tendency to increased markers of browning, but the results are not as consistent as in BAT.

Finally, the results did not show any changes in the respiratory quotient of *Mat1a* ASO treated mice (**Supplementary Fig. 1i**).

2. The authors hypothesize that there is a “channeling” of lipids to BAT of *MAT1* ASO treated animals? How is this channeling occurring? Is there an increased expression of LPL in BAT?

Response: This is an interesting question. We measured protein levels of LPL and CD36 in BAT (**Fig. 4e**), but did not detect any changes. As the activation of LPL depends in part, on the amount of the activator apoC2 and the inhibitor apoC3, levels of these apolipoproteins were measured in serum (**Fig. 4f**) and their respective gene expression in liver (**Fig. 4g**). Results showed that targeting *Mat1a* increased levels of apoC2 in liver and serum but levels of apoC3 were decreased (**Fig. 4f, g**). We also measured gene expression of *apoC2* and *apoC3* in liver specific *Fgf21*-KO (**Supplementary Fig. 8f**), and found that loss of liver-specific FGF21 reversed the gene expression changes that occurred when targeting liver *Mat1a*. These results suggest that the activity of LPL is increased by FGF21-driven apoC2 and 3 changes.

3. How do the authors reconcile the finding of decreased DNL (using an ex vivo assay) and decreased expression of key lipogenic enzymes with the absence of any difference in absolute VLDL secretion rates? Also, since ACC2 expression may regulate rates of lipid oxidation, mRNA expression of ACC1 and ACC2 should be assessed to determine if the main difference is in ACC1.

Response: This is a challenging question. The contribution to VLDL-TG from each source (DNL, dietary lipids and serum NEFA) is diverse, and is dependent on the metabolic status of the liver. It has been reported that in NAFLD patients, with high metabolic dysregulation, both elevated peripheral fatty acid flux and DNL contribute to hepatic and lipoprotein fat and that this fatty acid usage by the liver fluctuates from fasting to feeding (Donnelly KL et al, J Clin Invest, 2005). It has been previously reported by our group (Cano A et al, Hepatology 2011), that *Mat1* deficiency induces an increase in apoB secretion; in a metabolic context with excess apoB and increased adipose tissue lipolysis, the source of VLDL-TG could be enriched in peripheral fatty acids more than in those synthesized de novo. In this study, we have measured levels of *Acaca* and *Acacb* mRNA, but we did not detect any changes in gene expression with *Mat1a* ASO treatment (Figure below), suggesting that the downregulation of protein levels are likely related to protein stability. We have inserted a sentence on this in the discussion section.

Figure. mRNA expression of *Acaca* and *Acacb* in liver of *Mat1a* ASO and Control ASO treated HFD-fed mice.

4. Though there are impressive differences in liver TG and improvements in glucose tolerance, there may also be changes in muscle TG concentration. A reduction in muscle TG would better account for the improvements in glucose tolerance. It would be helpful to report plasma insulin concentration under fasting and during the glucose tolerance tests to better assess the changes in glucose tolerance.

Response: As suggested by the Reviewer, we measured muscle TG (**Supplementary Fig. 7a**), and found that TG levels were also decreased in muscles. Therefore, the decreased muscle TG levels could also have contributed to the improvement in insulin resistance. As suggested, we have now measured serum insulin concentration under fasting conditions (**Fig. 1f, Fig. 5e and supplementary Fig. 1g**) and also measured insulin concentration during glucose tolerance test (**Fig. 1f and supplementary Fig. 1h**). The results showed that targeting *Mat1a* led to reduced insulin levels.

5. The increase in NRF2 expression is of significant interest. The investigators present data that suggest that the increase in oxidative stress may account for the increased expression of NRF2. The experiments in Fig 4c/d (which would benefit from a clearer explanation) suggest that simply trying to manipulate methionine levels does not result in the expected changes in FGF21. Is it possible to experimentally modulate GSH concentration to see if this is the mechanism by which *MAT1 ko* regulates NRF2 expression? Perhaps using an ester of GSH (e.g. GSH-methyl or ethyl esters?) Alternatively, perhaps other mechanisms are at work? What about NRF2 mRNA expression? If transcription is altered, perhaps *MAT1 ko* is altering DNA methylation.. a mechanism that perhaps could also explain alteration in ACC and FAS expression? If transcription is not altered.

Response: This is an interesting point and, as suggested by Reviewer, experiments have been performed in hepatocytes isolated from ASO treated HFD-fed mice. The addition of GSH ether or N-acetylcysteine (NAC) (an antioxidant, precursor of cysteine) to the culture medium attenuated the amount of FGF21 secreted by hepatocytes (induced by the *Mat1a* ASO) (**Fig. 7e**). The addition of SAMe 6mM, which is upstream in the methionine cycle also induced the same effect (**Fig. 7f**).

Nrf2 expression levels were also measured in the liver (**figure below**), but we did not find any significant changes when targeting liver *Mat1a*. We did not include this data in this Revised manuscript.

Figure. mRNA expression of *Nrf2* in liver of *Mat1a* ASO and Control ASO treated HFD-fed mice.

6. Is *MAT1a* really a viable target? Some have suggested that *MAT1* deficiency might impair liver regeneration (PMID: 15033934). Are there negative consequences to *MAT1* inhibition that we should be aware of?

Response: This is a really interesting question. These results show that *Mat1a* is a viable target in the context where BAT might be activated. The beneficial effect of targeting *Mat1a* is two-fold: the activation of BAT and the increased secretion of FGF21. In lean mice, the deletion of *Mat1a* may lead to the development and progression of NAFLD during aging (Lu SC et al, PNAS 2001). However, in a context of obesity, *Mat1a* deletion leads to reduction of hepatosteatosis. Bariatric surgery has been shown to reduce cancer risk in adults with NAFLD and severe obesity (Vinod K. Rustgi et al, Gastroenterology 2021); therefore, targeting *Mat1a* in the obese may also have the beneficial effect of preventing the development of liver cancer.

Of **note:** In the context of obesity, we have also measured additional liver and kidney function parameters (**New table 1 and New supplementary table 1**) after targeting *Mat1a* with ASO or ASO2, respectively, in mice fed a HFD for 10 weeks. ASOs were administered from the 6th week. In the new **supplementary table 2**, the parameters of liver and kidney function were obtained after a long-term study (mice fed a HFD were treated with the ASOs from the 6th week of HFD until maximal weight loss efficiency with plateaued weight loss for 2 weeks was obtained, at week 16 (**Supplementary Fig. 1j**). Overall, the results obtained show that targeting *Mat1a* after body weight loss did not induce the alteration of liver and kidney function parameters out of the normal reference range for mice (**Table 1, supplementary table 1 and supplementary table 2**).

In line with the suggestion of the reviewer and the apparent context-dependent action of *Mat1a*, we have now undertaken a new project to investigate whether targeting *Mat1a* in obesity could also protect from liver cancer.

REVIEWER COMMENTS

Reviewer #1 (Remarks to the Author):

In the revised manuscript, the authors performed dozens of new experiments, which overall tremendously improved this already nice manuscript. I can only applaud the author's attempts to finalize this manuscript. In light of the enormous work that has been done since the first submission (not only based on my comments, but also on the other reviewers comments), I think we can only congratulate the authors at this point for providing such a comprehensive and well-designed article.

Reviewer #2 (Remarks to the Author):

My comments to the authors have been emailed to the editor as an attached PDF.

Reviewer #3 (Remarks to the Author):

The authors have done an outstanding job addressing the prior critiques, with new data and revisions. The resulting manuscript is quite strong and the data paint an intriguing picture.

I only have a few minor comments.

1. Abstract: The abstract should reframe the work a little to highlight the importance of the energetic phenotype. For example, the sentence "By using antisense oligonucleotides (ASO) and genetic depletion of Mat1a, we demonstrate that Mat1a deficiency in diet-induced obese or genetically obese mice prevented and reversed obesity and obesity-associated insulin resistance and hepatosteatosis" could have a phrase like "by increasing energy expenditure in an FGF-21 dependent fashion" included. After all, the authors have shown nicely that the effects of MAT1 deficiency seem to hinge on the ability of FGF21 to increase BAT metabolism.

2. I am still not enthusiastic about the use of the word "channeled" to describe the increase in FA uptake and oxidation. Can the authors rephrase this?

3. The data from Supp Fig 8a/b should be either shown in the main text or described in the main text more clearly (i.e. % decrease/increase).

4. It would be helpful to scale the Y-axis for the EE vs BW graphs similarly (Fig 6e, Sup Fig 9d).

Reviewer #2 (Remarks to the Author):

The original points raised by reviewer #2 are in bold font. The original responses of the authors are in regular font, and any new comments from the reviewer are in italics and underlined.

Reviewer #2 (Reviewer Comments to the Author): The Introduction describes an association between high methionine intake and obesity, and dietary methionine restriction and reduced adiposity. Previous studies have shown that knocking out MAT1A also produces hypermethionemia and results in spontaneous development of fatty liver disease. What was the rationale for proposing that knocking down MAT1A with oligonucleotides would result in a phenotype that was opposite from the MAT1A knockout animals with respect to hepatic lipid metabolism?

Response: As this reviewer indicated Mat1a knockout induces hypermethionemia, and this is due to the lack of usage of methionine in liver. The first reaction of the methionine cycle is the conversion of methionine into SAMe and it is catalyzed by MAT1/III, the product of Mat1a gene. So a knock out in the Mat1a gene leads to the decreased usage of methionine, which accumulates in liver and consequently, in serum. Given that the dietary methionine restriction leads to the reversion of obesity, insulin resistance and fatty liver, we wanted to know if the decreased usage of methionine in liver could lead to the same phenotype. In addition, it has been demonstrated in other mouse models that the knockdown of enzymes of the methionine cycle (e.g. NNMT) induce resistance to diet induced obesity, the associated insulin resistance and hepatosteatosis. On the other hand, it has been reported that the dietary methionine restriction leads to the increase in serum FGF21. Thus, given the known beneficial effect of FGF21 in nonalcoholic fatty liver disease, we wanted to know if in the context of obesity, targeting Mat1a could also provide those beneficial effects. The results here demonstrated, as anticipated, that when Mat1a-KO mice were fed a HFD, there was a resistance to body weight gain, to develop insulin resistance, a high increase in serum FGF21 levels, an increase in FAO in BAT and resistance to develop fatty liver. Thus, the results obtained here using the Mat1a-KO mice are similar to those obtained when Mat1a-ASOs were used. The conclusion is that resistance to weight gain and to insulin resistance, is driven by hepatocyte secretion of FGF21, which protects the liver from nonalcoholic fatty liver disease. It is widely recognized that weight loss is the cornerstone of effective NAFLD treatment and it has been recently published that bariatric surgery reduces cancer risk in adults with NAFLD and severe obesity (Vinod K. Rustgi et al, Gastroenterology 2021); hence the rationale for such a study.

The following three papers show that knocking out hepatic MAT1a creates significant problems with liver lipid metabolism, and eventually results in fatty liver. This was pointed out in the initial review. Yet the authors responded with “The results here demonstrated, as anticipated, that when Mat1a-KO mice were fed a HFD, there was a resistance to body weight gain, to develop insulin resistance, a high increase in serum FGF21 levels, an increase in FAO in BAT and resistance to develop fatty liver. Thus, the results obtained here using the Mat1a-KO mice are similar to those obtained when Mat1a-ASOs were used”. This reviewer is not quibbling with the authors findings so much as asking why they have not properly dealt with the fact that their findings are completely opposite from what would have been predicted in relation to published literature. Multiple publications have documented what happens to the health of the animal when hepatic MAT1a function is compromised. Did feeding a high fat diet somehow change the previously documented physiological changes that occur when MAT1a is deleted? Did previous studies also observe the weight loss that occurred in the present study?

Methionine adenosyltransferase 1A knockout mice are predisposed to liver injury and exhibit increased expression of genes involved in proliferation. Shelly C. Lu, Luis Alvarez, Zong-Zhi Huang, Lixin Chen, Wei An, Fernando J. Corrales, Matías A. Avila, Gary Kanel, and José M. Mato <https://doi.org/10.1073/pnas.091016398>

Methionine adenosyltransferase 1A gene deletion disrupts hepatic very low-density lipoprotein assembly in mice†‡ Ainara Cano, Xabier Buqué, Maite Martínez-Uña, Igor Aurrekoetxea, Ariane Menor, Juan L. García-Rodríguez, Shelly C. Lu, M. Luz Martínez-Chantar, José M. Mato, Beqoña Ochoa, Patricia Aspichueta <https://doi.org/10.1002/hep.24607>

S-adenosylmethionine in Liver Health, Injury, and Cancer. Shelly C. Lu, and José M. Mato <https://doi.org/10.1152/physrev.00047.2011>

Near the end of the Introduction, the authors state that chronic changes in liver SAME levels in lean mice have been associated with the onset and progression of NAFLD with age, but isn't that due to a decrease in MAT1A/MAT2A,2B expression ratios that decrease the production of SAME? Why wouldn't knocking down MAT1A produce the same changes in SAME (and serum methionine) and if those metabolites are causative to the onset and progression of NAFLD, why would those changes produce opposite metabolic effects in the current model?

Response: As mentioned above, the hypothesis was that the decreased usage of methionine due to the lack of Mat1a could be linked, as demonstrated in this study, to increased secretion of FGF21 by hepatocytes, and that this increased secretion will lead to increased plasma levels and all the beneficial effects that are associated, including protection from hepatosteatosis. As mentioned above, weight loss is associated with regression of hepatic steatosis, and we have shown in this report that the activation of BAT by FGF21 is the putative mechanism. These results overall, suggest that activation of BAT even under certain non-favourable metabolic conditions, may prevent development and progression of chronic liver disease. The authors have called ex vivo synthesis of fatty acids in pieces of liver de novo lipogenesis while the term is normally reserved for in vivo measurement of fatty acid synthesis. This needs to be clarified for the reader. Response: The in vivo fatty acid synthesis is performed when the radioactive substrates are added intravenously to the mice. Slices from freshly isolated liver were also incubated with the 3H-acetate; thus, this is an ex vivo assay. These have all been fully described in the Methods section and the ex vivo term has been removed.

What happens to hepatic FGF21 expression when hepatic MAT1a is knocked out? If knocking down MAT1a is the sole cause for the increased FGF21 expression, shouldn't knocking it out produce the same effect on FGF21? Do MAT1a KO mice show any of the same metabolic outcomes as seen in the present experiment. If not, there is something fundamentally different between the two models and it seems like the authors would be interested in rigorously addressing this question.

In the Results, the authors have interpreted their triolein uptake data to mean that the substrate fueling an increased thermogenic rate is coming from outside brown adipose tissue. The results are convincing that triolein uptake is increased in BAT but what is the evidence that it is being oxidized? Significant previous work in BAT has shown that DNL and fat oxidation are coupled during cold-induced increases in BAT thermogenesis. This work suggests that the fatty acids being oxidized are

being synthesized in BAT. These conclusions are supported by work showing that genetic or pharmacological impairment of DNL or oxidation compromises cold-induced thermogenesis.

Response: Thank you for the interesting comments. The results demonstrate that lipids in BAT are being oxidized and catabolized, no matter the source. We measured ACC, p-ACC and FAS protein levels, which are the main enzymes involved in de novo lipogenesis (Supplementary Fig. 5b). The expression of Acaca, Acacb and Fasn mRNA were also measured (Supplementary Fig. 5a). The results showed a decrease in the protein levels of ACC, p-ACC and FAS and decreased expression of Acaca suggesting that the de novo lipogenesis was decreased.

The authors provided a number of measurements suggesting that thermogenesis was activated in BAT after knocking down MAT1a, perhaps as a result of the increase of FGF21 release from the liver. The initial points I raised were an attempt to get the authors to question their conclusion about the source of the lipid being oxidized in BAT to fuel the thermogenesis. In addition to the studies alluded to showing that blocking either lipid synthesis or lipid oxidation within BAT rendered the animals unable to respond to cold exposure, another comprehensive study conducted in SH Adams lab (e.g., [Doi.org/10.1096/fj.01-0568com](https://doi.org/10.1096/fj.01-0568com)) examined the transcriptional changes in BAT during adaptation to different housing temperatures. The goal of their work was to understand whether the increased glucose uptake occurring in BAT during cold exposure was providing the substrate for lipid synthesis that were then oxidized to fuel thermogenesis. They documented significant increases in ACC, FAS, and SCD1, which are all essential for DNL and triglyceride synthesis. So how is it in the present work where thermogenesis is proposed as the mechanism for the weight loss, that knockdown of hepatic MAT1a produces a completely opposite effect on lipogenic potential in BAT and source of lipid to fuel the underlying thermogenesis?

The authors evidence of increased thermogenesis comes from Fig. 4F where they show that a beta3-AR agonist increases O2 consumption more in Mat1a ASO mice than Control ASO mice. However, these results are compromised by the problem of not describing how this experimental protocol was conducted and how the data were analyzed. How was the beta3-AR agonist given to the mice while they were in the calorimeters? The results in Fig. 4F show no break in the collection of data associated with this event. More troubling is how the VO2 consumption data are being scaled. The ordinate is labeled ml/kg/h, indicating that the O2 consumption data is being scaled by BWs, which are ~15g lower in the Mat1a ASO group. As described in multiple manuscripts describing the correct approach to analyzing IDC data in recent years, big differences in BW confound group comparisons by over-amplifying the calculated EE in smaller animals within the groups.

Response: Thank you for your comments. We agree, this reviewer is correct, and we have now removed Fig. 4F in the Revised manuscript. This reviewer is also correct in pointing out about how we should be representing energy expenditure. There is certainly debate how we should be interpreting energy expenditures in mice with varying body weights. The ANCOVA analysis of the energy expenditure vs the body weight is widely accepted (for guidance we followed: Tschop M et al. A guide to analysis of mouse energy metabolism. Nat Methods. 2011 Dec 28;9(1):57-63). Therefore, we have performed this analysis (also following the recommendation of reviewer 1) for all the measurements of energy expenditure in the Revised manuscript. To study the effects of β -adrenergic agonist, we performed the ANCOVA analysis at 30 and 45 mins after the administration of the drug. We also measured Ucp1 and Pgc1a expression in BAT after the after the β -adrenergic stimulation (Supplementary Fig. 4c).

Let me clarify the problem I'm having with these data. When the Mouse Metabolic Phenotyping Center consortium came into existence, they sought to develop a series of recommendations as to how indirect calorimetry experiments should be conducted so that investigators could obtain reliable, reproducible

data. One of their key recommendations was that it was necessary to allow mice to equilibrate for 24 to 48 h in the calorimeters before stable, trustworthy measurements of VO₂ consumption and CO₂ production could be measured. Therein lies the problem with the authors methods. Did they inject mice with the beta-3 agonist, place them in the calorimeters, and measure their gas exchange 30 to 45 min later and expect the reader to believe this is a valid assessment of the EE response of the two groups of mice to the treatment? Using ANCOVA to analyze the data is an improvement over their original methods but it does not solve the problem highlighted above.

The data presented in Fig. 7 provide support for the idea that knocking down Mat1a in liver activates FGF21 release from the liver through a NRF2-dependent mechanism. The authors propose that this occurs after nuclear localization of NRF2 and transcriptional activation of FGF21. If the authors are able to detect nuclear localization of NRF2 with available reagents, they should confirm the direct role of NRF2 using ChIP assays to show that NRF2 is in fact being recruited to the FGF21 promoter. If NRF2 is necessary and sufficient for transcriptional activation of FGF21 after Mat1a ASO treatment, they should also show that hepatic deletion of NRF2 blocks the induction of FGF21. In addition, more information is needed about the NRF2 inhibitor used in Fig. 7D. Its source needs to be identified, along with a reference to a publication demonstrating its efficacy and specificity.

Response: Thank you for this interesting suggestion. We have performed the ChIP assay in hepatocytes isolated from ASO treated HFD-fed mice. The results confirmed the binding of NRF2 in the Fgf21 promoter (Fig. 7d). We also performed an in vitro experiment in which Nrf2 was silenced by siRNAs, and found that silencing Nrf2 abolished the increased FGF21 secretion that occurred when targeting liver Mat1a (Supplementary Fig. 10b). A similar result was observed when ML385 (a specific NRF2 inhibitor) was used (Fig. 7a), and we have now included information on this inhibitor in the Methods section. In brief, the NRF2 inhibitor ML385 is a probe molecule that binds to NRF2 and inhibits its downstream target gene expression. Specifically, ML385 binds to Neh1, the Cap 'N' Collar Basic Leucine Zipper (CNCbZIP) domain of NRF2, and interferes with the binding of the V-Maf Avian Musculoaponeurotic Fibrosarcoma Oncogene Homolog G (MAFG)-NRF2 protein complex to regulatory DNA binding sequences. ML385 is a novel and specific NRF2 inhibitor (Singh A et al, ACS Chem Biol. 2016)

Specific Points 1. The description of the methods used to conclude that EE was affected by the treatments is inadequate. It refers to a previous publication but when that publication is examined it provides no details on how the calorimetry data were analyzed. The problem must be rectified in the current paper. Fig. 1G is used to make a claim that EE is increased by the MAT1A ASO but the data in the figure shows that EE was scaled by kg but does not define kg of what. Moreover, the accompanying figure panel indicates that there is no relationship between EE and BW over a weight range of 24-29 grams. This seems improbable since EE should increase roughly in proportion to BW raised to some power < 1. An additional problem is that Fig. 1C shows that the two groups of mice being compared in Fig. 1G have approximately the same spread in BWs of mice within each group while Fig. 1C shows that the mice in the two groups were ~15 g different in BW. Together, this creates a lot of confusion for the reader, even for one experienced in the analysis of indirect calorimetry data. Please see <https://www.nature.com/articles/s41592-019-0513-9> for guidance in analyzing and presenting EE data.

The reference provided above is an update of the original article published by Tschop et al. 8 years later. Progress and challenges in analyzing rodent energy expenditure, Rodrigo Fernández-Verdejo, Eric Ravussin, John R. Speakman & Jose E. Galgani, Nature Methods volume 16, pages797–799 (2019)

Response: We have removed this publication and we have added another one. On the other hand, as mentioned above, taking into account the reviewer 1, this reviewer suggestions and the published information (Tschop M et al. A guide to analysis of mouse energy metabolism. Nat Methods. 2011 Dec 28;9(1):57-63), we agree that the ANCOVA analysis should be used for comparison of energy expenditure when body mass or composition differs between groups. Thus, ANCOVA has been used for all these analyses, including former Fig. 1g, now Fig. 1h. 2.

OGTTs measure a composite of glucose absorption rates, alterations in B-cell sensitivity to glucose, and insulin-dependent glucose excursions. Given the relatively small improvement in insulin tolerance and only at the last time point, the substantial improvement in glucose tolerance shown in Fig 1E points to changes in the first two factors rather than solely insulin sensitivity.

<https://pubmed.ncbi.nlm.nih.gov/18670420/>

The 60 min time point shouldn't even be considered as a marker of insulin action. If you don't see an improvement in the first 20 min, it's probably not increased insulin action. What's more, the substantial improvement in glucose tolerance after Mat1a ASO is likely secondary to the loss in BW.

Response: Thank you for your comments. For this Revised manuscript, we have added New GTTs, ITTs and measured serum insulin levels in various time points (Fig. 1f and supplementary Fig. 1h). The levels of glucose in the ITTs have been represented as mg/dl, as recommended by the first reviewer; fasting insulin levels have also been quantified in the DIO induced mice (time 0 of the insulin levels of the GTT time points) (Fig. 1f and supplementary Fig. 1g) and in the ob/ob mice (Fig. 5e). In addition, the GTT, ITTs and measurements of insulin released during the GTTs have been performed at two different situations in the HFD-fed mice; 1) after the first injection of ASOs, when body weight loss is not that evident (supplementary Fig. 1h), and 2) by the end of the treatments (Fig. 1f). The results showed that after the first dose of ASOs (7th week of HFD), insulin levels were already lower before the bolus of glucose in the GTT (time 0), and after 30 min of the glucose administration. The results showed that the insulin released in each time point and the levels during fasting were markedly lower when targeting Mat1a. Fasting levels of insulin were also decreased when Mat1a was targeted in the ob/ob mice. These collective results show that targeting liver Mat1a protects against insulin resistance.

3. What were the fasting insulin concentrations in the two groups of mice?

Response: Fasting insulin concentrations were measured as described above, under various conditions.

4. From looking at supplementary Fig. 1F, there appears to be a problem in the way the MAT1A KO mice responded to the HF diet. After it was introduced, the mice lost ~5 g in the first week and then stabilized their weight over the following 8 wks while the WT mice gained over 20 g. It would help the reader to understand the data better if the authors would present the actual BWs of the mice in the

two groups and the actual mean food intakes for the two groups over time. Presenting the data as cumulative food intake makes it more difficult to see what is happening over time to food intake. This suggestion also applies to the MAT1A ASO mice, where it would be better to see the actual BWs of the mice in the two groups and their week by week food consumption rates. Looking at supplemental Figs 1f and 1g, it is hard to believe that mice differing in BW by 25 g are eating exactly the same amount of food. Lastly the actual BWs of the animals in each group is important to know relative to the analysis of the indirect calorimetry data. The authors should add another figure panel and show the actual BWs of the mice in each group.

Response: As the reviewer suggests we have included the BWs of all animal models (sometimes as supplementary figures due to the lack of space) and the food intake for all the different groups. In all these models, the results show always unchanged food intake. The energy expenditures are now all represented vs the body weight with the ANCOVA test performed so that the two parameters can be easily related. In all animal models tested, the inhibition of Mat1a was associated to higher energy expenditure. The fact that Mat1a ASO mice show a lower weight without changes in food intake is not that surprising, since many previous reports have found other animal models with a marked decrease in body weight in a feeding-independent manner. Just to mention some previously published examples from coauthors of the manuscript (Folgueira C et al, Nat Metab. 2019) or from other groups (Pirzgalska RM et al, Nat Med. 2017). On the other hand, we had the same thought when we showed this for the first time, so before sending the first version of the manuscript we performed again the analysis of the response of Mat1a-KO mice to the HFD; for this, we fed Mat1a-KO mice the HFD for a week and we obtained the same result. These results are shown in the figure below. Although we don't have a clear explanation for this, it seems that Mat1a KO mice are somehow protected against HFD-induced early metabolic dysfunctions. Indeed, this interesting question deserves to be investigated in a further study.

5. The variability of the Sirius Red and F4/80 measures is concerning, indicating that this approach to detecting fibrosis and inflammation is insensitive to the point of being useless.

Response: It is true that there is variability in the results from the histochemistry procedures; however, do not believe that histochemistries are 'useless'. We have now also analyzed the expression of genes involved in fibrosis and inflammation (Supplementary Fig. 6a). The heterogeneity of the results is also high but there were no statistical differences. We have performed the analysis also when using the Mat1a ASO2 (Supplementary Fig. 6c). The studies show that targeting Mat1a does not lead to any increase in the expression of genes involved in fibrosis or inflammation.

Let me further clarify my contention that the Sirius Red and F4/80 measures are of limited value by saying that when 0 is within the confidence interval around your mean, how can you conclude that your measurement is different from 0 (see Fig. 3A).

6. In Fig. 3D, triolein uptake is expressed per g of tissue. What do the results look like when corrected for differences in tissue weight between treatment groups?

Response: In the Figure below, we show the uptake of labelled lipids (DPM per total tissue or per mg of protein). The results confirm that targeting liver Mat1a leads to an increase in the uptake of dietary lipids in BAT.

7. There is a problem with the interpretation of the fatty acid oxidation data in the sense that the assays do not measure in situ rates of fatty acid oxidation. With a fixed amount of C14-palmitic acid present in the in vitro incubations, the assays are providing an estimate of Vmax or capacity of the tissues from the two groups to oxidize fatty acids. This is not a measure of the rate yet it is how the figures summarizing this data is labeled.

Response: We have performed the fatty acid oxidation analysis ex vivo, not in vitro, as previously reported and described in other published manuscripts (e.g. Hirschey MD et al, Nature 2010). The [1-14C]-palmitic acid that is added to the samples is always 0.5 μ Ci/ml, but this corresponds to a very low quantity of palmitic acid. Thus, we believe the reviewer is probably referring to the carrier that is usually added in the reaction, which is the cold palmitate (500 μ M in the assay that has been used in this study), usually added to improve the signal and to promote the uptake of the radioactive substrate, given that the quantity of [1-14C]-palmitic acid is so low. In this assay, the flux or the rate of fatty acid oxidation has been measured as reported in multiple publications (e.g., Hirschey MD et al, Nature 2010; Frank K. Huynh et al. Methods Enzymol. 2014; Gao X et al. Biochim Biophys Acta. 2015; Lucía Barbier-Torres et al, Nat Comm. 2020; Gonzalez-Romero F, Mestre D et al, Cancer Res 2021). In addition, in our mice models, as frequently happens in other projects with different treatments or diseases, the amount of fatty acids that reach the mitochondria varies, so in order to determine the FAO rate, one should know the mitochondria capacity to catabolize those lipids. For this, an external amount of fatty acids should always be added in the same concentration in all the groups being analyzed. Of relevance in this study, is the FAO rate analysis in these conditions taking into account that the lipid that reaches the BAT mitochondria is higher when targeting liver Mat1a; however, the amount of lipid in BAT is lower than when mice are treated with the Control ASO. To get an accurate result of the flux or the rate of FAO, the researchers need to add the same amount of external fatty acid in the two groups that are being analyzed. This is something widely accepted to measure the FAO rate as reported in the manuscripts above. If the reviewer does not feel comfortable with the terminology FAO rate (which is because the units are always per hour), we will remove rate when describing the results.

The pedantic comments of the authors notwithstanding, they have continued to refer to FAO measurements in BAT explants in terms of rate, implying that actual in vivo rates of FAO in BAT are different (e.g., HFD-fed Mat1a ASO (Fig. 2b), Mat1a ASO2 (Supplementary Fig. 4a), and Mat1a KO mice (Supplementary Fig. 4a), all exhibited an increase in FAO rate in BAT when compared to the corresponding controls. See narrative of P 7 of revised MS). They also indicate in the Fig. 2 legend that they are measuring FAO rate in BAT. This is an inaccurate reporting of what was actually measured, which was the rate of CO2 production from BAT explants in a test tube using single fixed concentrations of a mixture of cold and C14-palmitate. The fact that this reaction is allowed to proceed for a fixed period of time is where the rate term enters into the calculation, and although reaction conditions have been standardized across groups, the calculated rate is an enzymatic rate and not the rate of BAT FAO that is being implied by the way this data is presented. This same problem extends to Fig. 3b where the authors again imply that they are measuring liver fatty acid oxidation rates when what they measured

was FAO in hepatic explants. The implication that these are in vivo measurements is obvious and they are misleading.

This problem extends to the misleading description of de novo TG lipogenesis measurements. Fig. 3c legend indicates “Liver TG de novo lipogenesis determined by incorporation of [3H]-acetate into TG in HFD-fed control (n=7) and Mat1a (n=8) ASO-treated mice.” This legend strongly implies these were in vivo measurements but when one reads the Methods section it becomes clear that these measurements were made in liver explants. DNL is an in vivo process in which carbohydrates from circulation are converted into fatty acids that are then used for synthesizing either triglycerides or other lipid molecules, and there are well-defined methods for measuring the in vivo process (<https://www.nature.com/articles/1600744.pdf?origin=ppub>).

8. It's unclear what the isoproterenol-dependent increase in ex vivo lipolysis adds to the story and why the responses from the Control ASO CD mice were not included in Fig. 4d. Lipolytic responsiveness to beta-adrenergic agonists is impacted by a number of factors and what the authors are showing is probably mostly dependent on the decrease in adiposity and fat cell size in the Mat1a ASO group.

Response: The assay of lipolysis is frequently performed in the presence and/or absence of the β adrenergic receptors agonist isoproterenol as has been described in other studies (Bradlee L. Heckmann et al, J Biol Chem., 2014; Gao X et al. Biochim Biophys Acta. 2015), so that the capacity of that tissue under basal and stimulated conditions are measured. In this Revised manuscript, we have added the control ASO CD mice and have calculated and represented the lipolysis results in nmol of fatty acid or glycerol released per mg of tissue protein; these will provide clarity on the influence of fat cell size or adiposity (Supplementary Fig. 3c).

The authors misunderstood the point of the question about this data. What does it add in terms of helping the reader to understand the resulting phenotype. Do the authors believe that knocking down hepatic MAT1a alters the responsiveness of EWAT to mobilize glycerol and FFAs in response to a supraphysiological concentration of a beta-agonist that activates all beta-adrenergic receptor subtypes? What does this data add to understanding the resulting phenotype? Fig. 2D suggests that the HF diet significantly diminishes the capacity of the adipocytes to release glycerol in response to iso. What is the basis for this difference and why wasn't FFA release similarly compromised in this group?

9. Which WAT depot was used to produce the data presented in Fig 4? It doesn't seem to be specified in either the Methods or the fig legend. This is important because the capacity of WAT to undergo remodeling is depot specific.

Response: As has been frequently used for this kind of analysis, we have used epididymal WAT. This has been added to the methodology section.

10. As with Fig 1, please present the actual BW means of the mice in each group, as well as their actual food intakes each week.

Response: As mentioned above this have been performed.

11. Ob/ob mice typically have high fasting insulin levels by 12-15 wks of age. What impact did the 5-6 g difference in BW produced by the Mat1a ASA have on fasting insulin levels?

Response: We thank the reviewer for this suggestion. Now, we have been performed this analysis, levels of fasting insulin are markedly lower in Mat1a ASO than in the Control ASO ob/ob treated mice. In the new version of the manuscript has been added in Fig. 5e.

12. The problem noted above for FA oxidation measurements is also applicable in Fig. 5F. The authors are measuring the amount of palmitate oxidation with a fixed amount of palmitate and concluding that the measurement is indicative of in vivo fatty acid oxidation in BAT. The data do not fully support this conclusion.

Response: As mentioned above, the FA oxidation rate has been measured ex vivo. We did not perform in vivo fatty acid oxidation; this must be a misunderstanding. Our results show that FAO rate is increased in BAT when targeting liver Mat1a. FAO rate (nmol of ASM or CO₂ generated from palmitate per g of tissue and per hour) has been measured as described above. This assay measures the whole FAO rate and not just a measure of an enzyme activity. This method is well established in our lab (Lucía Barbier-Torres et al, Nat Comm. 2020; Gonzalez-Romero F, Mestre D et al, Cancer Res 2021) and is a well accepted assay (Hirschey MD et al, Nature 2010; Frank K. Huynh et al. Methods Enzymol. 2014; Gao X et al. Biochim Biophys Acta. 2015).

13. The authors measurement of PPAR α expression in response to Mat1a treatment is not an effective approach to determining whether the increase in serum FGF21 is due to PPAR α activation. The same criticism should be considered for measurement of nuclear ATF4 by Western, particularly since it is widely known in the field that reliable measurement of endogenous ATF4 by Western is difficult to establish. The data in supplementary Fig 5 certainly do not meet the standard for ruling out involvement of either transcription factor in Mat1a ASO-dependent induction of hepatic FGF21.

Response: We also performed experiments with an inhibitor of PPAR α or ER stress. In this Revised Manuscript, additional experiments where isolated hepatocytes from ASO treated HFD-fed mice were exposed to siRNAs to silence Ppara and Atf4 were performed (Supplementary Fig. 10a). Similar to prior experiments using PPAR α and of ER stress inhibitor, these new experiments showed that Ppara and Atf4 are not involved in the increased FGF21 secretion by hepatocytes (Supplementary Fig. 10b). We also measured levels of PPAR α and ATF4 in the nucleus, and found that targeting Mat1a did not lead to changes in their levels of expression (Supplementary Fig. 10c). In concert, both these studies confirm that neither PPAR α nor ATF4 are involved in the increased secretion of FGF21.

14. In the last paragraph of the Results, the authors examine FGF21 in the media of hepatocytes from mice after knocking down Mat1a and adding SAME to the media. How much SAME was being added to the media and what proportion of the amount being added is ending up inside the cells? There

seems to be an assumption that SAME is crossing the cell membrane. What is the evidence for that occurring?

Response: This is a very interesting point. We have now added 6mM of SAME to the media and have measured expression levels of Bhmt, Sahh and Cbs mRNA, genes involved in the methionine cycle (4 and 24 hours after adding SAME) and have also quantified SUMO1 and SUMO1-conjugated proteins (24 hours after adding SAME) (Supplementary Fig. 11b) (since SAME is a SUMOylation inhibitor; Pui Y. Lee-Law et al, J Hepatol 2020).

Additional Points 1. It would have been helpful if page numbers were provided in the MS.

Response: Page numbers are now provided.

2. Please define which oligos were used in supplementary table 1. Is ASO the control oligo and ASO2 the antisense oligo? In addition, measures of reproducibility must be provided.

Response: The ASOs sequence have now been provided in the methodology section and here: Control ASO: 5'-CCTTCCCTGAAGGTTCTCC-3' Mat1a ASO: 5'-CCACTTGTCATCACTCTGGT-3' Mat1a ASO2: 5'-GCTCAGGAGACATTGACCAT-3' IONIS Pharmaceuticals uses the same Control ASO. For this study, to knock-down Mat1a they created two different ASOs, which we have labelled as Mat1a ASO and Mat1a ASO2. In our preliminary studies, we found that the ASO dose required to silence 90% Mat1a was 25 mg/kg/week for Mat1a ASO and 50 mg/kg/week of Mat1a ASO2. The majority of studies were performed with the Mat1a ASO. Mat1a ASO2 were used in some experiments to confirm changes. 3.

Please specify the temperature at which the mice were housed.

Response: Mice were housed at 21-22 °C, this has been added now in the methodology section.

October, 2021

Point-by-point response to reviewers:

Reviewer #1 (Remarks to the Author):

In the revised manuscript, the authors performed dozens of new experiments, which overall tremendously improved this already nice manuscript. I can only applaud the author's attempts to finalize this manuscript. In light of the enormous work that has been done since the first submission (not only based on my comments, but also on the other reviewers comments), I think we can only congratulate the authors at this point for providing such a comprehensive and well-designed article.

New Response: Thank you very much. We really appreciate these words.

Reviewer #2 (Reviewer Comments to the Author):

The original points raised by reviewer #2 are in bold font. The original responses of the authors are in regular font, and any new comments from the reviewer are in italics and underlined.

New Response: Thank you. Now our new responses are indicated as "**New response**" and in a dark blue.

The Introduction describes an association between high methionine intake and obesity, and dietary methionine restriction and reduced adiposity. Previous studies have shown that knocking out MAT1A also produces hypermethionemia and results in spontaneous development of fatty liver disease. What was the rationale for proposing that knocking down MAT1A with oligonucleotides would result in a phenotype that was opposite from the MAT1A knockout animals with respect to hepatic lipid metabolism?

Response: As this reviewer indicated *Mat1a* knockout induces hypermethionemia, and this is due to the lack of usage of methionine in liver. The first reaction of the methionine cycle is the conversion of methionine into SAMe and it is catalyzed by MATI/III, the product of *Mat1a* gene. So a knock out in the

Mat1a gene leads to the decreased usage of methionine, which accumulates in liver and consequently, in serum. Given that the dietary methionine restriction leads to the reversion of obesity, insulin resistance and fatty liver, we wanted to know if the decreased usage of methionine in liver could lead to the same phenotype. In addition, it has been demonstrated in other mouse models that the knockdown of enzymes of the methionine cycle (e.g. NNMT) induces resistance to diet induced obesity, the associated insulin resistance and hepatosteatosis. On the other hand, it has been reported that the dietary methionine restriction leads to the increase in serum FGF21. Thus, given the known beneficial effect of FGF21 in nonalcoholic fatty liver disease, we wanted to know if in the context of obesity, targeting *Mat1a* could also provide those beneficial effects. The results here demonstrated, as anticipated, that when *Mat1a*-KO mice were fed a HFD, there was a resistance to body weight gain, to develop insulin resistance, a high increase in serum FGF21 levels, an increase in FAO in BAT and resistance to develop fatty liver. Thus, the results obtained here using the *Mat1a*-KO mice are similar to those obtained when *Mat1a*-ASOs were used. The conclusion is that resistance to weight gain and to insulin resistance, is driven by hepatocyte secretion of FGF21, which protects the liver from nonalcoholic fatty liver disease. It is widely recognized that weight loss is the cornerstone of effective NAFLD treatment and it has been recently published that bariatric surgery reduces cancer risk in adults with NAFLD and severe obesity (Vinod K. Rustgi et al, Gastroenterology 2021); hence the rationale for such a study.

The following three papers show that knocking out hepatic MAT1a creates significant problems with liver lipid metabolism, and eventually results in fatty liver. This was pointed out in the initial review. Yet the authors responded with “The results here demonstrated, as anticipated, that when Mat1a-KO mice were fed a HFD, there was a resistance to body weight gain, to develop insulin resistance, a high increase in serum FGF21 levels, an increase in FAO in BAT and resistance to develop fatty liver. Thus, the results obtained here using the Mat1a-KO mice are similar to those obtained when Mat1a-ASOs were used”. This reviewer is not quibbling with the authors findings so much as asking why they have not properly dealt with the fact that their findings are completely opposite from what would have been predicted in relation to published literature. Multiple publications have documented what happens to the health of the animal when hepatic MAT1a function is compromised. Did feeding a high fat diet somehow change

the previously documented physiological changes that occur when MAT1a is deleted? Did previous studies also observe the weight loss that occurred in the present study?

Methionine adenosyltransferase 1A knockout mice are predisposed to liver injury and exhibit increased expression of genes involved in proliferation. Shelly C. Lu, Luis Alvarez, Zong-Zhi Huang, Lixin Chen, Wei An, Fernando J. Corrales, Matías A. Avila, Gary Kanel, and José M. Mato
<https://doi.org/10.1073/pnas.091016398>

Methionine adenosyltransferase 1A gene deletion disrupts hepatic very low-density lipoprotein assembly in mice†‡ Aina Cano, Xabier Buqué, Maite Martínez-Uña, Iqor Aurrekoetxea, Ariane Menor, Juan L. García-Rodríguez, Shelly C. Lu, M. Luz Martínez-Chantar, José M. Mato, Begoña Ochoa, Patricia Aspichueta
<https://doi.org/10.1002/hep.24607>

S-adenosylmethionine in Liver Health, Injury, and Cancer. Shelly C. Lu, and José M. Mato
<https://doi.org/10.1152/physrev.00047.2011>

New Response: We apologize for not having been able to respond properly, there has been a misunderstanding. The manuscripts describing the effect in the liver after the deletion of *Mat1a* in chow diet fed mice show that with age they develop spontaneously NASH (information included in the introduction section and in the discussion of the manuscript) and that they are more susceptible to choline-deficient diet-induced fatty liver (Lu S et al, PNAS 2001). However, as mentioned, some data regarding the loss of body weight and the beneficial effects of a methionine restriction diet (Latimer MN et al, Front Endocrinol 2018) and of the knock-down of other enzymes of the methionine cycle such as NNMT (Kraus D et al, Nature 2014), made us think about the possibility of targeting *Mat1a* to prevent/reverse obesity and maybe also some comorbidities. To be honest, we had no clue of what we were going to find in liver given the current literature.

In Cano et al, Hepatology 2011, we found that 3 month-old *Mat1a*-KO mice had no signs of liver injury and normal liver histology, normal body weight while VLDL-TG secretion was decreased. However, 8 month-old *Mat1a*-KO mice displayed macrovesicular steatosis and the secretion of TG in VLDL was increased together with a dyslipidemia. Again, we did not know what to expect regarding TG levels in serum or VLDL-TG secretion rate in the context of obesity, given the beneficial effects of body weight

loss in these parameters. As the Reviewer mentions, the results obtained here have been different, since we have found that knocking down *Mat1a* in high-fat diet (HFD) fed mice, led to decreased serum TG levels while VLDL-TG secretion maintained unaltered. Thus, in our opinion and based on the facts explained above, HFD-induced obesity causes a metabolic switch when *Mat1a* is deleted, activating the liver-BAT axis. This action prevents body weight gain (in *Mat1a*-KO mice), reverses obesity (in *Mat1a* ASO treated mice) and seems to protect against the development of fatty liver (in both models).

Last, answering if previous studies observed the weight loss that occurred in the present study; it is important to highlight that previous works have been performed in *Mat1a*-KO mice fed a chow diet (except from Lu S et al, PNAS 2001, in which a choline deficient diet was also used) and no changes were observed in body weight (Cano et al, Hepatology 2011), as in the present study (**Rebuttal Fig. 1a and b and results section of the manuscript as data not shown**). In the present study, weight loss is only observed when targeting *Mat1a* in HFD-fed mice. Even more, under prolonged HFD feeding, *Mat1a* ASO treatment resulted in initial weight loss, but body weight stabilized when it reached a level comparable with CD-fed mice (**Supplementary Fig 1j and Rebuttal Fig. 1c**).

Rebuttal Fig. 1. a) Body weight in chow diet (CD)-fed *Mat1a*-KO mice; b) body weight in CD-fed *Mat1a* ASO treated mice; c) Body weight changes, body weight and food intake during 16 weeks of HFD. Statistical differences comparing Control ASO CD vs Control ASO HFD are indicated by "#", and Control ASO HFD vs. *Mat1a* ASO HFD by "*". * $p < 0.05$, ** $p < 0.01$, and *** $p < 0.001$ (Student's test). The same for #.

Near the end of the Introduction, the authors state that chronic changes in liver SAME levels in lean mice have been associated with the onset and progression of NAFLD with age, but isn't that due to a decrease in MAT1A/MAT2A,2B expression ratios that decrease the production of SAME? Why wouldn't knocking down MAT1A produce the same changes in SAME (and serum methionine) and if those metabolites are causative to the onset and progression of NAFLD, why would those changes produce opposite metabolic effects in the current model?

Response: As mentioned above, the hypothesis was that the decreased usage of methionine due to the lack of *Mat1a* could be linked, as demonstrated in this study, to increased secretion of FGF21 by hepatocytes, and that this increased secretion will lead to increased plasma levels and all the beneficial effects that are associated, including protection from hepatosteatosis. As mentioned above, weight loss is associated with regression of hepatic steatosis, and we have shown in this report that the activation of BAT by FGF21 is the putative mechanism. These results overall, suggest that activation of BAT even under certain non-favourable metabolic conditions, may prevent development and progression of chronic liver disease.

What happens to hepatic FGF21 expression when hepatic MAT1a is knocked out? If knocking down MAT1a is the sole cause for the increased FGF21 expression, shouldn't knocking it out produce the same effect on FGF21? Do MAT1a KO mice show any of the same metabolic outcomes as seen in the present experiment. If not, there is something fundamentally different between the two models and it seems like the authors would be interested in rigorously addressing this question.

New Response: We have measured FGF21 in *Mat1a*-KO mice and *Mat1a* ASO treated mice fed a chow diet (CD). We have found in both models that serum FGF21 is increased (**Rebuttal Fig. 2 a and b**) and body weight (Rebuttal Fig. 1a and b) and liver TG levels unaltered (**Rebuttal Fig 2c and d**). When feeding a HFD, FGF21 was also found increased in *Mat1a*-KO mice and *Mat1a* ASO treated mice, as shown in the manuscript (**Fig. 6a of the manuscript**). As mentioned above, new studies, performed for the first revision, showed that under prolonged HFD feeding, *Mat1a* ASO treatment resulted in initial weight loss, but body weight stabilized when it reached a level comparable with CD-fed mice (**Supplementary Fig 1j of the manuscript and Rebuttal Fig. 1c**). Thus, the results show that *Mat1a* ASO induces body weight loss in obese mice until they reach a weight of a lean mouse. The fact that *Mat1a*-KO mice are protected from a HFD-induced body weight gain (**Supplementary Fig 2a of the manuscript**), is in concordance with these findings.

Rebuttal Fig. 2. a) Serum FGF21 levels in chow diet (CD)-fed *Mat1a*-KO mice; b) serum FGF21 levels in CD-fed *Mat1a* ASO treated mice; c) Liver triglycerides (TG) in CD-fed *Mat1a*-KO mice; d) Liver TGs in CD-fed *Mat1a* ASO treated mice. Statistical differences were indicated as ** $p < 0.01$ (Student's test).

The authors have called ex vivo synthesis of fatty acids in pieces of liver de novo lipogenesis while the term is normally reserved for in vivo measurement of fatty acid synthesis. This needs to be clarified for the reader.

Response: The *in vivo* fatty acid synthesis is performed when the radioactive substrates are added intravenously to the mice. Slices from freshly isolated liver were also incubated with the ^3H -acetate; thus, this is an *ex vivo* assay. These have all been fully described in the Methods section and the *ex vivo* term has been removed.

In the Results, the authors have interpreted their triolein uptake data to mean that the substrate fueling an increased thermogenic rate is coming from outside brown adipose tissue. The results are convincing that triolein uptake is increased in BAT but what is the evidence that it is being oxidized? Significant previous work in BAT has shown that DNL and fat oxidation are coupled during cold-induced increases in BAT thermogenesis. This work suggests that the fatty acids being oxidized are being synthesized in BAT. These conclusions are supported by work showing that genetic or pharmacological impairment of DNL or oxidation compromises cold-induced thermogenesis.

Response: Thank you for the interesting comments. The results demonstrate that lipids in BAT are being oxidized and catabolized, no matter the source. We measured ACC, p-ACC and FAS protein levels, which are the main enzymes involved in *de novo* lipogenesis (**Supplementary Fig. 5b**). The expression of *Acaca*, *Acacb* and *Fasn* mRNA were also measured (**Supplementary Fig. 5a**). The results showed a decrease in the protein levels of ACC, p-ACC and FAS and decreased expression of *Acaca* suggesting that the *de novo* lipogenesis was decreased.

The authors provided a number of measurements suggesting that thermogenesis was activated in BAT after knocking down MAT1a, perhaps as a result of the increase of FGF21 release from the liver. The initial points I raised were an attempt to get the authors to question their conclusion about the source of the lipid being oxidized in BAT to fuel the thermogenesis. In addition to the studies alluded to showing that blocking either lipid synthesis or lipid oxidation within BAT rendered the animals unable to respond to cold exposure, another comprehensive study conducted in SH Adams lab (e.g., Doi.org/10.1096/fj.01-0568com) examined the transcriptional changes in BAT during adaptation to different housing temperatures. The goal of their work was to understand whether the increased glucose uptake occurring in BAT during cold exposure was providing the substrate for lipid synthesis that were then oxidized to fuel thermogenesis. They documented significant increases in ACC, FAS, and SCD1, which are all essential for DNL and triglyceride synthesis. So how is it in the present work where thermogenesis is proposed as the mechanism for the weight loss, that knockdown of hepatic MAT1a produces a completely opposite effect on lipogenic potential in BAT and source of lipid to fuel the underlying thermogenesis?

New Response: As the reviewer points out in the study conducted in SH Adams lab, mice fed a chow diet exhibit increased levels of lipogenic genes in BAT during cold exposure. As mentioned by the

Reviewer, knock-down of hepatic *MAT1a* produces an opposite effect on lipogenic potential, according to protein levels, in BAT. However, it has also been described that BAT activity controls plasma triglyceride clearance (Alexander Bartelt et al, Nat Med, 2011). The authors showed that cold exposure drastically accelerates plasma clearance of triglycerides (TG) as a result of increased uptake into BAT. They conclude that “BAT activity controls vascular lipoprotein homeostasis by inducing a metabolic program that boosts triglyceride-rich lipoproteins turnover and channels lipids into BAT. Activation of BAT might be a therapeutic approach to reduce elevated triglyceride concentrations and combat obesity in humans”. Our work shows similar results. Targeting *Mat1a* in HFD-fed mice, decreased serum TG levels, increased chylomicron TG clearance and increased radioactivity in BAT after gavage of ^3H -triolein (Fig. 4a, c and d of the manuscript and presented bellow as **Rebuttal Fig. 3**). Given that fatty acid oxidation is also increased, the results suggest that, in this context of dietary lipid abundance (HFD), serum lipids are fueling lipid oxidation in BAT.

Rebuttal Fig. 3. a) Serum triglyceride levels in HFD-fed *Mat1a* ASO and Control ASO treated mice. b) chylomicron triglyceride clearance in HFD-fed *Mat1a* ASO and Control ASO treated mice. c) dietary lipid distribution in HFD-fed *Mat1a* ASO and Control ASO treated mice. Statistical differences were indicated as * $p < 0.05$ and ** $p < 0.01$ (Student’s test).

The authors evidence of increased thermogenesis comes from Fig. 4F where they show that a beta3-AR agonist increases O₂ consumption more in Mat1a ASO mice than Control ASO mice. However, these results are compromised by the problem of not describing how this experimental protocol was conducted and how the data were analyzed. How was the beta3-AR agonist given to the mice while they were in the calorimeters? The results in Fig. 4F show no break in the collection of data associated with this event. More troubling is how the VO₂ consumption data are being scaled. The ordinate is labeled ml/kg/h, indicating that the O₂ consumption data is being scaled by BWs, which are ~15g lower in the Mat1a ASO group. As described in multiple manuscripts describing the correct approach to analyzing IDC data in recent years, big differences in BW confound group comparisons by over-amplifying the calculated EE in smaller animals within the groups.

Response: Thank you for your comments. We agree, this reviewer is correct, and we have now removed Fig. 4F in the Revised manuscript. This reviewer is also correct in pointing out about how we should be representing energy expenditure. There is certainly debate how we should be interpreting energy expenditures in mice with varying body weights. The ANCOVA analysis of the energy expenditure vs the body weight is widely accepted (for guidance we followed: Tschop M et al. A guide to analysis of mouse energy metabolism. *Nat Methods*. 2011 Dec 28;9(1):57-63). Therefore, we have performed this analysis (also following the recommendation of reviewer 1) for all the measurements of energy expenditure in the Revised manuscript. To study the effects of β -adrenergic agonist, we performed the ANCOVA analysis at 30 and 45 mins after the administration of the drug. We also measured *Ucp1* and *Pgc1a* expression in BAT after the after the β -adrenergic stimulation (**Supplementary Fig. 4c**).

Let me clarify the problem I'm having with these data. When the Mouse Metabolic Phenotyping Center consortium came into existence, they sought to develop a series of recommendations as to how indirect calorimetry experiments should be conducted so that investigators could obtain reliable, reproducible data. One of their key recommendations was that it was necessary to allow mice to equilibrate for 24 to 48 h in the calorimeters before stable, trustworthy measurements of VO₂ consumption and CO₂ production could be measured. Therein lies the problem with the authors methods. Did they inject mice with the beta-3 agonist, place them in the calorimeters, and measure their gas exchange 30 to 45 min

later and expect the reader to believe this is a valid assessment of the EE response of the two groups of mice to the treatment? Using ANCOVA to analyze the data is an improvement over their original methods but it does not solve the problem highlighted above.

New Response: Thank you very much for clarifying your concern. We are aware of the protocol using the indirect calorimetry system as it is one of the equipments that we use on daily basis and it has been essential for many of our publications (some recent examples in Lhomme T et al, *J Clin Invest* 2021; Quiñones M et al, *Redox Biol* 2021, Folgueira C et al, *Nat Metab* 2019). Mice were indeed placed in the metabolic cages 24 hours before injecting the beta-3-agonist. Actually, in the first version of the manuscript we submitted a graph representing the oxygen consumption 4 hours before injecting the beta-3-agonist (**Rebuttal Fig. 4**).

Rebuttal Fig. 4. Oxygen consumption in HFD-fed Control ASO and *Mat1a* ASO mice.

We changed this graph to show ANCOVA analyses in the second version as requested by the Reviewer (**Supplementary fig. 4b of the manuscript**). To avoid any type of confusion, we are now providing the graph representing oxygen consumption 24 hours before injecting the beta-3-agonist and after the treatment (**Rebuttal Fig. 5**). In this graph, it is possible to see that:

- a) Both Control and *Mat1a* ASO HFD show a higher oxygen consumption during the dark phase, meaning that these mice have the expected cycling.
- b) *Mat1a* ASO HFD mice show higher oxygen consumption than control mice, as described throughout the manuscript.

Rebuttal Fig. 5. Oxygen consumption in HFD-fed Control ASO and *Mat1a* ASO mice.

We apologize for misunderstanding the comment of the Reviewer and hope that showing this graph clarifies this point.

The data presented in Fig. 7 provide support for the idea that knocking down *Mat1a* in liver activates FGF21 release from the liver through a NRF2-dependent mechanism. The authors propose that this occurs after nuclear localization of NRF2 and transcriptional activation of FGF21. If the authors are able to detect nuclear localization of NRF2 with available reagents, they should confirm the direct role of NRF2 using ChIP assays to show that NRF2 is in fact being recruited to the FGF21 promoter. If NRF2 is necessary and sufficient for transcriptional activation of FGF21 after *Mat1a* ASO treatment, they should

also show that hepatic deletion of NRF2 blocks the induction of FGF21. In addition, more information is needed about the NRF2 inhibitor used in Fig. 7D. Its source needs to be identified, along with a reference to a publication demonstrating its efficacy and specificity.

Response: Thank you for this interesting suggestion. We have performed the CHIP assay in hepatocytes isolated from ASO treated HFD-fed mice. The results confirmed the binding of NRF2 in the *Fgf21* promoter (**Fig. 7d**). We also performed an *in vitro* experiment in which *Nrf2* was silenced by siRNAs, and found that silencing *Nrf2* abolished the increased FGF21 secretion that occurred when targeting liver *Mat1a* (**Supplementary Fig. 10b**). A similar result was observed when ML385 (a specific NRF2 inhibitor) was used (**Fig. 7a**), and we have now included information on this inhibitor in the Methods section. In brief, the NRF2 inhibitor ML385 is a probe molecule that binds to NRF2 and inhibits its downstream target gene expression. Specifically, ML385 binds to Neh1, the Cap 'N' Collar Basic Leucine Zipper (CNC-bZIP) domain of NRF2, and interferes with the binding of the V-Maf Avian Musculoaponeurotic Fibrosarcoma Oncogene Homolog G (MAFG)-NRF2 protein complex to regulatory DNA binding sequences. ML385 is a novel and specific NRF2 inhibitor (Singh A et al, ACS Chem Biol. 2016)

Specific Points

1. The description of the methods used to conclude that EE was affected by the treatments is inadequate. It refers to a previous publication but when that publication is examined it provides no details on how the calorimetry data were analyzed. The problem must be rectified in the current paper. Fig. 1G is used to make a claim that EE is increased by the MAT1A ASO but the data in the figure shows that EE was scaled by kg but does not define kg of what. Moreover, the accompanying figure panel indicates that there is no relationship between EE and BW over a weight range of 24-29 grams. This seems improbable since EE should increase roughly in proportion to BW raised to some power < 1 . An additional problem is that Fig. 1C shows that the two groups of mice being compared in Fig. 1G have approximately the same spread in BWs of mice within each group while Fig. 1C shows that the mice in the two groups were ~ 15 g different in BW. Together, this creates a lot of confusion for the reader, even for one experienced in the analysis of indirect calorimetry data. Please see <https://www.nature.com/articles/s41592-019-0513-9> for guidance in analyzing and presenting EE data.

Response: We have removed this publication and we have added another one. On the other hand, as mentioned above, taking into account the reviewer 1, this reviewer suggestions and the published information (Tschop M et al. A guide to analysis of mouse energy metabolism. *Nat Methods*. 2011 Dec 28;9(1):57-63), we agree that the ANCOVA analysis should be used for comparison of energy expenditure when body mass or composition differs between groups. Thus, ANCOVA has been used for all these analyses, including former Fig. 1g, **now Fig. 1h**.

The reference provided above is an update of the original article published by Tschop et al. 8 years later. Progress and challenges in analyzing rodent energy expenditure, Rodrigo Fernández-Verdejo, Eric Ravussin, John R. Speakman & Jose E. Galqani, Nature Methods volume 16, pages797–799 (2019)

New Response: Thank you. We have now changed the reference to this one.

2. OGTTs measure a composite of glucose absorption rates, alterations in B-cell sensitivity to glucose, and insulin-dependent glucose excursions. Given the relatively small improvement in insulin tolerance and only at the last time point, the substantial improvement in glucose tolerance shown in Fig 1E points to changes in the first two factors rather than solely insulin sensitivity. <https://pubmed.ncbi.nlm.nih.gov/18670420/> The 60 min time point shouldn't even be considered as a marker of insulin action. If you don't see an improvement in the first 20 min, it's probably not increased insulin action. What's more, the substantial improvement in glucose tolerance after Mat1a ASO is likely secondary to the loss in BW.

Response: Thank you for your comments. For this Revised manuscript, we have added New GTTs, ITTs and measured serum insulin levels in various time points (**Fig. 1f and supplementary Fig. 1h**). The levels of glucose in the ITTs have been represented as mg/dl, as recommended by the first reviewer; fasting insulin levels have also been quantified in the DIO induced mice (time 0 of the insulin levels of the GTT time points) (**Fig. 1f and supplementary Fig. 1g**) and in the *ob/ob* mice (**Fig. 5e**). In addition, the GTT, ITTs and measurements of insulin released during the GTTs have been performed at two different situations in the HFD-fed mice; 1) after the first injection of ASOs, when body weight loss is not that evident (**supplementary Fig. 1h**), and 2) by the end of the treatments (**Fig. 1f**). The results showed that after the first dose of ASOs (7th week of HFD), insulin levels were already lower before the bolus of

glucose in the GTT (time 0), and after 30 min of the glucose administration. The results showed that the insulin released in each time point and the levels during fasting were markedly lower when targeting *Mat1a*. Fasting levels of insulin were also decreased when *Mat1a* was targeted in the *ob/ob* mice. These collective results show that targeting liver *Mat1a* protects against insulin resistance.

3. What were the fasting insulin concentrations in the two groups of mice?

Response: Fasting insulin concentrations were measured as described above, under various conditions.

4. From looking at supplementary Fig. 1F, there appears to be a problem in the way the MAT1A KO mice responded to the HF diet. After it was introduced, the mice lost ~5 g in the first week and then stabilized their weight over the following 8 wks while the WT mice gained over 20 g. It would help the reader to understand the data better if the authors would present the actual BWs of the mice in the two groups and the actual mean food intakes for the two groups over time. Presenting the data as cumulative food intake makes it more difficult to see what is happening over time to food intake. This suggestion also applies to the MAT1A ASO mice, where it would be better to see the actual BWs of the mice in the two groups and their week by week food consumption rates. Looking at supplemental Figs 1f and 1g, it is hard to believe that mice differing in BW by 25 g are eating exactly the same amount of food. Lastly the actual BWs of the animals in each group is important to know relative to the analysis of the indirect calorimetry data. The authors should add another figure panel and show the actual BWs of the mice in each group.

Response: As the reviewer suggests we have included the BWs of all animal models (sometimes as supplementary figures due to the lack of space) and the food intake for all the different groups. In all these models, the results show always unchanged food intake. The energy expenditures are now all represented vs the body weight with the ANCOVA test performed so that the two parameters can be easily related. In all animal models tested, the inhibition of *Mat1a* was associated to higher energy expenditure. The fact that *Mat1a* ASO mice show a lower weight without changes in food intake is not that surprising, since many previous reports have found other animal models with a marked decrease in body weight in a feeding-independent manner. Just to mention some previously published examples

from coauthors of the manuscript (Folgueira C et al, Nat Metab. 2019) or from other groups (Pirzgalska RM et al, Nat Med. 2017).

On the other hand, we had the same thought when we showed this for the first time, so before sending the first version of the manuscript we performed again the analysis of the response of *Mat1a*-KO mice to the HFD; for this, we fed *Mat1a*-KO mice the HFD for a week and we obtained the same result. These results are shown in the figure below. Although we don't have a clear explanation for this, it seems that *Mat1a* KO mice are somehow protected against HFD-induced early metabolic dysfunctions. Indeed, this interesting question deserves to be investigated in a further study.

Figure: Body weight change and body weight of *Mat1a*-KO (n=3) and WT mice (n=5) fed a HFD for 7 days.

5. The variability of the Sirius Red and F4/80 measures is concerning, indicating that this approach to detecting fibrosis and inflammation is insensitive to the point of being useless.

Response: It is true that there is variability in the results from the histochemistry procedures; however, do not believe that histochemistries are 'useless'. We have now also analyzed the expression of genes involved in fibrosis and inflammation (**Supplementary Fig. 6a**). The heterogeneity of the results is also high but there were no statistical differences. We have performed the analysis also when using the *Mat1a* ASO2 (**Supplementary Fig. 6c**). The studies show that targeting *Mat1a* does not lead to any increase in the expression of genes involved in fibrosis or inflammation.

Let me further clarify my contention that the Sirius Red and F4/80 measures are of limited value by saying that when 0 is within the confidence interval around your mean, how can you conclude that your measurement is different from 0 (see Fig. 3A).

New Response: Thank you. There are few mice that present 0 for F4/80. However, the same heterogeneity of the data is present in the 3 groups and the analyzed expression of genes involved in fibrosis and inflammation reinforce this part (**Supplementary Fig. 6a of the manuscript**). We have now included a sentence in the results section to be aware of the dispersion of the data.

6. In Fig. 3D, triolein uptake is expressed per g of tissue. What do the results look like when corrected for differences in tissue weight between treatment groups?

Response: In the Figure below, we show the uptake of labelled lipids (DPM per total tissue or per mg of protein). The results confirm that targeting liver *Mat1a* leads to an increase in the uptake of dietary lipids in BAT.

Figure. Uptake of dietary lipids. A. uptake per tissue; B. uptake per mg of tissue protein. Abrev: C ASO: Control ASO, M ASO: *Mat1a* ASO.

7. There is a problem with the interpretation of the fatty acid oxidation data in the sense that the assays do not measure in situ rates of fatty acid oxidation. With a fixed amount of C14-palmitic acid present in the in vitro incubations, the assays are providing an estimate of Vmax or capacity of the tissues from the two groups to oxidize fatty acids. This is not a measure of the rate yet it is how the figures summarizing this data is labeled.

Response: We have performed the fatty acid oxidation analysis *ex vivo*, not *in vitro*, as previously reported and described in other published manuscripts (e.g. Hirschey MD et al, Nature 2010). The [1-¹⁴C]-palmitic acid that is added to the samples is always 0.5 µCi/ml, but this corresponds to a very low quantity of palmitic acid. Thus, we believe the reviewer is probably referring to the carrier that is usually added in the reaction, which is the cold palmitate (500 µM in the assay that has been used in this study), usually added to improve the signal and to promote the uptake of the radioactive substrate, given that the quantity of [1-¹⁴C]-palmitic acid is so low. In this assay, the flux or the rate of fatty acid oxidation has been measured as reported in multiple publications (e.g., Hirschey MD et al, Nature 2010; Frank K. Huynh et al. Methods Enzymol. 2014; Gao X et al. Biochim Biophys Acta. 2015; Lucía Barbier-Torres et al, Nat Comm. 2020; Gonzalez-Romero F, Mestre D et al, Cancer Res 2021). In addition, in our mice models, as frequently happens in other projects with different treatments or diseases, the amount of fatty acids that reach the mitochondria varies, so in order to determine the FAO rate, one should know the mitochondria capacity to catabolize those lipids. For this, an external amount of fatty acids should always be added in the same concentration in all the groups being analyzed. Of relevance in this study, is the FAO rate analysis in these conditions taking into account that the lipid that reaches the BAT mitochondria is higher when targeting liver *Mat1a*; however, the amount of lipid in BAT is lower than when mice are treated with the Control ASO. To get an accurate result of the flux or the rate of FAO, the researchers need to add the same amount of external fatty acid in the two groups that are being analyzed. This is something widely accepted to measure the FAO rate as reported in the manuscripts above. If the reviewer does not feel comfortable with the terminology FAO rate (which is because the units are always per hour), we will remove rate when describing the results.

The pedantic comments of the authors notwithstanding, they have continued to refer to FAO measurements in BAT explants in terms of rate, implying that actual in vivo rates of FAO in BAT are different (e.g., HFD-fed Mat1a ASO (Fig. 2b), Mat1a ASO2 (Supplementary Fig. 4a), and Mat1a KO mice

(Supplementary Fig. 4a), all exhibited an increase in FAO rate in BAT when compared to the corresponding controls. See narrative of P 7 of revised MS). They also indicate in the Fig. 2 legend that they are measuring FAO rate in BAT. This is an inaccurate reporting of what was actually measured, which was the rate of CO₂ production from BAT explants in a test tube using single fixed concentrations of a mixture of cold and C¹⁴-palmitate. The fact that this reaction is allowed to proceed for a fixed period of time is where the rate term enters into the calculation, and although reaction conditions have been standardized across groups, the calculated rate is an enzymatic rate and not the rate of BAT FAO that is being implied by the way this data is presented. This same problem extends to Fig. 3b where the authors again imply that they are measuring liver fatty acid oxidation rates when what they measured was FAO in hepatic explants. The implication that these are in vivo measurements is obvious and they are misleading.

New Response: We apologize if our comments looked pedantic, but that was not our intention at all. We understand the point raised by the Reviewer and in the new version of the manuscript all the BAT FAO rates have now been removed and have been substituted by FAO. Sorry for the misunderstanding.

This problem extends to the misleading description of de novo TG lipogenesis measurements. Fig. 3c legend indicates “Liver TG de novo lipogenesis determined by incorporation of [3H]-acetate into TG in HFD-fed control (n=7) and Mat1a (n=8) ASO-treated mice.” This legend strongly implies these were in vivo measurements but when one reads the Methods section it becomes clear that these measurements were made in liver explants. DNL is an in vivo process in which carbohydrates from circulation are converted into fatty acids that are then used for synthesizing either triglycerides or other lipid molecules, and there are well-defined methods for measuring the in vivo process (<https://www.nature.com/articles/1600744.pdf?origin=ppub>).

New Response: Once again, we agree with the Reviewer. The misleading description has now been changed, and it has been added to the text that this measurement was performed in liver pieces. We are sorry about this.

8. It's unclear what the isoproterenol-dependent increase in ex vivo lipolysis adds to the story and why the responses from the Control ASO CD mice were not included in Fig. 4d. Lipolytic responsiveness to beta-adrenergic agonists is impacted by a number of factors and what the authors

are showing is probably mostly dependent on the decrease in adiposity and fat cell size in the Mat1a ASO group.

Response: The assay of lipolysis is frequently performed in the presence and/or absence of the β -adrenergic receptors agonist isoproterenol as has been described in other studies (Bradlee L. Heckmann et al, J Biol Chem., 2014; Gao X et al. Biochim Biophys Acta. 2015), so that the capacity of that tissue under basal and stimulated conditions are measured.

In this Revised manuscript, we have added the control ASO CD mice and have calculated and represented the lipolysis results in nmol of fatty acid or glycerol released per mg of tissue protein; these will provide clarity on the influence of fat cell size or adiposity (Supplementary Fig. 3c).

The authors misunderstood the point of the question about this data. What does it add in terms of helping the reader to understand the resulting phenotype. Do the authors believe that knocking down hepatic MAT1a alters the responsiveness of EWAT to mobilize glycerol and FFAs in response to a supraphysiological concentration of a beta-agonist that activates all beta-adrenergic receptor subtypes? What does this data add to understanding the resulting phenotype? Fig. 2D suggests that the HF diet significantly diminishes the capacity of the adipocytes to release glycerol in response to iso. What is the basis for this difference and why wasn't FFA release similarly compromised in this group?

New Response: We understand the point of the Reviewer and we agree that, even though this assay is commonly used, it represents an artificial situation, as the dose of the beta agonist is indeed supraphysiological. To avoid misunderstandings and to send a clearer message to the reader, we have now removed the effect of the beta-agonist (**Fig. 2d and Supplementary fig.3c of the manuscript**) Thank you for the clarification.

9. Which WAT depot was used to produce the data presented in Fig 4? It doesn't seem to be specified in either the Methods or the fig legend. This is important because the capacity of WAT to undergo remodeling is depot specific.

Response: As has been frequently used for this kind of analysis, we have used epididymal WAT. This has been added to the methodology section.

10. As with Fig 1, please present the actual BW means of the mice in each group, as well as their actual food intakes each week.

Response: As mentioned above this have been performed.

11. *Ob/ob* mice typically have high fasting insulin levels by 12-15 wks of age. What impact did the 5-6 g difference in BW produced by the *Mat1a* ASA have on fasting insulin levels?

Response: We thank the reviewer for this suggestion. Now, we have been performed this analysis, levels of fasting insulin are markedly lower in *Mat1a* ASO than in the Control ASO *ob/ob* treated mice. In the new version of the manuscript has been added in **Fig. 5e**.

12. The problem noted above for FA oxidation measurements is also applicable in Fig. 5F. The authors are measuring the amount of palmitate oxidation with a fixed amount of palmitate and concluding that the measurement is indicative of *in vivo* fatty acid oxidation in BAT. The data do not fully support this conclusion.

Response: As mentioned above, the FA oxidation rate has been measured *ex vivo*. We did not perform *in vivo* fatty acid oxidation; this must be a misunderstanding. Our results show that FAO rate is increased in BAT when targeting liver *Mat1a*. FAO rate (nmol of ASM or CO₂ generated from palmitate per g of tissue and per hour) has been measured as described above. This assay measures the whole FAO rate and not just a measure of an enzyme activity. This method is well established in our lab (Lucía Barbier-Torres et al, Nat Comm. 2020; Gonzalez-Romero F, Mestre D et al, Cancer Res 2021) and is a well accepted assay (Hirschey MD et al, Nature 2010; Frank K. Huynh et al. Methods Enzymol. 2014; Gao X et al. Biochim Biophys Acta. 2015).

13. The authors measurement of PPAR α expression in response to *Mat1a* treatment is not an effective approach to determining whether the increase in serum FGF21 is due to PPAR α activation. The same criticism should be considered for measurement of nuclear ATF4 by Western, particularly since it is widely known in the field that reliable measurement of endogenous ATF4 by Western is difficult to

establish. The data in supplementary Fig 5 certainly do not meet the standard for ruling out involvement of either transcription factor in Mat1a ASO-dependent induction of hepatic FGF21.

Response: We also performed experiments with an inhibitor of PPAR α or ER stress. In this Revised Manuscript, additional experiments where isolated hepatocytes from ASO treated HFD-fed mice were exposed to siRNAs to silence *Ppara* and *Atf4* were performed (**Supplementary Fig. 10a**). Similar to prior experiments using PPAR α and of ER stress inhibitor, these new experiments showed that *Ppara* and *Atf4* are not involved in the increased FGF21 secretion by hepatocytes (**Supplementary Fig. 10b**). We also measured levels of PPAR α and ATF4 in the nucleus, and found that targeting *Mat1a* did not lead to changes in their levels of expression (**Supplementary Fig. 10c**). In concert, both these studies confirm that neither PPAR α nor ATF4 are involved in the increased secretion of FGF21.

14. In the last paragraph of the Results, the authors examine FGF21 in the media of hepatocytes from mice after knocking down Mat1a and adding SAME to the media. How much SAME was being added to the media and what proportion of the amount being added is ending up inside the cells? There seems to be an assumption that SAME is crossing the cell membrane. What is the evidence for that occurring?

Response: This is a very interesting point. We have now added 6mM of SAME to the media and have measured expression levels of *Bhmt*, *Sahh* and *Cbs mRNA*, genes involved in the methionine cycle (4 and 24 hours after adding SAME) and have also quantified SUMO1 and SUMO1-conjugated proteins (24 hours after adding SAME) (**Supplementary Fig. 11b**) (since SAME is a SUMOylation inhibitor; Pui Y. Lee-Law et al, J Hepatol 2020).

Additional Points

1. It would have been helpful if page numbers were provided in the MS.

Response: Page numbers are now provided.

2. Please define which oligos were used in supplementary table 1. Is ASO the control oligo and ASO2 the antisense oligo? In addition, measures of reproducibility must be provided.

Response: The ASOs sequence have now been provided in the methodology section and here:

Control ASO: 5'-CCTTCCCTGAAGGTTCTCC-3'

Mat1a ASO: 5'-CCACTTGTCATCACTCTGGT-3'

Mat1a ASO2: 5'-GCTCAGGAGACATTGACCAT-3'

IONIS Pharmaceuticals uses the same Control ASO. For this study, to knock-down *Mat1a* they created two different ASOs, which we have labelled as *Mat1a* ASO and *Mat1a* ASO2. In our preliminary studies, we found that the ASO dose required to silence 90% *Mat1a* was 25 mg/kg/week for *Mat1a* ASO and 50 mg/kg/week of *Mat1a* ASO2. The majority of studies were performed with the *Mat1a* ASO. *Mat1a* ASO2 were used in some experiments to confirm changes.

3. Please specify the temperature at which the mice were housed.

Response: Mice were housed at 21-22 °C, this has been added now in the methodology section.

Reviewer #3 (Remarks to the Author):

The authors have done an outstanding job addressing the prior critiques, with new data and revisions. The resulting manuscript is quite strong and the data paint an intriguing picture. I only have a few minor comments.

New Response: Thank you very much.

1. Abstract: *The abstract should reframe the work a little to to highlight the importance of the energetic phenotype. For example, the sentence "By using antisense oligonucleotides (ASO) and genetic depletion of Mat1a, we demonstrate that Mat1a deficiency in diet-induce obese or genetically obese mice prevented and reversed obesity and obesity-associated insulin resistance and hepatosteatosis" could have a phrase like "by increasing energy expenditure in an FGF-21 dependent fashion" included. After all the authors have shown nicely that the effects of MAT1 deficiency seem to hinge on the ability of FGF21 to increase BAT metabolism.*

New Response: Thank you for this suggestion, we have included this sentence

2. I am still not enthusiastic about the use of the word "channeled" to describe the increase in FA uptake and oxidation. Can the authors rephrase this?

New Response: We have changed "channeled" to "mobilized"

3. The data from Supp Fig 8a/b should be either shown in the main text or described in the main text more clearly (i.e % decrease/increase).

New Response: We have now described in detail in the results section the data showed in the Supp Fig. 8a/b.

4. It would be helpful to scale the Y-axis for the EE vs BW graphs similarly (Fig 6e, Sup Fig 9d).

New Response: In our opinion to scale the Y-axis similarly in these two figures let some graphs not as clear as we would like (Figures bellow, **Rebuttal Fig. 6**). Instead, since the scale was not exactly the same in each figure, we have now scaled exactly the figures in **Fig. 6e and Supplementary Fig 9d of the manuscript.**

a

Figure 6e in the manuscript

b

Supplementary Figure 9d in the manuscript

Rebuttal Fig. 6. Y-axis scaled equally in a) Fig. 6e and b) Supplementary Fig. 9d.

REVIEWERS' COMMENTS

Reviewer #2 (Remarks to the Author):

The authors have addressed each of the questions raised in the previous review.